# A cocktail nanovaccine targeting key entry glycoproteins elicits high neutralizing antibody levels against EBV infection

Ling Zhong[1,7], Wanlin Zhang[1,7], Hong Liu[2,7], Xinyu Zhang[1,7], Zeyu Yang[3], Zhenfu Wen[3], Ling Chen[1], Haolin Chen[3], Yanran Luo[1], Yanhong Chen[2], Qisheng Feng[1], Mu-Sheng Zeng ®[1], Qinjian Zhao[4], Lixin Liu ®[3], Claude Krummenacher ®[5] ✉, Yi-Xin Zeng ®[1] ✉, Yongming Chen ®[6] ✉, Miao Xu ®[1] ✉ & Xiao Zhang ®[1,4] ✉

Epstein-Barr virus (EBV) infects more than 95% of adults worldwide and is closely associated with various malignancies. Considering the complex life cycle of EBV, developing vaccines targeting key entry glycoproteins to elicit robust and durable adaptive immune responses may provide better protection. EBV gHgL-, gB- and gp42-specific antibodies in healthy EBV carriers contributed to sera neutralizing abilities in vitro, indicating that they are potential antigen candidates. To enhance the immunogenicity of these antigens, we formulate three nanovaccines by co-delivering molecular adjuvants (CpG and MPLA) and antigens (gHgL, gB or gp42). These nanovaccines induce robust humoral and cellular responses through efficient activation of dendritic cells and germinal center response. Importantly, these nanovaccines generate high levels of neutralizing antibodies recognizing vulnerable sites of all three antigens. IgGs induced by a cocktail vaccine containing three nanovaccines confer superior protection from lethal EBV challenge in female humanized mice compared to IgG elicited by individual NP-gHgL, NP-gB and NP-gp42. Importantly, serum antibodies elicited by cocktail nanovaccine immunization confer durable protection against EBV-associated lymphoma. Overall, the cocktail nanovaccine shows robust immunogenicity and is a promising candidate for further clinical trials.

Epstein-Barr virus (EBV) belongs to the gamma-herpesvirus subfamily and is carried by almost all adults worldwide[1]. EBV primary infection in young children and adolescents remains asymptomatic, but young adults infected by EBV often develop infectious mononucleosis[2]. EBV infection also leads to a 32-fold higher risk of developing multiple sclerosis[3,4]. In addition, EBV, as the first identified oncogenic human virus, is associated with Burkitt's lymphoma, Hodgkin's lymphoma, natural killer/T-cell lymphoma, gastric carcinoma and nasopharyngeal

[1]State Key Laboratory of Oncology in South China, Collaborative Innovation Center for Cancer Medicine, Guangdong Key Laboratory of Nasopharyngeal Carcinoma Diagnosis and Therapy, Sun Yat-sen University Cancer Center, Guangzhou 510060, China. [2]Translational Medical Center of Huaihe Hospital, Henan University, Kaifeng 475004, China. [3]Center for Functional Biomaterials, School of Materials Science and Engineering, Key Laboratory for Polymeric Composite and Functional Materials of Ministry of Education, Sun Yat-sen University, Guangzhou, China. [4]College of Pharmacy, Chongqing Medical University, Chongqing, PR China. [5]Department of Biological and Biomedical Sciences, Rowan University, Glassboro, NJ, USA. [6]College of Chemistry and Molecular Science, Henan University, Zhengzhou 450046, China. [7]These authors contributed equally: Ling Zhong, Wanlin Zhang, Hong Liu, Xinyu Zhang. ✉e-mail: krummenacher@rowan.edu; zengyx@sysucc.org.cn; chenym35@mail.sysu.edu.cn; xumiao@sysucc.org.cn; zhangxiao@cqmu.edu.cn

carcinoma[5,6]. It is estimated that EBV causes approximately 230,000–350,000 new cases of malignancies worldwide and 130,000–200,000 deaths in 2020[7]. Although EBV causes heavy disease burdens, no prophylactic vaccine or therapeutic agents are available. Besides, EBV-infected B cells induce T cell responses against EBV-associated antigens as well as non-EBV tumor-associated antigens, which provides new strategies for future cancer therapy[8,9].

Previous EBV vaccine studies mainly focused on the most abundant glycoprotein gp350, which is required for EBV infection of B cells but not epithelial cells[10]. However, in cottontop tamarins and common marmosets or human clinical trials, sterile immunity was not achieved by gp350-based vaccines[10]. The entry complexes gHgL or gHgLgp42 and the fusogen gB play essential roles in the EBV infection process[11]. EBV mainly exhibits two tropisms toward epithelial cells and B cells relying on the surface density of complexes gHgL and gHgLgp42[12]. Membrane fusion signal is transmitted from gHgL binding to ephrin receptor A2 (EphA2) or gHgLgp42 binding to human leukocyte antigens class II (HLA-II) to fusogen gB to initiate infection of epithelial cells and B cells, respectively[11]. Sera of healthy EBV carriers contain antibodies that effectively bind to gHgL, gp42 and gB[13–15]. In this study, we confirm that anti-gHgL, anti-gB and anti-gp42 sera antibodies of healthy EBV carriers contribute to sera neutralizing ability. Moreover, monoclonal neutralizing antibodies (nAbs) targeting gHgL, gp42 and gB have been reported to protect humanized mice against EBV challenges[16]. Taken together, in addition to gp350, gHgL, gp42 and gB are the antigens with the highest potential for vaccine development[14,17–19].

Optimal antigen selection or combination and the induction of cellular immune responses are two important aspects of the development of effective vaccines against EBV. The entry process of EBV requires a well-orchestrated teamwork involving multiple glycoproteins[16,20]. The limited antigen selection in vaccine formulations is likely one of the reasons that previous clinical trials failed to generate sterile immunity. Broadening the antigen selection and combining more potent antigens in a single dose vaccine may enhance the anti-EBV immune response. Moreover, the induction of a T cell immune response has proven to be important for the protective effect of vaccines, especially vaccines against herpesviruses[21,22]. Indeed, T cell responses play predominant roles in eliminating EBV-infected cells and controlling EBV-induced lymphomas[23].

The immunogenicity of soluble protein antigens is weak so that adjuvant is needed to significantly reduce the dose of antigen and enhance or/and prolong the immune responses[24,25]. Recently, the molecular adjuvants based on PAMP (pathogen-associated molecular pattern), like the CpG (agonists of Toll-like receptor 9 (TLR-9)) and MPLA (agonists of TLR-4) were used widely for subunit-based vaccines[24,25]. However, the molecular adjuvant that strongly activates innate immune receptors may induce systemic toxicity and the soluble antigens are susceptible to enzymes in vivo[26]. Nanoscale vehicles provide an effective platform for targeted and controlled delivery of mRNA, subunit antigens and adjuvants for the development of new generation of vaccines[27–29]. Nanovaccines co-loaded with soluble antigens and adjuvants could protect antigens from rapid clearance, achieve a better pharmacokinetics and pharmacodynamics profile, enhance antigen immunogenicity and avoid systemic side effects[30–32]. Besides, nanovaccines effectively induce T cell immune responses[33,34]. Nanovaccines that allow encapsulation of antigens with adjuvants combine advantages that will benefit the development of prophylactic vaccination against EBV.

Here, we developed EBV nanovaccines using 1,2-dioleoyl-3-trimethylammonium propane (DOTAP), poly (lactic-co-glycolic) acid (PLGA) and 1,2-distearoyl-sn-glycero-3-phosphoethanolamine-N-[(polyethylene glycol)-2000] (DSPE-PEG2000) to co-deliver adjuvants CpG and MPLA with EBV antigens gHgL, gB and gp42. All three nanovaccines with single antigens (gHgL, gB or gp42) induced potent and durable adaptive immune responses in mice attributed to efficiently transported to LNs, enhanced antigen internalization and maturation of dendritic cells, and induced robust germinal center (GC) responses. Serum antibodies induced by these nanovaccines had higher neutralizing titers than antibodies induced by non-encapsulated antigens due to enhanced production of antibodies targeting immunodominant epitopes. Importantly, NP-cocktail, which contains all three individual nanovaccines induced more potent neutralizing antibodies to block EBV infection in vivo and in vitro than each nanovaccine alone.

Collectively, our data demonstrate that nanovaccine immunization is highly effective in inducing significant anti-EBV immune responses. Importantly, a cocktail of nanovaccines is superior in providing complete protection in vivo and can be considered as an attractive formulation for further evaluation in clinical trials.

## Results

### Anti-gHgL, gB and gp42 antibodies contribute to the neutralizing ability of sera from EBV healthy carriers

Recombinant forms of the ectodomains of EBV gHgL, gB, gp42 and gp350 were expressed in 293F cells to ensure proper posttranslational modifications (Fig. S1). These proteins were used in ELISA to determine antibody levels and in vaccine preparation. To determine the neutralizing contribution of each glycoprotein specific antibodies, we first tested anti-gp350, anti-gHgL, anti-gB, and anti-gp42 IgG titers and neutralizing titers of sera from healthy EBV carriers (Table S1). The binding IgG titers against specific glycoproteins ranged from 4-6 (log10) (Fig. 1A). Sera antibodies induced by natural infection in healthy EBV carriers neutralized EBV infection of B cells (mean lgID50 at 1.76) and epithelial cells (mean lgID50 at 1.96) in vitro (Fig. 1B). Then we performed serum antibody depletion assay by 293T cells overexpressing each glycoprotein. After depletion, the IgG titers against gp350, gHgL, gB, and gp42 were decreased by more than 90% (Fig. S2). Using B cell and epithelial cell infection models, we determined the neutralizing titer of each serum before and after the depletion of over 90% of each glycoprotein-specific antibody. Compared to undepleted sera, the B cell neutralizing activities decreased by approximately ~ 37.72%, ~ 16.53%, ~ 23.18% and ~ 16.95% after depletion of gp350, gHgL, gB and gp42-specific antibodies, respectively (Fig. 1C). The epithelial cell neutralizing titers decreased on average by ~ 3.04%, ~ 45.13%, ~ 27.14% and ~ 3.65% after depletion by gp350, gHgL, gB and gp42-specific antibodies, respectively (Fig. 1C). As anticipated, gp350- and gp42-specific antibody depletion did not change the ability of sera to neutralize infection of epithelial cells since these glycoproteins are not needed for EBV infection of such cells (Fig. 1C). However, gp42-specific antibodies in sera were also important contributors to B cell neutralization (Fig. 1C). The key roles of gB as the fusogen in EBV infection of B cells and epithelial cells was reflected by the fact that gB-specific antibodies in sera contributed to neutralization in both epithelial cells and B cells infection (Fig. 1C). Consistent with a previous study by Bu. et al., our data showed that, in sera from healthy EBV carriers, gp350-specifc antibodies are the major contributor (~ 37.72%) to neutralize EBV infection of B cells while gHgL-specific antibodies are the most important antibodies (~ 45.13%) to neutralize EBV infection of epithelial cells[14].

Although anti-gp350 antibodies are the major contributors to B cell neutralization, gp350-based vaccines failed to completely block EBV infection in clinical trials and gp350-specific nAb 72A1 did not efficiently neutralize EBV infection in humanized mice[10,14,35]. According to our data and the previous studies, depletion of gHgL-specific, gB-specific and gp42-specific antibodies from sera reduced their ability to neutralize EBV infection in vitro to various degrees[14,15]. Besides, multiple nAbs targeting gHgL, gB and gp42 have been isolated and characterized[16]. Thus, gHgL, gB and gp42 are potential antigens for the development of efficient EBV vaccines. We separately encapsulated gHgL, gB or gp42 with a combination of two adjuvants (CpG and MPLA) to develop three nanovaccines.

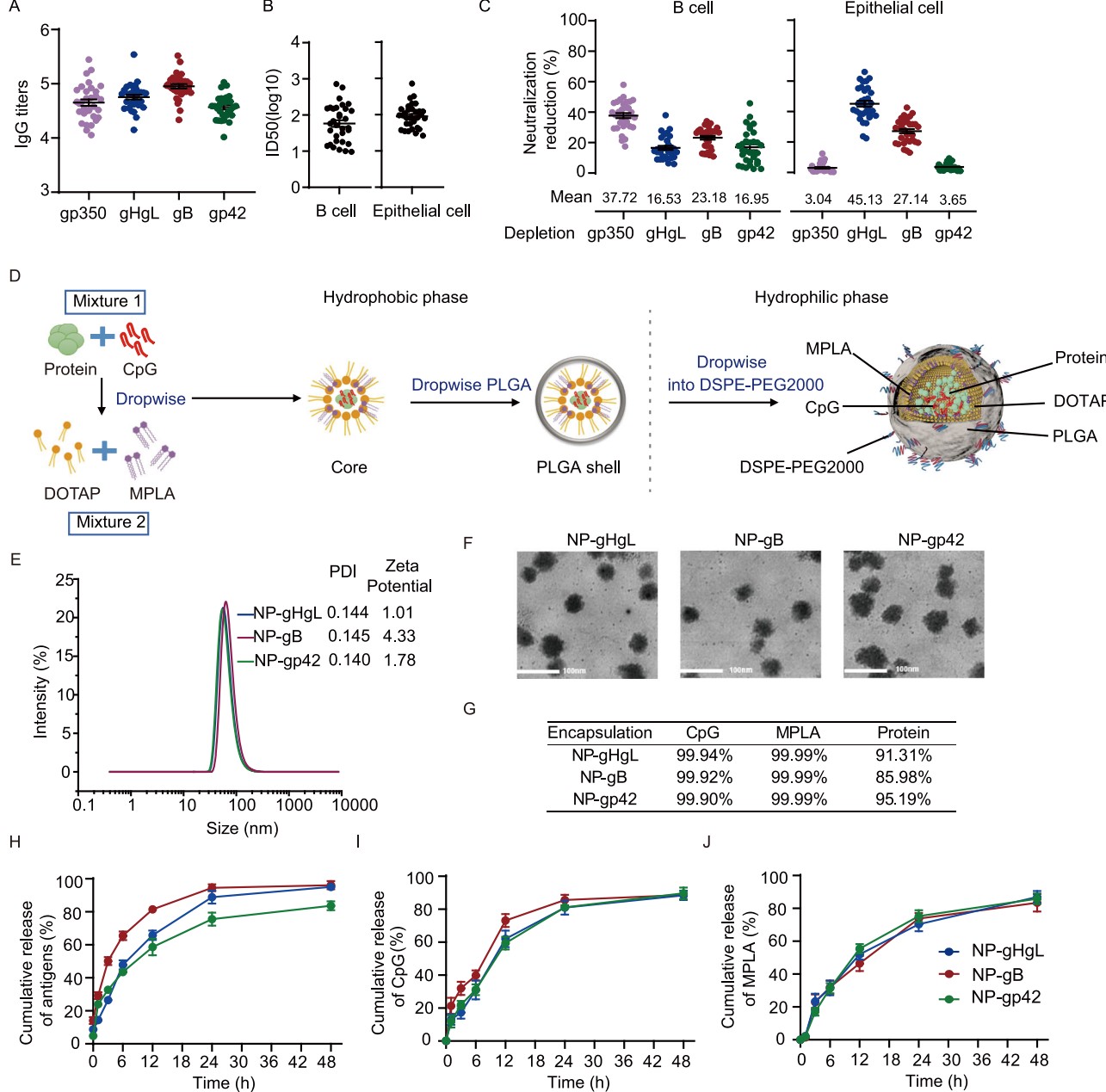

**Fig. 1 | gHgL, gB and gp42 are potential antigens to develop nanovaccines against EBV infection. A** Anti-gp350, anti-gHgL, anti-gB and anti-gp42 IgG titers in sera of 32 healthy EBV carriers. Data points are shown as the mean ± SEM. **B** Neutralization of CNE2-EBV infection of Akata B cells (left panel) and Akata-EBV infection of HNE1 cells (right panel) by sera collected from 32 healthy EBV carriers. Half maximal inhibitory dilution fold (ID50) was calculated by sigmoid trend fitting. Data points are shown as the mean ± SEM. **C** Reduction of neutralizing ability of EBV infection of Akata B cells and HNE1 cells by sera after glycoprotein-specific antibodies depletion. The percentage of neutralization reduction was calculated by (1-ID50-depleted/ID50-before) × 100%. Data points are shown as the mean ± SEM (*n* = 32). **D** Schematic illustration of nanovaccine preparation. The figure elements were created from Biorender.com. **E** Size, polydispersity index (PDI) and zeta potential of the NP-gHgL, NP-gB and NP-gp42 nanovaccines. **F** Transmission electron microscope (TEM) images of NP-gHgL, NP-gB and NP-gp42 nanovaccines (Scale bar = 100 nm) (*n* = 3). **G** Efficacy of encapsulation of antigens and adjuvants into NP-gHgL, NP-gB and NP-gp42. The average percentages of encapsulation are shown for three independent experiments for each nanovaccine. Cumulative release of antigens (**H**), CpG (**I**) and MPLA (**J**) in vitro. NP-gHgL, NP-gB and NP-gp42 were diluted in PBS with Tween 80 and incubated at 37 °C under continuous shaking at 100 rpm. Data represent the mean ± SEM of three independent experiments of each nanovaccine. Source data are provided as a Source Data file. DOTAP, 1,2-dioleoyl-3-trimethylammonium propane; MPLA, monophosphoryl lipid A; PLGA, poly (lactic-co-glycolic) acid; DSPE-PEG2000, 1,2-distearoyl-sn-glycero-3-phosphoethanolamine-N-[(polyethylene glycol)−2000].

## Preparation and characterization of nanovaccines

The nanovaccines (NP-gHgL, NP-gB and NP-gp42) with a core-shell structure were prepared by co-encapsulating gHgL, gB or gp42 respectively and two adjuvants, CpG and MPLA, using DOTAP, PLGA and DSPE-PEG2000 (Fig. 1D)[36]. Antigen and CpG were self-assembled with DOTAP via electrostatic interaction to form the cores and MPLA was encapsulated through hydrophobic forces. In the shell, PLGA was used to stabilize the core and DSPE-PEG2000 was used to improve water solubility. The hydrodynamic sizes of NP-gHgL, NP-gB and NP-gp42 were 88.9 nm, 81.0 nm and 72.0 nm, respectively (Fig. 1E). Moreover, NP-gHgL, NP-gB and NP-gp42 showed a narrow size distribution with 0.144, 0.145 and 0.140 polydispersity index (PDI) respectively and showed a positive charge at 1.01, 4.33 and 1.78 mV, respectively (Fig. 1E). These nanovaccines were pretty uniform in size

and morphology as shown by TEM (Fig. 1F). The size and immunogenicity of the nanovaccines were found to be stable at 4 °C for up to 30 days. (Fig. S3). Besides, NP-gHgL, NP-gB and NP-gp42 showed good biocompatibility in vitro (Fig. S4)[37].

Furthermore, the nano delivery system showed a high encapsulated efficiency for adjuvants (loading efficiency close to 100% for MPLA and CpG) and antigens (gHgL: 91.1%; gB: 84.8%; gp42: 95.2%) (Fig. 1G). The antigen release profiles were determined by incubating the nanovaccines in PBS with 0.1% w/v Tween 80[38]. The release profiles of the three nanovaccines showed sustained release in 24 h. During the first 6 h, 40–70% of the antigens were released together with 40% of the CpG and MPLA contents (Fig. 1H–J). The release of all three components continued slowly to reach its maximum at 24 h, which indicates that the nanovaccines may sustainably stimulate the immune system to enhance the magnitude, quality and persistence of immune responses (Fig. 1H–J)[25]. We also analyzed gB before encapsulation and gB released from nanoparticles. The appearance of both forms is consistent with a trimer under native non-reducing conditions (Fig. S5A). This indicates that all gB encapsulated and released was uniformly in a trimer conformation. Under reducing conditions, both forms of gB showed two bands (Fig. S5B). Collectively, we successfully developed three nanovaccines with a narrow size distribution, high encapsulation efficiency and slow-release kinetics of antigens gHgL, gB and gp42 and molecular adjuvants CpG and MPLA to induce potent immune responses.

## Nanovaccines induced potent and durable adaptive immune responses in mice

To assess the immunogenicity of nanovaccines, C57BL/6 J mice were immunized with various vaccine formulations three times subcutaneously at week 0, 2 and 4 (Fig. S6; Table S2). Free-form vaccines adjuvanted with CpG and MPLA (Free-gHgL, Free-gB and Free-gp42), alum-adjuvanted vaccines (Al-gHgL, Al-gB and Al-gp42), eNP (empty NP with no antigen, no adjuvant), eNP-C-M (NP, without antigen, with CpG and MPLA) and PBS were used as controls.

The negative controls, eNP and eNP-C-M, could not induce antigen-specific humoral or cellular immune responses (Fig. S7). To compare the immunogenicity of the various vaccine formulations, sera from immunized mice were collected at week 1, 3, 5, 8, 12, 20, 30 and 50 (Fig. S6, Table S2). The antigen-specific IgG titers were detected by ELISA (Figs. 2A–C and S11A–C). The nanovaccines, NP-gHgL, NP-gB and NP-gp42, induced significantly higher antigen-specific antibodies than the corresponding free-form groups, while the alum-adjuvanted vaccines elicited the lowest antibody titers (Figs. 2A–C and S11A–C). Neutralizing titers of sera collected at week 1, 3, 5 and 50 were evaluated using B cell and epithelial cell infection models (Fig. S8 and S9). Neutralizing titers increased after vaccine boosts (Fig. S8 and S9). After three immunizations, each nanovaccine induced higher titers of neutralizing antibodies in serum against both B cell infection and epithelial cell infection compared to Free- and Al- groups (Fig. 2D, E). B cell neutralizing titers in sera from NP-gHgL, NP-gB and NP-gp42 immunized mice were 1.9, 1.9 and 2.1 (log10), respectively (Figs. 2D and S8G–I; Table S3). In the B cell neutralizing assay, gp42 was the most effective antigen to induce neutralizing antibodies (Figs. 2D and S10). This is not surprising since gp42 plays a key role in receptor binding during B cell infection. As expected, in the epithelial cell neutralizing model, sera from mice immunized with vaccines containing gp42 did not inhibit infection (Figs. 2E and S9I). Epithelial cell neutralizing titers in sera from mice immunized with NP-gHgL and NP-gB were 2.9 and 2.7 (log10), respectively (Figs. 2E and S9G, H; Table S3). Moreover, at week 50, B cell neutralizing titers in sera from NP-gHgL, NP-gB and NP-gp42 immunized mice were 1.8, 1.8 and 2.0 (log10), respectively (Fig. S8J–L). At week 50, epithelial cell neutralizing titers in sera from mice immunized with NP-gHgL and NP-gB were 2.6 and 2.4 (log10), respectively (Fig. S9J–L). Interestingly, the NP

formulation of all three antigens strongly enhanced the production of IgA (Fig. 2F). Although C57BL/6 J mice are biased to induce Th1 immune responses, the immune polarization mainly determined by adjuvants according to our data[39]. Alum-adjuvanted vaccines still induced Th2-biased immune responses in the immunized mice (Fig. 2G). Nanovaccines and their corresponding free forms induced higher IgG2c and IgG2a titers but similar IgG1 titers compared to antigens adjuvanted with alum (Fig. S11D–F). According to the IgG2c/IgG1 ratios, nanovaccines and their free forms induced Th1-polarized responses, indicating that CpG and MPLA adjuvants tend to elicit Th1 responses (Fig. 2G). In addition, NP-gHgL, NP-gB and NP-gp42 immunization significantly increased the size of the memory B cell population in the spleen and among peripheral blood mononuclear cells (PBMCs) compared to other vaccine formulations (Figs. 2H, S12 and S13C).

To evaluate the cellular immune responses elicited by different vaccine formulations, the splenic lymphocytes were collected at week 5 and restimulated with specific antigens in vitro. After restimulation, IFN-γ+, TNF-α+ or IL-2+ CD4+ or CD8+ T cells were detected by flow cytometry (Fig. S14). Compared to non-encapsulated vaccines, nanovaccines induced robust T cell responses manifested by the increased number of antigen-specific TNF-α, IFN-γ and IL-2 producing T cells after restimulation (Fig. 2I–K). Moreover, nanovaccine formulations were the most efficient at increasing the effector memory CD4+ and CD8+ T cells populations in the spleen (Figs. 2L, S13A, B). Besides, nanovaccines induced higher levels of effector memory CD8+ T cells in peripheral blood and NP-gHgL elicited more effector memory CD4+ T cells in peripheral blood (Fig. S13D, E).

To address the cytotoxicity of nanovaccines in vivo, several tissues were collected from immunized mice one week after the third immunization (Figs. S6 and S15). Hematoxylin and eosin (HE) staining showed no detectable tissue damage or toxicity in the heart, liver, spleen, lung and kidney suggesting that these nanovaccines are safe in mice (Fig. S15A). The total cholesterol (TC), alanine transaminase (ALT), total bilirubin (TBIL), triglycerides (TG), aspartate aminotransferase (AST), alkaline phosphatase (ALP) and UREA levels of sera from nanovaccine immunized mice collected at week 5 were similar to the control PBS group, which further confirmed the safety of nanovaccines (Fig. S15B–H).

In summary, a combination of CpG and MPLA was overall more effective than the traditional alum adjuvant in inducing an immune response to EBV glycoproteins in mice. Most importantly, the encapsulation of antigens with CpG and MPLA into nanoparticles significantly enhanced the adaptive immune responses to gHgL, gB and gp42. Regarding neutralization, NP-gHgL and NP-gB elicited a high level of neutralizing antibodies to block infection of both B lymphocytes and epithelial cells. In contrast, NP-gp42 immunization induced the highest level of B cell neutralizing antibodies, but did not induce neutralizing antibodies that protected epithelial cells. This is consistent with the predominant role of gp42 in the infection of B cells only. All three nanovaccines induced antigen-specific T cell responses and increased the size of the population of memory B cells and effector-memory T cells in spleen.

## Nanovaccines efficiently targeted lymph nodes and promoted internalization and maturation of BMDCs as well as GC formation

To understand how the nanovaccines induced the potent immune responses, we generated nanovaccines containing Cy5-labeled antigens and two adjuvants. First, we evaluated the internalization efficiency of nanovaccines by bone marrow-derived dendritic cells (BMDCs). Internalization of immunogens by antigen presenting cells (APCs) is a key step for efficient humoral and cellular immune responses. The internalization efficiency of NP-gHgL, NP-gB and NP-gp42 uptake by BMDCs was determined by flow cytometry. BMDCs

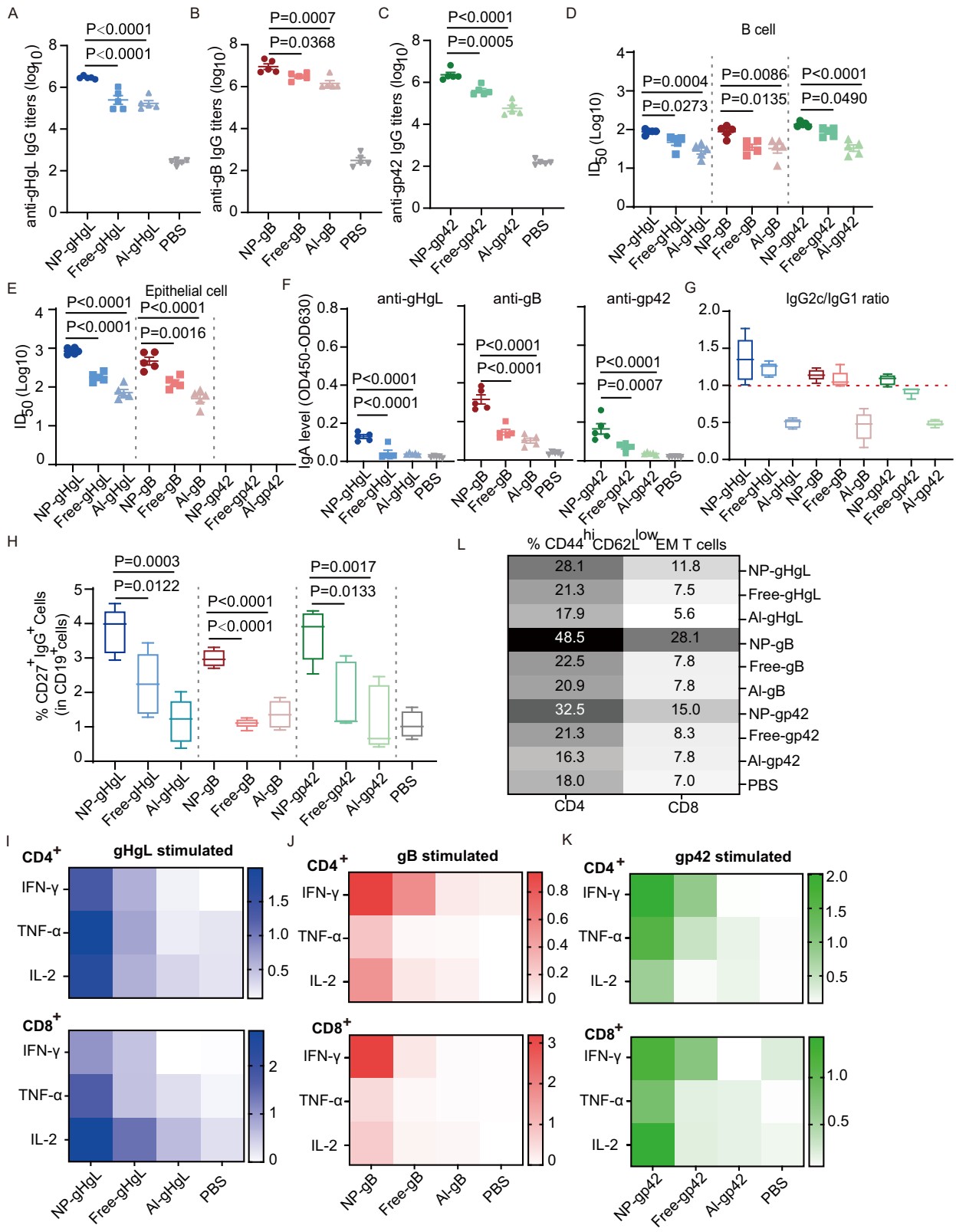

were incubated with nanovaccines or free-form vaccines containing Cy5-labeled antigens for 4 h. Cells incubated with PBS were served as negative control (mock). The percentages of Cy5-positive BMDCs after internalization of NP-gHgL, NP-gB and NP-gp42 were ~7.3-fold, ~4.1-fold and ~3.7-fold higher than the percentage of Cy5-positive cells exposed to the corresponding free forms, respectively (Figs. 3A and S16A). These data indicate that nanoparticle

encapsulation resulted in enhanced uptake of antigens (Figs. 3A and S16A). Following uptake, the nanovaccines stimulate the maturation of BMDCs to initiate antigen processing and presentation. After incubation with different vaccine formulations or controls for 24 h, expression of the co-stimulatory molecules CD80 and CD86 was measured by flow cytometry. Overall co-expression of these markers significantly increased in cells treated with nanovaccines compared to

**Fig. 2 | Nanovaccines induced potent humoral and cellular immune responses.**
**A** Anti-gHgL IgG titers induced by NP-gHgL, Free-gHgL, Al-gHgL and PBS, respectively. **B** Anti-gB IgG titers induced by NP-gB, Free-gB, Al-gB and PBS, respectively. **C** Anti-gp42 IgG titers induced by NP-gp42, Free-gp42, Al-gp42 and PBS, respectively. Neutralization of CNE2-EBV infection of Akata B cells (**D**) and Akata-EBV infection of HNE1 epithelial cells (**E**) by sera collected on day 35 from C57BL/6 J mice immunized with different vaccine formulations. Half maximal inhibitory dilution fold (ID50) was calculated by sigmoid trend fitting. Sera induced by gp42-based vaccines cannot neutralize EBV infection of epithelial cells in vitro. **F** Sera IgA levels (day 35) of C57BL/6 J mice immunized with different vaccine formulations. **G** IgG2c/IgG1 ratio of sera (day 35) from C57BL/6 J mice immunized with different vaccine formulations($n = 5$). **H** Memory B cells in the spleen of C57BL/6 J mice immunized with different vaccine formulations (day 35) were evaluated by flow cytometry($n = 5$). Antigen-specific CD4[+] (top panel) and CD8[+] (bottom panel) T cell

responses in the spleen on day 35 were measured by intracellular cytokine staining assay after restimulation with gHgL (**I**), gB (**J**) and gp42 (**K**) in vitro. The numbers indicate the percentage of IFN-γ[+], TNF-α[+] or IL-2[+] cells in CD4[+] T cells or CD8[+] T cells after restimulation with specific antigens in vitro. Data are shown as the mean percentage of IFN-γ[+], TNF-α[+] or IL-2[+] cells in CD4[+] T cells or CD8[+] T cells ($n = 5$). **L** Effector memory (EM) CD4[+] T cells (left panel) and CD8[+] T cells (right panel) in the spleen on day 35 were evaluated by flow cytometry (strategy in Fig. S12). Data are according to the mean percentage of CD4[+] and CD8[+] EM T cells ($n = 5$). **A–F** Data points are shown as the mean ± SEM ($n = 5$). **G, H** The center line indicates the median, upper and lower box lines show quartiles; and whiskers show the maximum and minimum values. *P* values calculated using one-way ANOVA with Dunnett's multiple comparison are shown as precise values. Source data are provided as a Source Data file.

cells exposed to free-form or alum-adjuvanted (Al) vaccines (Figs. 3B and S16B). This effect is partially due to the nanoparticles itself, which also induced the up-regulation of CD80 and CD86 in BMDCs without adjuvants (Fig. S16D)[40]. In addition, the expression of MHC-II was also significantly enhanced after incubation with each nanovaccine (Figs. 3C and S16C). These data indicate that nanovaccines comprising EBV entry glycoproteins significantly stimulated BMDCs maturation and antigen presentation after being efficiently internalized by BMDCs.

Next, we evaluated how nanovaccines were transported and distributed in the host. The effectiveness of vaccination depends on the vaccine's ability to induce adaptive immune responses, which are mainly initiated in lymph nodes (LNs)[41]. Hence, targeted delivery of vaccines from injection sites to LNs is essential for inducing effective adaptive immune responses[42]. The nanovaccines that are less than 100 nm in size could be directly drained to LNs within hours after administration[42,43]. To evaluate the transportation of NP-gHgL, NP-gB and NP-gp42 to LNs, 5 µg Cy5-labeled gHgL, gB or gp42 were encapsulated with both adjuvants to prepare nanovaccines, NP-gHgL, NP-gB or NP-gp42, respectively. The simple mixture of Cy5-labeled gHgL, gB or gp42 and the two adjuvants were named Free-gHgL, Free-gB or Free-gp42, respectively. These formulations were injected subcutaneously into C57BL/6 J mice at the root of the tail. All three nanovaccines were efficiently transported to the proximal (inguinal) and distal (axillary) LNs 6 h post-injection (Fig. 3D). By comparison, less fluorescence associated with free-form antigens was detected in LNs (Fig. 3D). Furthermore, more antigens from NP-gHgL, NP-gB and NP-gp42 could be found in the cortex and paracortex of inguinal LNs, where B and T cells are located (Figs. 3E and S16E). The nanovaccines were retained in LNs for at least 24 h whereas accumulation in other organs remained largely undetectable (Fig. S17). The high accumulation and prolonged retention of NP-gHgL, NP-gB and NP-gp42 in LNs are important properties to induce strong and long-lasting immune responses.

The follicular helper CD4[+] T (Tfh) cells play crucial roles in the induction of durable protective humoral immune responses[44,45]. Nanovaccines more efficiently promoted Tfh cell differentiation compared to the Free- and Alum-groups in spleen (Fig. 3F). Finally, we detected the germinal centers (GCs) formation induced by different vaccine formulations. GCs are defined as clusters of GL7[+] activated T and B cells surrounded by a mantle of IgD[+] naive B cells[46]. GC formation is critical for the generation of high-affinity antibodies and memory B cells, which are responsible for long-term protection after immunization[47]. After three immunizations, NP-gHgL, NP-gB and NP-gp42 more efficiently induced GCs responses than other vaccine formulations (Fig. 3G). The robust humoral immune responses elicited by NP-gHgL, NP-gB and NP-gp42 might be attributed to the potent GCs formation.

Overall, the responses to nanovaccines observed in APCs, lymph nodes and spleen may explain how NP-gHgL, NP-gB and NP-gp42, induced robust adaptive immunity.

## A cocktail nanovaccine induced more potent neutralizing antibodies

The EBV infection cycle is complex. EBV infects B cells and epithelial cells through distinct mechanisms that necessitate multiple glycoproteins[11,20]. Infection of B cells involves gHgLgp42 binding to HLA-II and triggering gB, while infection of epithelial cells involves gHgL binding to EphA2 and triggering gB (Fig. S18A)[48]. Hence, to develop a vaccine with an enhanced protective effect, we combined the three nanovaccines to prepare a cocktail vaccine (NP-cocktail) by mixing three NPs at 1:1:1 ratio (Fig. S18A; Table S2).

The same immunization procedure described above was applied (Fig. S6; Table S2). A cocktail of the free-form vaccine (Free-cocktail) and alum-adjuvanted cocktail vaccine (Al-cocktail) as well as PBS were used as controls (Table S2). The combined antigens in NP-cocktail did not impact each other's immunogenicity and titers of anti-gHgL, anti-gB and anti-gp42 IgG were similar to those obtained after separate immunizations (Figs. 4A–C and 2A–C). NP-cocktail elicited higher total IgG titers against gHgL, gB and gp42 than the Free-cocktail and Al-cocktail (Figs. 4A–C and S18B). As for the induction of cellular immune response, NP-cocktail was much more efficient than Free-cocktail and Al-cocktail (Figs. 4D and S18C, D).

High levels of neutralizing antibodies were induced by NP-cocktail vaccination compared to Free-cocktail and Al-cocktail since sera from NP-cocktail immunized mice showed higher neutralizing titers for both B cell and epithelial cell infection (Figs. 4E, F and S19, S20). The B cell neutralizing titer induced by NP-cocktail was 290.6, which was ~ 3.2, ~ 2.8 and ~ 2.1-fold higher than NP-gHgL, NP-gB and NP-gp42 alone, respectively (Fig. 4G). The epithelial cell neutralizing titer for sera of NP-cocktail immunized mice was 2674.6, which was ~ 3.1 and ~ 5.2-fold higher than NP-gHgL and NP-gB, respectively (Fig. 4G). Thus, considering the complexity of EBV infection, broadening antigen spectrum is important for generation of potent neutralizing antibodies targeting different EBV envelop glycoproteins, which may completely block EBV infection in vivo.

As seen in separate immunizations, NP-cocktail and Free-cocktail induced Th1-biased immune responses, which were reflected by a high IgG2c/IgG1 ratio (Fig. S21A–D). Furthermore, NP-cocktail elicited higher IgA levels in sera compared to other vaccine formulations, which may also contribute to neutralization of EBV infection (Fig. S21E)[49]. Effective GCs formation was observed in draining LNs of NP-cocktail immunized mice (Fig. S22). Fewer GCs formation were observed after immunization with Free-cocktail and Al-cocktail (Fig. S22). Moreover, NP-cocktail vaccination induced more memory B cells and effector memory T cells in the spleen (Fig.S23A–C) and more memory B cells and effector-memory CD8[+] T cells were also found in peripheral blood (Fig. S23D–F).

Considering that gHgL, gB and gp42 are involved in the key steps during EBV entry, we next used a virus-free cell-cell fusion assay[14] to evaluate the fusion inhibitory abilities of sera antibodies elicited by different vaccine formulations. Effector cells expressing EBV

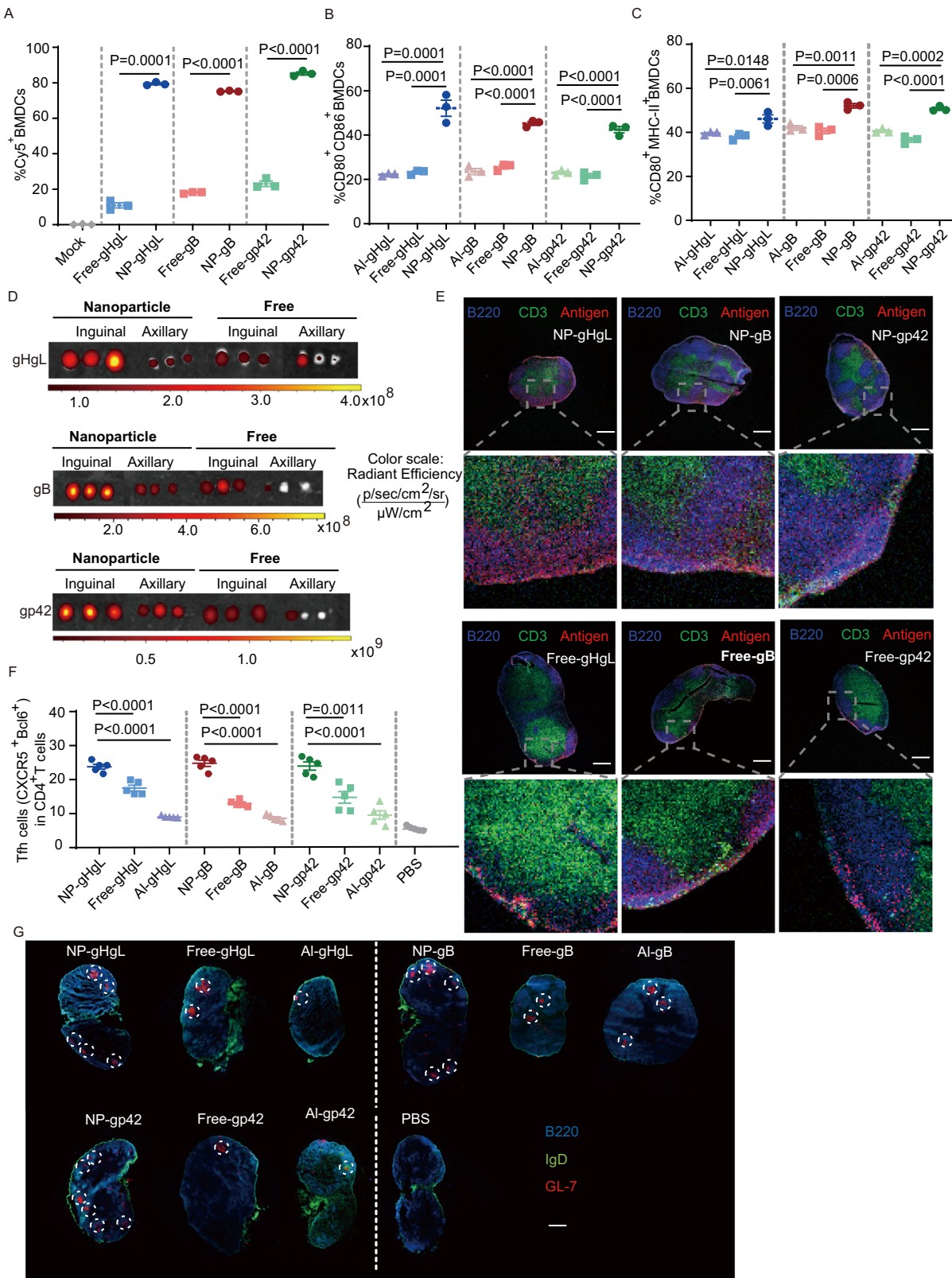

glycoproteins were mixed with target epithelial or B cells carrying endogenous receptors. A decrease in cell-cell fusion in the presence of sera reflects their fusion-inhibitory potential. Serum antibodies induced by NP-cocktail, Free-cocktail and NP-gp42 efficiently blocked viral glycoprotein-mediated B cell fusion (Fig. 4H). Serum antibodies induced by NP-cocktail more efficiently inhibited viral glycoprotein-mediated and epithelial cell fusion than other vaccine formulations

(Fig. 4I). Immunization with NP-gp42 induced antibodies blocked B cell fusion but not epithelial cell fusion, because gp42 is not required for epithelial cell infection (Fig. 4I).

We also compared the immunogenicity of NP-gHgL, NP-gB, NP-gp42 and NP-cocktail with that of NP-gp350 (Fig. S24). As expected, sera from mice immunized with NP-gp350 showed no neutralizing ability toward EBV infection of epithelial cells since gp350 is not

**Fig. 3 | Nanovaccines delivery to lymph nodes and enhancement of BMDCs uptake and maturation as well as GC formation. A** Cy5[+] BMDCs after 4 h incubation with PBS (Mock), free or nanoparticle forms of Cy5-labeled gHgL, gB and gp42. Percentages of positive cells was determined by flow cytometry in three independent experiments (*n* = 3). Expression detection of CD80 and CD86 (**B**) and MHC-II and CD86 (**C**) on BMDCs incubated for 24 h with free (antigen mixed with MPLA and CpG adjuvants), Al (antigen without NP with alum) or nanoparticle (NP with CpG, MPLA and antigen) forms of gHgL, gB and gp42 (*n* = 3). **D** Lymph nodes (LNs) images of mice immunized with vaccines containing Cy5-labeled antigens (*n* = 3). C57BL/6 J mice were injected with 10 μg free or nanoparticle (NP) forms of Cy5-labeled gHgL, gB or gp42. Axillary (distal, further away from the vaccine injection site) and inguinal (proximal, near the vaccine injection site) LNs were harvested after 6 h, and Cy5 fluorescence was measured using the IVIS optical imaging system (*n* = 3). **E** Confocal images of inguinal LNs at 6 h post-immunization (Scale bar = 400 μm). LNs were stained with Alexa Fluor 594-B220 to detected B cells and FITC-CD3 to detected T cells (*n* = 3). **F** Tfh cells generation in spleens of mice immunized by different vaccine formulations on day 35. Each group contains five female C57BL/6 J mice (*n* = 5). **G** Germinal center formation in inguinal lymph nodes of C57BL/6 J mice immunized with different vaccine formulations. LNs were harvested on day 35 after the first of three immunizations and stained with Alexa Fluor 594-IgD, Alexa Fluor 488-B220 and Alexa Fluor 647-GL7 (Scale bar = 400 μm) (*n* = 5). NP-: nanovaccine containing antigen and two adjuvants CpG and MPLA. Free-: antigen mixed with two adjuvants CpG and MPLA. Al-: antigen mixed with Alum adjuvants. **A–C** and **F** Data are shown as mean ± SEM. Source data are provided as a Source Data file. **A** P values calculated using unpaired two-tailed Welch's *t* test are shown as precise values. **B**, **C** and **F** P values calculated using one-way ANOVA with Dunnett's multiple comparison are shown as precise values.

involved in the epithelial cell infection process (Fig. S24B)[16]. In contrast, NP-gp350 elicited antibodies efficiently blocked EBV infection of B cells (Fig. S24C). The B cell neutralizing titers of sera from NP-gp350 immunized mice were similar to those from NP-gHgL, NP-gB and NP-gp42 immunized mice (Fig. S24D). Interestingly, although antibodies against gp350 are the major contributors to neutralization of EBV infection of B cells in healthy EBV carriers, immunization with NP-cocktail, which does not contain NP-gp350, induced higher B cell neutralizing titers (Fig. S24D)[14]. NP-cocktail targeting key entry glycoproteins induced higher sera neutralizing titers compared to NP-gp350 may result from the essential roles of gHgL, gp42 and gB in EBV infection process[11].

The multi-target cocktail vaccines induced robust humoral and cellular immune responses. Importantly, NP-cocktail elicited higher neutralizing titers than separate NP-gHgL, NP-gB and NP-gp42 formulations. NP-cocktail induced the production of antibodies that more efficiently blocked cell-cell fusion, reflecting their inhibitory effect on EBV glycoproteins involved in key steps during entry. Interestingly, increasing the dose of a single antigen (gHgL) only slightly increased the neutralizing titers of sera from immunized mice (Fig. S25A, B). Hence, broadening the antigen spectrum rather than solely increasing the dose of the single antigen is a superior strategy to improve neutralizing abilities, considering that EBV infection involves multiple glycoproteins. These data demonstrated the importance of combining antigens to develop an efficient vaccine against EBV.

## Nanovaccines induced more antibodies targeting immunodominant epitopes

To evaluate the epitopes targeted by antibodies generated by different vaccine formulations, the competition ELISA was used to assess the ability of a panel of nAbs against gHgL, gB or gp42 to block the binding of antibodies from sera of immunized mice (collected at week 5).

In the case of gHgL, antibodies AMMO1, 6H2 and 10E4 recognize different gHgL epitopes and neutralize EBV infection through different mechanisms. AMMO1 targets gH domain I/II and gL[50], 6H2 targets gH domain IV[51] and 10E4 targets gH domain I and gL[52]. AMMO1 effectively protected humanized mice against EBV challenge and rhesus macaques against oral transmission of rhesus lymphocryptovirus, an ortholog of EBV[35]. Besides, 6H2[51] and 10E4[52] also neutralized EBV infection in vitro and in vivo. Therefore, the epitopes recognized by AMMO1, 6H2 and 10E4 represent key vulnerable sites of gHgL. The chimeric 6H2 antibody did not react with anti-mouse HRP-conjugated secondary antibody, thus it would not interfere with the results of the competitive ELISA (Fig. S26). More than 40% of the gHgL-binding antibodies induced by NP-cocktail or NP-gHgL were blocked by AMMO1 (Fig. 4J). This indicates that a large portion of antibodies elicited by the nanovaccines detected, at least partly, the epitope targeted by AMMO1. Importantly, compared to Free- and Al- groups, NP-gHgL and NP-cocktail generated a larger proportion of antibodies targeting the immunodominant neutralizing epitopes defined by AMMO1, 6H2 and 10E4 (Fig. 4J).

In the case of gB, antibody 3A5 recognizes domain IV, while antibodies 3A3 and AMMO5 recognize different epitopes located in domain II[15,53]. Antibodies 3A3 and 3A5 potently neutralized EBV infection of B cells and epithelial cells, and conferred complete protection in a humanized mouse model[15]. AMMO5 only neutralized EBV infection of B cells effectively in vitro but also protected humanized mice from EBV challenge[15,53]. Moreover, the epitopes recognized by 3A3 and 3A5 are the major targets of naturally acquired anti-gB antibodies in healthy EBV carriers[15]. The majority of anti-gB antibodies induced by NP-cocktail and NP-gB recognized the epitopes of either 3A3 or 3A5, as indicated by the competing ability of these nAbs to block the detection of gB by antibodies from sera of mice immunized by these nanovaccines (Fig. 4K). Again, compared to Free- and Al-formulations, NP-groups generated a higher proportion of antibodies against the neutralizing epitopes recognized by 3A3, 3A5 and AMMO5 (Fig. 4K). This indicates that nanovaccines enhance the generation of antibodies against dominant neutralizing epitopes in domains II and IV of EBV gB.

In the case of gp42, previously reported monoclonal antibodies against the N-terminus are non-neutralizing[54], while antibodies 5E3[52], 1A7[55] and 6G7[55] against the C-terminus show robust B cell neutralizing ability. Hence, the C-terminus of gp42 is a crucial target for B cell neutralization. Competition assay showed that a large proportion of antibodies induced by NP-cocktail and NP-gp42 could be blocked by antibodies 5E3 and 6G7 (Fig. 4L), which have shown potent B cell neutralizing abilities and protected humanized mice from EBV-associated lymphoma[52,55]. Again, compared to Free- and Al- formulations, NP-vaccines generated a higher proportion of serum antibodies against the vulnerable sites defined by the nAbs at the gp42 C-terminus (Fig. 4L).

Overall, nanovaccines whether containing a single antigen or a cocktail of three antigens, generated a higher proportion of serum antibodies against neutralizing epitopes identified by effective monoclonal antibodies, compared to Free- and Al-formulations (Fig. 4J–L). The enhanced targeting of these dominant neutralizing epitopes explains why sera from mice immunized with nanovaccines more effectively neutralized EBV infection in vitro (Fig. 4E, F). In contrast, free-form and alum-adjuvanted vaccines all induced a lower proportion of antibodies against these epitopes, which may result in the lower neutralizing efficiency of sera from mice immunized with such formulations (Fig. 4E, F and J–L). These data demonstrate that nanovaccines more efficiently induced desirable neutralizing antibodies to EBV entry glycoproteins.

## Nanovaccines elicited potent humoral immune responses in rabbits

We next compared the immunogenicity of NP-gHgL, NP-gB, NP-gp42, NP-cocktail and Free-cocktail in rabbits. PBS was used as the negative control. Rabbits were immunized with different vaccine formulations three times intramuscularly at two-week intervals and blood was collected on day 14, 28 and 42 (Fig. 5A).

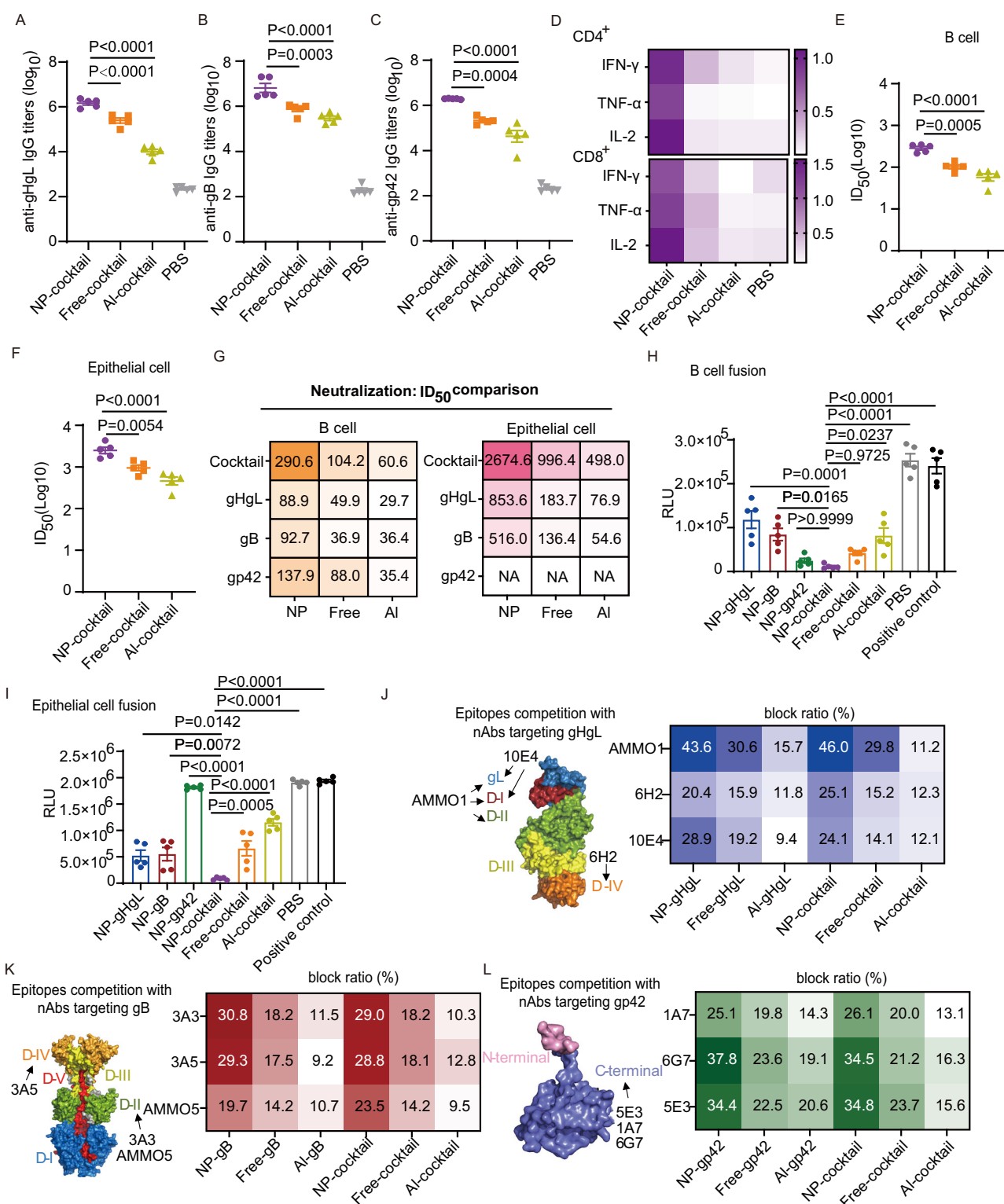

Consistent with the results from immunization of mice, the multi-target NP-cocktail vaccine generated similar antigen-specific antibody titers compared to individual NP-gHgL, NP-gB and NP-gp42 (Figs. 5B–D and S27A–C). The Free-cocktail formulations also generated lower titers of antigen-specific antibodies than the nanovaccines in rabbits (Fig. 5B–D and S27A–C). Neutralizing titers against B cell infection and epithelial cell infection increased after the two boosts (Fig. S27D–I). After the third immunization, the B cell infection and epithelial cell infection neutralizing titers of sera induced by NP-cocktail are 2.8 (log10) and 3.5 (log10), respectively

(Figs. 5E, F and S27H, I). Individually, NP-gHgL, NP-gB and NP-gp42 vaccination induced similar B cell neutralizing antibody titers in rabbits (2.4 (log10), 2.4 (log10) and 2.5 (log10), respectively) (Fig. 5E; Table S3). Among the different vaccine formulations, NP-cocktail induced the highest B cell neutralizing titers (Fig. 5E). In parallel, NP-gHgL vaccination induced higher epithelial cell neutralizing titers (3.2; log10) than NP-gB (2.8; log10) (Fig. 5F; Table S3). These data indicate that nanoparticle encapsulation is beneficial to generate serum antibodies that more efficiently neutralized EBV infection in vitro in rabbits than the Free-group.

**Fig. 4 | Cocktail nanovaccine elicits more robust immune responses against EBV antigens in mice.** Total anti-gHgL (**A**), anti-gB IgG (**B**) and anti-gp42 IgG (**C**) titers of sera collected on day 35 from C57BL/6 J mice immunized with different vaccine formulations. **D** Antigen-specific CD4$^+$ (top panel) and CD8$^+$ (bottom panel) T cell responses on day 35 (5 weeks) in the spleen of mice immunized with the indicated vaccines. Antigen specific T cells were measured by intracellular cytokine staining assay after restimulation with gHgL, gB and gp42 in vitro ($n = 5$). The numbers indicate the percentages of IFN-γ$^+$, TNF-α$^+$ or IL-2$^+$ cells in CD4$^+$ T cells or CD8$^+$ T cells after restimulation in vitro ($n = 5$). Neutralization of CNE2-EBV infection of Akata B cells (**E**) and Akata-EBV infection of HNE1 epithelial cells (**F**) by sera collected on day 35 from five C57BL/6 J mice immunized with different vaccine formulations. **G** Comparison of B cell neutralizing titers (left) and epithelial cell neutralizing titers (right) of sera collected on day 35 from C57BL/6 J mice immunized with cocktail or individual vaccines. The indicated half maximal inhibitory

dilution fold (ID50) values were calculated by sigmoid trend fitting. Data are shown as mean ID50 ($n = 5$). Inhibition of B cell-cell fusion (**H**) and epithelial cell-cell fusion (**I**) by sera collected on day 35 from C57BL/6 J mice immunized with different vaccine formulations. Effector cells which were not incubated with any sera were used as positive control. RLU, relative luminescence units. Antibody binding competition of sera antibodies and gHgL-specific nAbs (**J**), gB-specific nAbs (**K**) and gp42-specific nAbs (**L**) were defined by ELISA. Glycoprotein structures show the location of neutralizing epitopes. The data represent the ability of each indicated monoclonal neutralizing antibody to compete with serum antibodies collected on day 35 from C57BL/6J mice immunized with different vaccine formulations. The block ratios are shown as the mean block ratio for each condition ($n = 5$). Source data are provided as a Source Data file. **A**–**C**, **E**, **F**, **H** and **I** Data points are shown as the mean ± SEM ($n = 5$). *P* values calculated using one-way ANOVA with Dunnett's multiple comparison are shown as precise values.

Body weight and temperature were recorded weekly and both remained stable in rabbits from all groups throughout the experiment (Fig. 5G, H). Furthermore, no tissue damage or inflammation was observed in the heart, liver, spleen, lung and kidney tissues of immunized rabbits (Fig. S28A). Besides, the level of biochemical functional indicators remained stable in the sera from nanovaccine immunized rabbits (Fig. S28B–H). These results indicate that nanovaccines were safe for rabbits.

## IgG induced by NP-cocktail protected humanized mice from lethal EBV challenge

We further evaluated whether the antibodies generated by different vaccines in rabbits could protect humanized mice against a lethal EBV challenge. The immunized rabbits were euthanized at day 42 and the serum IgGs from blood pooled from three rabbits of each group were purified by protein G affinity (Fig. S29A, B). For each purified pool of IgG, the half-maximal effective concentration (EC50) value was determined for binding to their respective antigens by ELISA (Fig. S29C–E). The EC$_{50}$ values for purified IgG induced by NP-cocktail and individual NPs were similar (Fig. S29C–E). By comparison, EC50 values for IgG elicited by Free-cocktail were generally higher, thus reflecting again a lower efficacy of this formulation (Fig. S29C–E). We then determined the neutralization titers in vitro using B cells and epithelial cells as before. The B cell and epithelial cell neutralizing titers of purified IgG elicited by NP-cocktail were 18.6 μg/mL and 2.7 μg/mL, respectively (Fig. 6A). The B cell neutralizing titers of IgG induced by NP-cocktail was ~17-fold, ~8-fold, ~6-fold and ~10-fold higher than that induced by NP-gHgL, NP-gB, NP-gp42 and Free-cocktail, respectively (Fig. 6A). The epithelial cell neutralizing titers of IgG induced by NP-cocktail was ~5-fold, ~61-fold and ~33-fold higher than titers induced by NP-gHgL, NP-gB and Free-cocktail, respectively (Fig. 6A). Overall the data indicate that NP-cocktail is superior at eliciting neutralizing antibodies, even when the total amount of serum IgG is similar between NP-cocktail and individual NPs. These purified IgGs were further evaluated in vivo using a humanized mouse model of EBV infection and tumor formation.

Human T cells, B cells and monocytes are maintained over a long period in CB17-SCID mice reconstituted with human peripheral blood leukocytes (PBL)[56]. However, SCID mice injected intraperitoneally with $5 \times 10^7$ PBL from healthy EBV seropositive donors developed fatal EBV lymphoproliferative disease derived from human B cells[57]. Therefore, we developed a humanized mouse model based on SCID mice engrafted with a lower amount of PBMCs from healthy EBV seropositive donors with undetectable EBV copy numbers (Fig. S30A–E)[58,59]. PBMCs used for humanized mice reconstitution were susceptible to EBV infection in vitro (Fig. S31)[45,60,61]. At least 48 h after infection, the majority of B cells in donors' PBMCs were infected by EBV and T cells only killed a fraction of infected B cells (Fig. S31C, D).

Briefly, 4-week-old SCID-beige mice were irradiated and treated with clodronate liposomes 24 h before being engrafted through

intraperitoneal injection with $1.5 \times 10^7$ human PBMCs (Fig. 6B). 4 weeks after engraftment, the percentage of human CD45$^+$ cells, T cells (CD3$^+$) and B cells (CD19$^+$) in PBMCs were determined to assess whether mice were successfully reconstituted with human cells (Fig. S32). Sufficient numbers of human CD45$^+$ cells, T cells and B cells were detected in PBMCs of engrafted mice indicating that we successfully generated humanized mice suitable for EBV challenge experiments. Figure S33 shows the percentages of human CD45$^+$ cells, T cells and B cells in these mice, as they were randomly distributed in groups to receive the IgG elicited from different vaccines. Humanized mice were injected intraperitoneally with rabbit IgG (1 mg/mouse) induced by different vaccines followed by a challenge with 25,000 green Raji units (GRU) Akata strain EBV 24 h later (Fig. 6B). Following the EBV challenge, body weight, survival ratio, EBV DNA copy numbers and antigen-specific IgG titers were monitored weekly. Clinically ill mice (>20% body weight loss) were euthanized. Humanized mice in the mock group were not challenged with EBV.

The titers of rabbit IgG injected in sera of humanized mice were measured weekly by ELISA over 8 weeks following injection (Fig. 6C–E). At the time of EBV challenge (1 day post-IgG injection), rabbit IgG titers against gHgL, gB and gp42 were similar for each group (Fig. 6C–E). Rabbit IgG generated by nanovaccines became undetectable in mouse sera after 4 weeks post-administration, while IgG induced by the Free-cocktail vaccine became undetectable at week 3 post-administration (Fig. 6C–E). The body weight of mice treated with rabbit IgG induced by PBS significantly decreased from 3 weeks post EBV challenge. In contrast, mice in the other groups, which all received rabbit IgG elicited by various vaccines, maintained a stable body weight (Fig. 6F). Survival analysis showed that 8 out of 8 mice treated with IgG induced by PBS died, while only 1 out of 8 mice died in each of the groups treated with IgG induced by NP-gHgL and NP-gB (Fig. 6G). All mice treated with IgG induced by NP-gp42, NP-cocktail and Free-cocktail survived until the end of the experiment (Fig. 6G). We then analyzed whether antibodies induced by the various vaccines could prevent EBV viremia by measuring EBV DNA in blood of the humanized mice following the challenge. All mice treated with IgG induced by NP-cocktail remained free of viremia (Fig. 6I, purple line). EBV DNA copy numbers in several mice treated with NP-gB and NP-gp42 were temporarily above the detection limited, but remained low overall (Fig. 6H). In contrast, 2 out of 8 mice treated with IgG induced by NP-gHgL, 2 out of 8 mice treated with IgG induced by Free-cocktail, 8 out of 8 mice received non-immune IgG (PBS control) showed sustained viremia (Fig. 6H, I). As expected, EBV DNA copy numbers remained undetectable in blood and tissue of mice from the unchallenged mock group (Fig. 6I, J). Similarly, EBV DNA copy numbers remained undetectable in spleens, mesenteric LNs and kidneys of all mice treated with IgG induced by NP-cocktail, indicating that IgG induced by this formulation is the most effective at preventing EBV infection in this model (Figs. 6J and S34). However, EBV DNA copy numbers increased in the spleens and LNs of one or two mice in each group treated with IgG

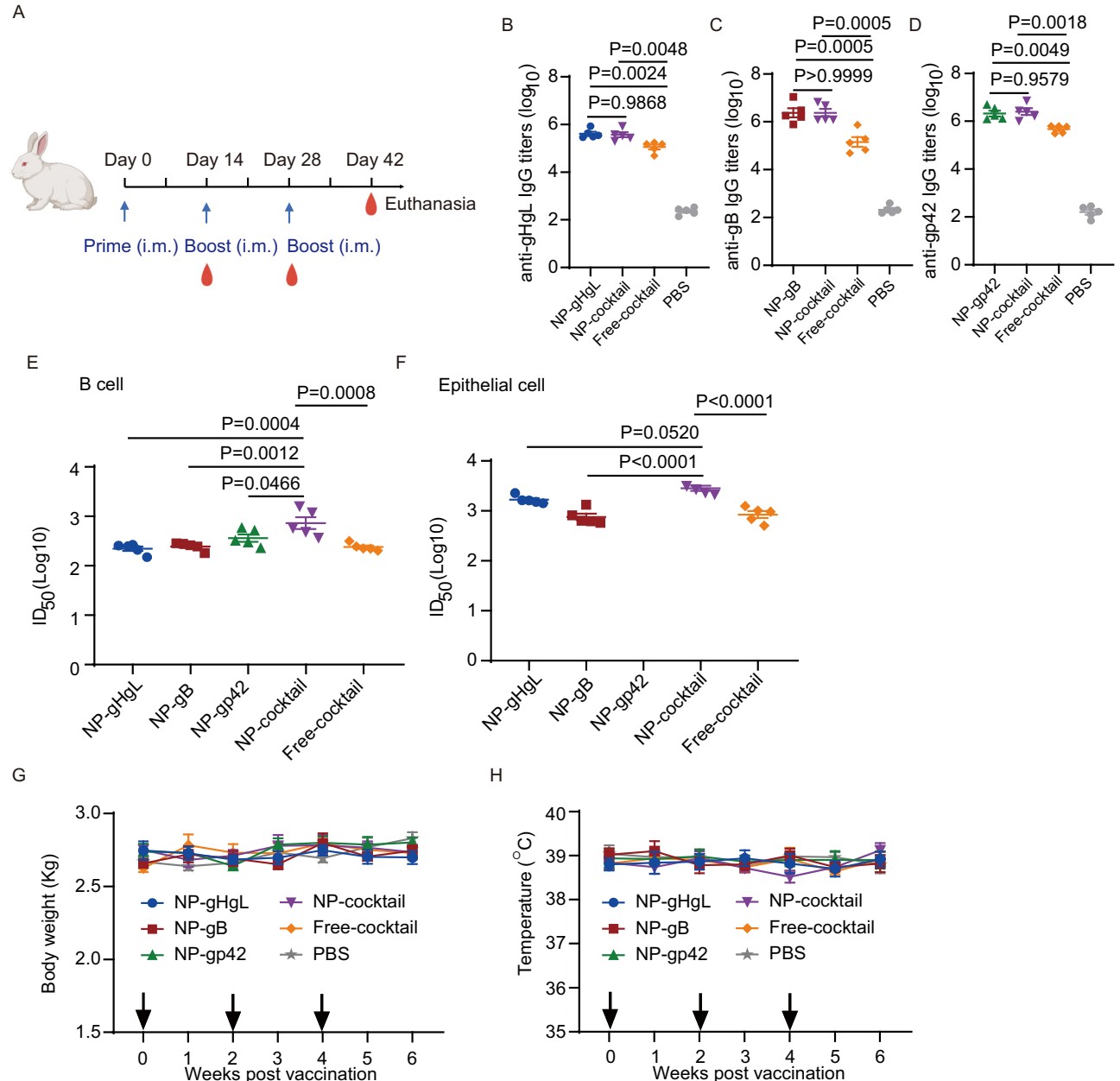

**Fig. 5 | Rabbit immunization and induction of humoral immune responses.** (**A** Schedule of rabbit immunization. The figure was created from Biorender.com. Rabbits were inoculated intramuscularly with different vaccine formulations three times on day 0, 14 and 28 with doses described in Table S2. Blood was collected on day 0, 14, 28 and 42. Rabbits were euthanized on day 42. **B**–**D** IgG titers of sera collected on day 42 from rabbits immunized with different vaccine formulations. **B** Anti-gHgL IgG titers elicited by NP-gHgL, NP-cocktail, Free-cocktail and PBS (**C**) Anti-gB IgG titers elicited by NP-gB, NP-cocktail, Free-cocktail and PBS. **D** Anti-gp42 IgG titers elicited by NP-gp42, NP-cocktail, Free-cocktail and PBS. Neutralization of CNE2-EBV infection of Akata B cells (**E**) and Akata-EBV infection of HNE1 epithelial cells (**F**) by sera collected on day 42 from rabbits immunized with different vaccine formulations. Half maximal inhibitory dilution fold ($ID_{50}$) values were calculated by sigmoid trend fitting. Body weight (**G**) and temperature (**H**) were monitored after immunization of rabbits with the indicated vaccine formulations. Black arrows indicate times of immunization. **B**–**H** Data points are shown as the mean ± SEM ($n = 5$). **B**–**F** $P$ values calculated using one-way ANOVA with Turkey's multiple comparison are shown as precise values. Source data are provided as a Source Data file.

elicited in response to nanovaccines containing single antigens or from Free-cocktail vaccine (Fig. 6J). Nevertheless, all IgG-treated groups showed reduced EBV DNA copy numbers compared to mice treated with non-immune IgG (PBS group) where all 6 mice showed a dramatic increase of EBV DNA in spleen and LNs.

Altogether, we evaluated the protective efficacy of rabbit IgG elicited by the various vaccine formulations that protected humanized mice against EBV challenge. Among the different vaccines, NP-cocktail was the most effective at generating IgGs able to control EBV infection in vivo.

## Protection against EBV-associated lymphoma in humanized mice

Next, we evaluated the pathological changes caused by the EBV challenge in humanized mice. The spleens of the majority of mice in treated groups were overall normal in morphology, without visible tumors (Fig. 7A). However, the spleens of 6 mice treated with non-immune IgG (induced by PBS) were obviously enlarged and heavier (Fig. 7A, B). The in situ hybridization of Epstein-Barr virus-encoded RNAs (EBERs) and hCD20 staining confirmed that a large number of CD20+EBER+ cells infiltrated the spleen of mice treated with IgG induced by NP-gHgL,

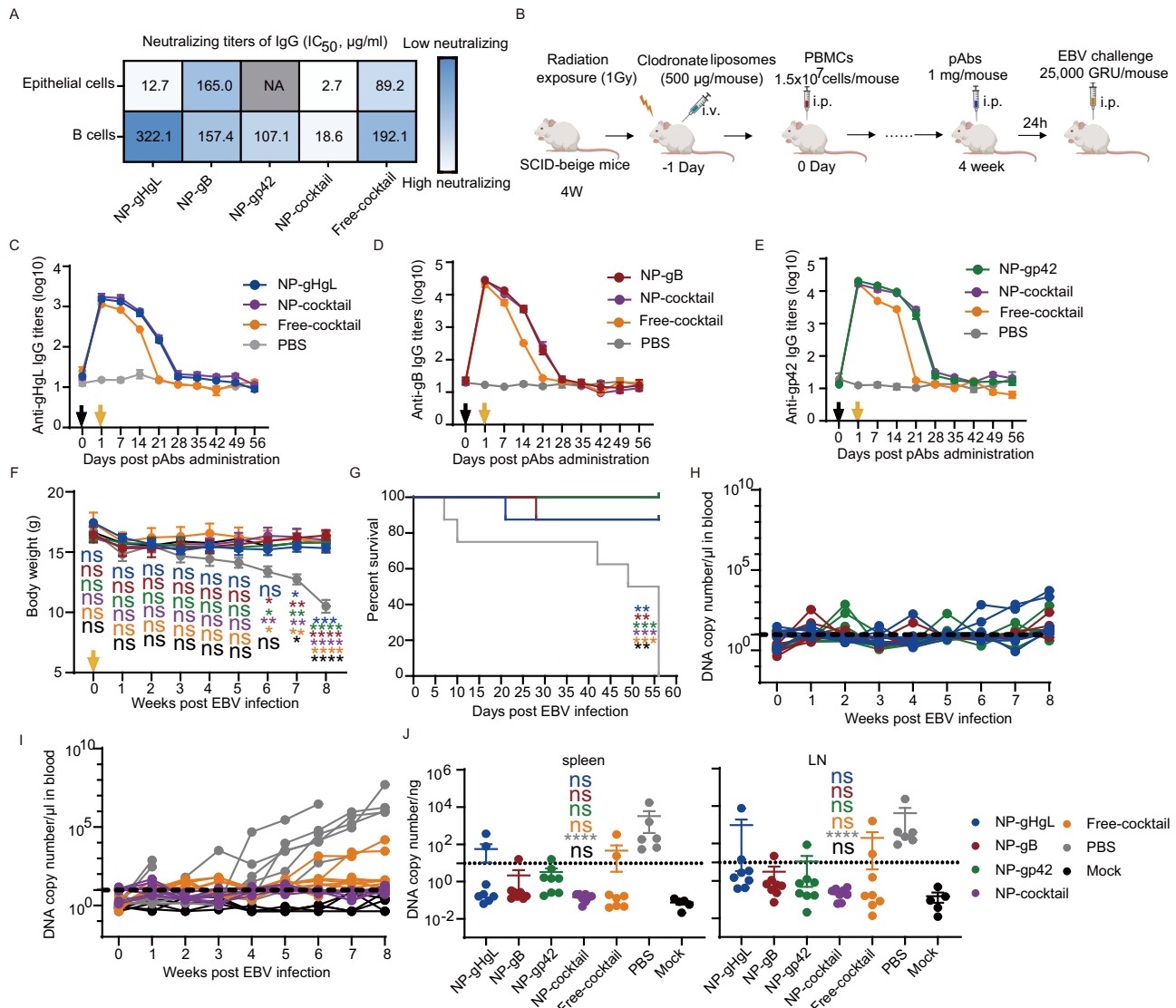

**Fig. 6 | Total IgG antibodies purified from rabbit sera following immunization with NP-cocktail protected humanized mice from lethal EBV challenge.**
**A** Neutralization of Akata-EBV infection of HNE1 epithelial cells (top panel) and CNE2-EBV infection of Akata B cells (bottom panel) by IgG purified from sera of three rabbits immunized with the indicated vaccines. The half-maximal inhibitory concentration (IC$_{50}$) was calculated by sigmoid trend fitting. **B** Schematic illustration of humanized mice reconstitution and passive protection against EBV lethal challenge. i.v., intravenously; i.p., intraperitoneally. 53 female SCID-beige mice were used. The figure was created from Biorender.com. Titers of anti-gHgL (**C**), anti-gB (**D**) and anti-gp42 (**E**) rabbit IgG in sera from humanized mice. The black arrow indicates the administration of IgG and the yellow arrow indicates the administration of EBV. Data are shown as mean ± SEM ($n = 8$). **F** Body weight of mice post-EBV challenge. The yellow arrow indicates the day of EBV infection. Data are shown as mean ± SEM. Humanized mice in the mock group were not challenged with EBV.

**G** Survival plot of humanized mice after EBV challenge. *P* values are shown in the Source data file. The color of the asterisks denotes the statistical difference. **H, I** EBV DNA copy numbers in peripheral blood of each group. Each line represents an individual mouse and the dashed line indicates the detection limit. **H, I** Summarize the same experiment and data have been separated in two graphs for clarity. **J** EBV DNA copy numbers in the spleen (left panel) and mesenteric lymph nodes (right panel) of humanized mice treated with different IgG. The dashed line indicates the detection limit. Samples of two mice in the PBS group which died on day 7 and 10 were not collected because of decay. Data are shown as mean ± SEM ($n = 8$ for the NP-gHgL, NP-gB, NP-gp42, NP-cocktail and Free-cocktail groups, $n = 6$ for the PBS group and $n = 5$ for the mock group). **F, J** Statistical analysis was performed using one-way ANOVA with Dunnett's multiple comparison and precise *P* values are shown in the Source data file. The color of the asterisks or ns denotes the statistical difference. Source data are provided as a Source Data file.

Free-cocktail and PBS (Fig. 7A). Sporadic CD20⁺EBER⁺ cells were detected in the spleen of mice treated with IgG induced by NP-gB and NP-gp42 while no CD20⁺EBER⁺ cells were observed in mice treated with IgG induced by NP-cocktail (Fig. 7A). Tumors were observed in the abdominal cavity of 1 out of 8 mice treated with IgG induced by NP-gHgL, 2 out of 8 mice treated with IgG induced by Free-cocktail and 5 out of 8 mice treated with non-immune IgG (induced by PBS) (Fig. 7C–H). Each one of these tumors contained high EBV DNA copy numbers and a large number of human CD45⁺ cells, T cells and B cells (Figs. 7D–G and S18). The tumor cells were CD20⁺EBER⁺, which are

typical of EBV-associated lymphoma (Fig. 7H). As expected, no spontaneous tumor formation was observed in the uninfected mock group. Importantly, mice treated with IgG induced by NP-cocktail, NP-gB and NP-gp42 were all tumor free. Besides, no obvious tumor was observed in the heart, liver, lung and kidney of all mice (Fig. S35).

Finally, human CD45⁺ cells, T cells and B cells were detected in PBMCs, peritoneal lavage fluids, LNs and spleen of mice from each group (Figs. S32 and S36). These data confirmed that human T cells and B cells could be maintained over long periods in a humanized mouse model based on SCID-beige mice engrafted with PBMCs from healthy

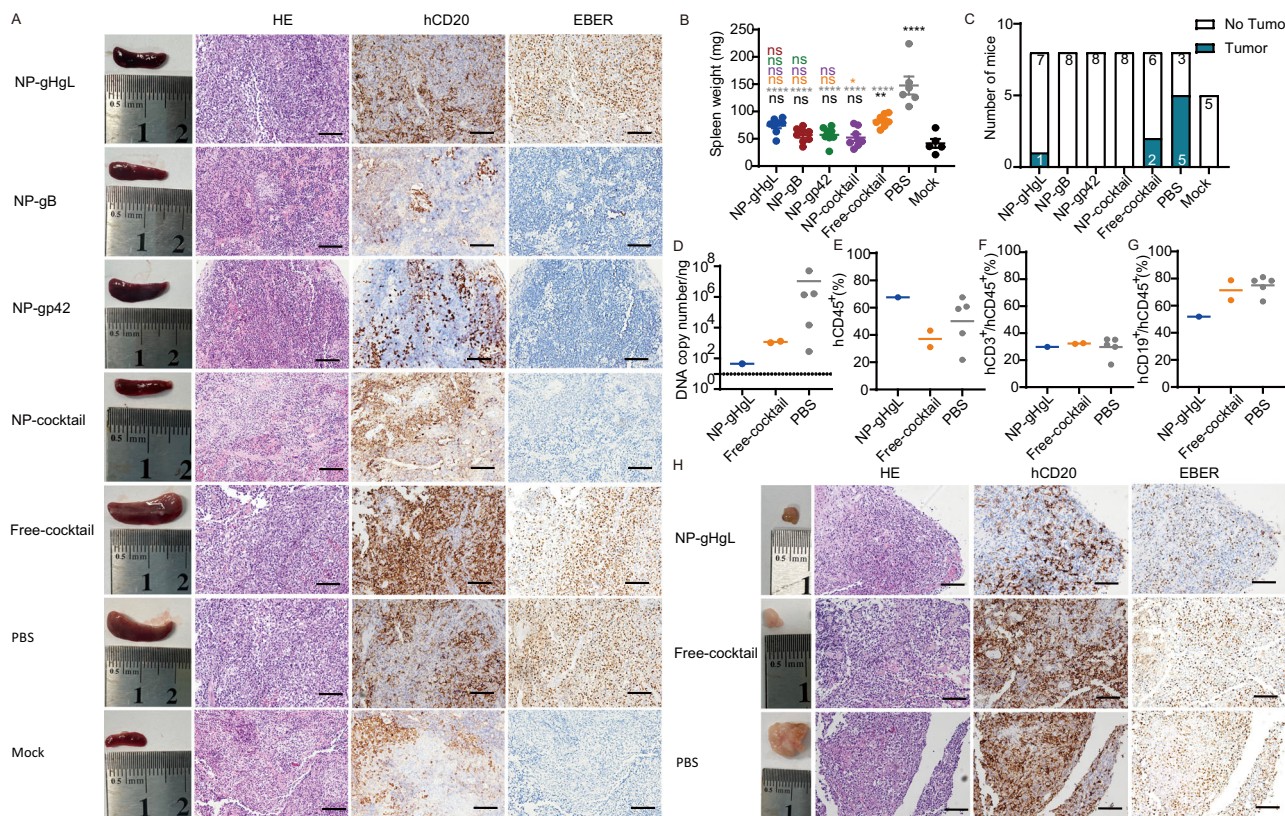

**Fig. 7 | Humanized mice injected with IgG induced by NP-cocktail were free of EBV-associated lymphoma. A** Representative macroscopic spleen and spleen sections stained by hematoxylin and eosin (HE), immunostained for hCD20[+] and in situ hybridized for EBV-encoded RNA (EBER). Each image is representative of the experimental group (Scale bar = 50 μm) (NP-gHgL, NP-gB, NP-gp42, NP-cocktail and Free-cocktail: $n = 8$; PBS, $n = 6$; Mock, $n = 5$). **B** Spleen weight of individual mice in each group (NP-gHgL, NP-gB, NP-gp42, NP-cocktail and Free-cocktail: $n = 8$; PBS, $n = 6$; Mock, $n = 5$) at the end of the experiments or earlier at the time of euthanasia (>20% body weight loss). Samples of two mice of PBS group died on day 7 and 10 were not collected because of decay. Data are shown as mean ± SEM. Statistical analysis was performed using one-way ANOVA with Turkey's multiple comparison test and the precise *P* values are shown in the source data file. The color of the asterisks or ns denotes the statistical difference. **C** 1/8 mice, 2/8 mice and 5/8 mice in NP-gHgL, Free-cocktail and PBS IgG-treated group, respectively, developed EBV-associated lymphoma, defined by histological observation. DNA copy numbers (**D**) and the percentages of hCD45[+] (**E**), hCD3[+] (**F**), hCD19[+] (**G**) cells are shown for each tumor. The horizontal line represents the mean of each group and dashed line indicates the detection limit. (NP-gHgL, $n = 1$; Free-cocktail, $n = 2$; PBS, $n = 5$). **H** Representative macroscopic tumor and tumor tissue stained (NP-gHgL, $n = 1$; Free-cocktail, $n = 2$; PBS, $n = 5$) by HE, hybridized for EBER, and immunostained for hCD20[+] at necropsy (Scale bar = 50 μm). Source data are provided as a Source Data file.

donors. The mice that developed EBV-associated lymphoma also had higher numbers of hCD45[+] cells, which may have resulted from the proliferation of infected B cells or from the proliferation of antigen-specific T cells (Fig. S36). We further determined the percentage of the activated (hCD69[+]hCD137[+]) hCD8[+] T cells in humanized mice spleen at the end of the experiment. Consistent with the previous studies, an increase of hCD69[+]hCD137[+] activated hCD8[+] T cell populations were observed in unprotected humanized mice, which still developed lymphoma after EBV challenge (Fig. S37)[15,50,52]. Thus, the previously existing EBV-specific T cells in PBMCs from EBV seropositive healthy donor may not effectively kill infected B cells under conditions where there is a relatively high EBV dose and a very low percentage of EBV-specific T cells[62–64]. However, the specific mechanisms of inefficient killing of EBV infected B cells by the activated T cells remain to be determined.

Overall, this analysis showed that IgG elicited by NP-cocktail conferred superior protection to humanized mice against viremia and the development of lymphoma following lethal EBV challenge. IgG elicited by other vaccine formulations showed various levels of protection but did not confer sterile protection in all treated mice.

## Durability of the protection induced by nanovaccines

Moreover, we further characterized the passive protection conferred by sera collected at week 5 of mice immunized with different vaccine formulations (Fig. S38). The body weight of humanized mice treated with sera from NP-cocktail immunized mice remained stable and all treated mice survived until the end of the experiment (Fig. S38B). Sera from NP-cocktail immunized mice protected all humanized mice from EBV-associated lymphoma (Fig. S38F, G), although one recipient mouse showed a late but detectable increase of EBV DNA in blood (Fig. S38D, E).

To address the long-term persistence of protective immune responses in immunized mice, we tested the passive protection conferred by the serum of mice collected at week 50 in a challenged experiment (Fig. S39). After challenge, the body weight of humanized mice pre-inoculated with sera from mice immunized with NP-cocktail remained stable and only one humanized mouse in this group died at week 8 post EBV challenge (Fig. S39B, C). In contrast, two out of five mice treated with sera from NP-gHgL, NP-gB, NP-gp42 and Free-cocktail immunized mice died after EBV challenge, respectively (Fig. S39C). An increase in EBV DNA copy numbers was detected in only one of the humanized mice treated with sera from NP-cocktail immunized mice and all these mice were free of lymphoma (Fig. S39D–F). In contrast, two mice treated with sera from NP-gHgL, NP-gB and Free-cocktail immunized mice, respectively and one mouse treated with sera from NP-gp42 developed EBV-associated lymphoma (Fig. S39F, G). Although the in vivo protection efficiency decreased from sera collected at week 5 to sera collected at week 50, NP-cocktail still retained high level of protection against lymphoma and effectively limited EBV DNA loads in PBMCs (Figs. S38 and S39).

## Discussion

An effective vaccine could reduce the heavy diseases burden caused by EBV infection. Here, we rationally designed nanovaccines consisting of the EBV core fusion machinery glycoproteins, gHgL, gB or gp42, and two adjuvants CpG and MPLA. These nanovaccines induced potent humoral and cellular immune responses without obvious side effects through promoted APCs internalization, target LN transportation and enhanced GC formation. Single nanovaccine immunization generated high levels of neutralizing antibodies mainly targeting the vulnerable sites of each antigen. However, NP-cocktail induced much more potent neutralizing and fusion-inhibitory antibodies through broadening the antigen spectrum. IgG generated by NP-cocktail conferred superior protection against viremia and EBV-associated lymphoma in humanized mice. Importantly, serum antibodies induced by NP-cocktail reduced the incidence of EBV-associated lymphoma for an extended period.

Soluble antigens adjuvanted with the appropriate molecular adjuvant could enhance the magnitude, breadth and durability of vaccine-induced immune responses. However, the easy clearance of soluble antigens and the potential systemic toxicity induced by the molecular adjuvants have limited their applications. Nanodelivery systems could not only improve the pharmacokinetics of antigens and adjuvants but also enhance the immunogenicity of antigens in vaccines against several pathogens, including enterovirus 71, hepatitis B virus and influenza virus[27,65,66]. We formulated the nanovaccines by using PLGA, DOTAP and DSPE-PEG2000 to co-encapsulate the antigens and adjuvants via electrostatic interaction and hydrophobic forces. The nanovaccines were hydrophilic, positively charged, round and rough-surfaced nanoparticles with a diameter less than 100 nm. The composition and structure of these nanovaccines suggest their good biocompatibility and release profile of antigens and adjuvants in vivo. Moreover, the shape, size, surface charge, particle geometry and hydrophobicity determine the biodistribution and circulation of nanovaccines[67]. We observed that EBV nanovaccines more effectively transported to LNs, activated BMDCs and enhanced GCs formation. Compared with the free-form vaccines and alum-adjuvanted vaccines, nanovaccines induced more robust adaptive immune responses. Taken together, we demonstrated that encapsulation of antigen and adjuvants in nanoparticles enhanced vaccine efficacy against EBV infection in animal models and provided a valuable strategy for the development of glycoprotein-based EBV vaccines.

Antigen selection is a crucial part for vaccine development. The EBV infection process involves multiple glycoproteins and the optimum selection or combination of antigens to develop vaccines remains to be determined. Previous studies demonstrated that the core entry glycoproteins, gHgL, gB and gHgLgp42 induced higher neutralizing antibody titers than gp350 alone[14,17]. A pentavalent virus-like particle-based vaccine containing EBV gp350, gB, gp42, gH and gL induced neutralizing antibodies protected both B and epithelial cells infection in vitro[68]. IgG induced by ferritin-based nanoparticles vaccines combining gH/gL/gp42+gp350D$_{123}$ or gH/gL+gp350D$_{123}$ protected humanized mice from lymphoma[19]. A recently developed mRNA vaccine containing four mRNAs encoding gH, gL, gp42 and gp220 is evaluated in clinical trials (mRNA-1189; NCT05164094). Previous studies showed that gp42 required co-expression of gHgL to bind to its receptor or recognized by the nAb F-2-1[69,70]. However, in this study, we found that serum antibodies from healthy EBV carriers efficiently bound to gp42 alone (Fig. 1A), and NP-gp42 immunization could elicit neutralizing antibodies to block EBV infection of B cells (Fig. 2D). Besides, we previously detected the binding of HLA-II to gp42 alone or gHgLgp42 by surface plasmon resonance assay and isolated several nAbs against gp42 from rabbits immunized with gp42 alone[52,55]. One of these nAbs, 5E3, was also able to bind gHgLgp42 complex[52]. Hence, the presence of partially exposed neutralizing epitopes on the gp42 indicates that gp42 alone is also a promising candidate for the development of prophylactic vaccines against EBV.

In the present study, individually, NP-gHgL, NP-gB and NP-gp42 all induced more potent and durable humoral and cellular immune responses than Free-form and Al-adjuvanted vaccines. Still, the in vitro neutralizing titers and fusion block ability of antibodies elicited by single nanovaccines were lower than the multi-target NP-cocktail nanovaccine. Indeed, IgG induced by NP-cocktail in rabbits provided superior protection to humanized mice against viremia and EBV-associated lymphoma. This efficacy was better than that provided by NP comprising individual antigens or by a cocktail of non-encapsulated antigens and adjuvants (Free-cocktail). NP-cocktail immunization generated antibodies targeting different steps of EBV infection. Hence, the combination of antigens and the nanoparticle-based formulation shows two advantages to be considerd in the development of effective anti-EBV vaccines. Combination of different antigens involved infection process is a trend for further improvement of EBV prophylactic vaccine design[10].

Neutralizing antibodies raised by vaccines are correlated to the efficiency of infection control[71,72]. Nanovaccines induced higher levels of neutralizing and fusion-inhibitory antibodies. The majority of the targeted epitopes overlap with those of reported neutralizing antibodies against each antigen. The identification of the neutralizing and non-neutralizing epitopes of EBV glycoproteins during natural infection and vaccine immunization will improve the rational design and development of EBV vaccines[73]. In addition to robust humoral responses, nanovaccines induced potent EBV-specific CD4$^+$ and CD8$^+$ T cell immune responses, which are also important for anti-viral protection[21,22]. Although multiple CD8$^+$ and CD4$^+$ T cell epitopes have been identified on gL, gH, gp42 and gB, EBV vaccine-induced glycoprotein-specific T cell immune responses were not thoroughly studied[74,75]. The nanovaccine-induced CD4$^+$ T cell response was Th1-biased as demonstrated by the pattern of cytokine produced and the subtypes of antibodies that were generated. However, whether nanovaccine-induced T cell immune responses could control EBV infection or reactivation remains to be determined.

It is reported that the delayed EBV infection in young adults often causes infectious mononucleosis, which also increases the risk of Hodgkin's lymphoma and multiple sclerosis[76,77]. Hence, the durability of the protective effect elicited by the vaccine is a very important question for the development of a prophylactic vaccine against EBV. Our data suggest that NP-cocktail vaccination could at least reduce the incidence of EBV-associated lymphoma for a long time. Evaluation of the durability of the protective immune responses is essential for the future development of vaccines against EBV.

Taken together, nanoparticle encapsulation and co-delivery with adjuvants significantly enhanced the immunogenicity of EBV gHgL, gB and gp42. Among the different vaccine formulations evaluated in this study, the NP-cocktail vaccine induced the most potent protective effect, which is highly promising for further evaluation in clinical trials. For future developments of EBV vaccines, this nanoparticle system co-delivering antigens and adjuvant provides a solid basis for vaccines to induce robust humoral and cellular immune responses. It is worth contemplating the introduction of additional EBV antigens in this delivery system to prevent EBV-associated diseases.

## Methods

### Animal studies and human specimens

All mice experiments were performed under protocols approved by the Sun Yat-sen University Cancer Center Animal Care and Use Committee. All rabbits experiment were performed under protocols approved by the Sun Yat-sen University Animal Care and Use Committee. According to approved guidelines, animals were humanely euthanized at the end of each experiment. Although we don't have sex bias, we have used female mice/rabbits in the study because male mice/rabbits tend to fight. All mice were housed 5 per cage in a controlled environment (temperatures of ~18-23 °C with 40–60%

humidity) in standard bedding with a standard 12-hour daylight cycle, cessation of light at 6 PM, and free access to standard chow diet and water. All rabbits were housed 1 per cage with a standard 12 h daylight cycle. Experiments were conducted during the light cycle, excluding continuous dietary interventions. The maximal tumor size/burden permitted is 20 mm in diameter or 2000 mm³ in volume of humanized mice, and the maximal tumor size/burden was not exceeded in the experiments.

Human PBMCs used for humanized mice reconstitution were collected from anonymous blood donors. This study was approved by the Institutional Ethics Committee of the Sun Yat-sen University Cancer Center, Guangdong, China.

Sera from healthy individuals were collected from Sun Yat-sen University Cancer Center and was approved by the Institutional Ethics Committee of the Sun Yat-sen University Cancer Center, Guangdong, China. Written informed consent was obtained from all participants. No self-selection bias was involved in this study. The only criteria used for recruitment of sera samples was EBV seropositivity. Sex and gender-based analysis is not relevant for the study.

## Plasmids

The coding sequences of the ectodomains of gH (residues 19 to 678), gL (residues 24 to 137), gp42 (residues 34 to 223), gp350 (residues 2-425) and gB (residues 22 to 672, with residues WY[112-113] and WLIW[193-196] replaced with HSV-1 residues HR[177-178] and RVEA[258-261]) were PCR-amplified from the EBV M81 strain (GenBank accession no. KF373730.1). The coding sequences of gL and gH were connected by a (GGGGS)$_3$ linker. The gHgL and gB ectodomain sequences were cloned into the pCDNA3.1(+) expression vector with a 6× His tag at the C-terminus (pCDNA3.1-gHgL and pCDNA.1-gB). The gp350 and gp42 ectodomain was cloned into the pVRC8400 expression vector with a 6× His tag at the C-terminus (pVRC8400-gp42).

pCAGGS expression plasmids for EBV gH, gL, gB, T7, and pT7EMCLuc (which carries a luciferase reporter gene under the control of the T7 promoter) were kindly provided by Professor Richard Longnecker[78]. Full-length gp42 was cloned into pCAGGS for the B cell fusion assay.

Full-length BALF5 was PCR-amplified from EBV M81 BAC and then inserted into the pUC19 vector, which was used to plot the standard curve for quantifying EBV DNA copy numbers[79].

## Cell lines

All cell lines were cultured at 37 °C in humidified air containing 5% $CO_2$. 293 T cells (IMMOCELL, Xiamen China) were cultured in Dulbecco's modified Eagle medium (DMEM; Invitrogen) with 10% fetal bovine serum (FBS; Invitrogen) and antibiotics (penicillin, 100 U/mL; streptomycin, 100 μg/mL; Invitrogen). 293 F cells were cultured in Union 293 medium (Union). DC2.4 cells, Akata cells and HNE1 cells[80] were cultured in RPMI 1640 (Invitrogen) with 10% FBS (Invitrogen), and antibiotics (penicillin, 100 U/mL; streptomycin, 100 μg/mL; Invitrogen). CNE2-EBV cells[81] and Akata-EBV cells[82], were propagated in RPMI 1640 (Invitrogen) with 10% FBS (Invitrogen), antibiotics (penicillin, 100 U/mL; streptomycin, 100 μg/mL; Invitrogen), and maintained under G418 selection (700 μg/mL; MP Biomedicals). CHO-K1 cells (ATCC) were cultured in F-12K Medium with 10% FBS (Invitrogen).

## Proteins and antibodies expression

293 F cells were transiently transfected with pCDNA3.1-gHgL, pCDNA3.1-gB, pVRC8400-gp350 or pVRC8400-gp42 using polyetherimide (PEI) and cultured for 5–7 days. After supernatant filtration through 0.22 μm filters, the 6-His-tagged gHgL, gB, gp350 and gp42 ectodomains were purified with Ni²⁺ Sepharose™ 6 Fast Flow beads (GE Healthcare). The targeted proteins were eluted with elution buffer (250 mM imidazole, 20 mM HEPES, 250 mM NaCl, pH 8.0), dialyzed overnight into PBS and concentrated using a 10 kDa ultrafiltration tube

(Millipore). The protein molecular weight and purity were verified by SDS-PAGE staining with Coomassie brilliant blue and Western blot.

To generate antibodies for competition ELISA, 293F cells were transfected with plasmids encoding heavy and light chains of antibody AMMO1[50], human-mice chimeric 6H2[51], human-rabbit chimeric 10E4[52], human-rabbit chimeric 3A3[15], human-rabbit chimeric 3A5[15], AMMO5[50,53], human-rabbit chimeric 1A7[55], human-rabbit chimeric 6G7[55] and human-rabbit chimeric 5E3[52]. The antibodies were purified with Protein A affinity chromatography (GE Healthcare).

## Antibody depletion from sera of healthy EBV carriers

Antibody depletion assay was performed as previously reported[15]. Briefly, 100 μl human serum samples were added to a tube containing pellet $3 \times 10^6$ 293 T cells that stably expressed gp350, gB, gp42 or gHgL. The mixture was incubated on ice for 1 h to deplete the gp350, gB, gp42 or gHgL-specific antibodies. The depletion step was repeated 15 times. The total anti-gp350, anti-gB, anti-gp42 and anti-gHgL IgG titers in each serum were determined by ELISA before and after depletion. The percentage of IgG titer reduction was calculated by the formula (1-IgG titer-depleted/IgG titer-before) × 100%. The reduction in the neutralizing titer of sera was calculated by the formula (1-ID50-depleted/ID50-before) × 100%.

## Preparation of nanovaccines and controls

The 1,2-dioleoyl-3-trimethylammonium propane (DOTAP; Sigma-Aldrich) was dissolved into ethanol (1 mg/mL). The poly (lactic-co-glycolic) acid (PLGA; Sigma-Aldrich) was dissolved into DMSO (5 mg/mL). The 1,2-distearoyl-sn-glycero-3-phosphoethanolamine-N-[(polyethylene glycol)-2000] (DSPE-PEG2000; Sigma-Aldrich) was dissolved into sterile water for injection (1 mg/mL). The adjuvant CpG (Invivogen) was dissolved in sterile water for injection (5 mg/mL). The adjuvant monophosphoryl lipid A (MPLA; Sigma-Aldrich) was dissolved into ethanol (2.5 mg/mL).

Purified gHgL, gB, gp42 or gp350 was mixed with CpG to produce mixture 1. MPLA was mixed with DOTAP to produce mixture 2. Mixture 1 was added dropwise into mixture 2 with continuous stirring to form a hydrophobic core, which was then added dropwise to 5 mg/mLPLGA with continuous stirring. Finally, the particles were added dropwise to a solution of 1 mg/mL DSPE-PEG2000 to form hydrophilic nanovaccines. The nanovaccines were dialyzed into PBS overnight at 4 °C. The volume ratio of mixture 1: mixture 2: PLGA: DSPE-PEG2000 was 1:18:18:120.

Free-form vaccines, Free-gHgL, Free-gB and Free-gp42, were prepared by mixing the corresponding antigens with CpG and MPLA. Alum-adjuvanted vaccines, Al-gHgL, Al-gB and Al-gp42 were prepared by mixing the corresponding antigens with Inject™ Alum adjuvant (Rockford). The cocktail nanovaccines was prepared by mixing NP-gHgL, NP-gB and NP-gp42 at equal ratios. The free-form cocktail vaccine was prepared by mixing gHgL, gB and gp42 with adjuvants (CpG and MPLA). The alum-adjuvanted cocktail vaccine was prepared by mixing gHgL, gB and gp42 with Inject™ Alum adjuvanted (Rockford). Amounts are indicated in Table S1.

## Characterization of nanovaccines

The size, polydispersity index (PDI) and zeta potential of nanovaccines were characterized using Zetasizer Nano ZS (Malvern) at 25 °C. The surface morphology of the nanovaccines was observed by transmission electron microscopy (TEM) using a Tecnai T12 instrument at 100 kV.

To evaluate the encapsulation efficiency of nanovaccines, gHgL, gB and gp42 were pre-labeled with Cy5-NHS (Lumiprobe). 50 μg Cy5-labled gHgL, gB or gp42 was mixed with 100 μg CpG adjuvant (Invivogen) to produce mixture 1. The adjuvant MPLA (100 μg) was mixed with 1 mg/mL DOTAP to produce mixture 2. Mixture 1 was added dropwise into mixture 2 with continuous stirring, which was then

added dropwise to 5 mg/mL PLGA with continuous stirring. Finally, the particles were added dropwise to a solution of 1 mg/mL DSPE-PEG2000 to form hydrophilic nanovaccines. The volume ratio of mixture 1: mixture 2: (PLGA): DSPE-PEG2000 was 1:18:18:120. The nanovaccines were ultracentrifuged at $40,000 \times g$ for 2 h to pellet the components encapsulated into the nanoparticles. The concentration of the unencapsulated components in the supernatant was measured.

Encapsulation efficiency of antigens: the fluorescence intensity of unencapsulated antigens was detected using a HITACHI F-7000 fluorescence spectrophotometer. A series of diluted Cy5-labled gHgL, gB or gp42 was used as standard curves. The encapsulation efficiency was calculated by the equation (1-amount of unencapsulated antigen/Total amount of antigen) × 100%.'

Encapsulation efficiency of CpG: the unencapsulated amount of CpG was detected by Quant-iT OliGreen™ ssDNA Assay Kit (Thermo Scientific) and measured with Spark 10 M multimode microplate reader (TECAN) according to the manufacturer's protocol. The encapsulation efficiency was calculated by the equation (1-amount of unencapsulated CpG/Total amount of CpG) × 100%.'

Encapsulation efficiency of MPLA: The amount of MPLA was detected using Tachypleus Amebocyte Lysate test kit (BIOENDO). The encapsulation efficiency was calculated by the equation (1-amount of unencapsulated MPLA/Total amount of MPLA) × 100%.

To determine release profiles, nanovaccines were mixed with PBS at 1:1 (v/v) and 0.1% w/v Tween 80 and incubated at 37 °C under continuous shaking at 100 rpm. At different times from 0 h to 48 h, samples were ultracentrifuged at $40,000 \times g$ for 2 h and the concentration of the released components in the supernatant was measured as indicated above.

For long-term storage stability, the nanovaccines were stored at 4 °C and their size was determined at different time points using a Zetasizer Nano ZS (Malvern) at 25 °C.

## Cell viability quantification
Murine dendritic DC2.4 cells were seeded into 96-well plates and incubated at 37 °C for 24 h with two-fold serial dilutions of each nanovaccines from 300 μg/mL. The Cell Counting Kit 8 (CCK-8; Dojindo) was used to determine cell viability according to the manufacturer's protocols. Absorbance was measured using a Bio-Tek EPOCH microplate reader.

## Lymph nodes (LNs) imaging assay
4-week-old female C57BL/6 J mice were subcutaneously injected at the root of tail with nanovaccines or corresponding free-form vaccines at an equivalence dose of 5 μg Cy5-labelled gHgL, gB or gp42 per mouse. Mice were euthanized at 6 h ($n = 3$) and 24 h ($n = 3$) post-injection and inguinal and axillary LNs were excised for imaging using the IVIS Spectrum In vivo Imaging System (PerkinElmer). Other organs were isolated 24 h post injection for imaging to evaluate the bio-distribution of vaccines.

To determine the distribution of vaccines in LNs, mice were injected with NP-gHgL, NP-gB, NP-gp42, Free-gHgL, Free-gB and Free-gp42 following the protocol above. Inguinal LNs were embedded with OCT cryostat sectioning medium (SAKURA Tissue-Tek) 6 h post injection and subsequently cryostat-sectioned at 8 μm thickness. LN sections were fixed with cold acetone for 15 min, washed with PBS three times and blocked with blocking buffer (1% BSA in PBS) at room temperature for 2 h. Sections were stained with Alexa Flour 594-B220 (Biolegend) and FITC-CD3 (Biolegend) at 4 °C overnight. The slides were washed by PBS three times and mounted with Prolong™ Glass antifade mountant (Invitrogen). The images were collected using an LSM 980 confocal microscope (Zeiss).

## BMDCs internalization and maturation evaluation
Bone marrow-derived DCs (BMDCs) from femurs and tibias of 4-week-old female C57BL/6 J mice ($n = 15$) were isolated and cultured as previously reported[83]. Briefly, isolated BMDCs were cultured in RPMI 1640 medium supplemented with 10% FBS, 1% penicillin/streptomycin, 20 ng/mL GM-CSF (Beyotime Biotechnology) and 10 ng/mL IL-4 (Beyotime Biotechnology). The medium was half replaced every 2 days and immature BMDCs were collected on day 6 for subsequent experiments.

To evaluate the internalization efficiency of different vaccines, $5 \times 10^5$ BMDCs were seeded in a 24-well plate and incubated with PBS, free-forms or nanovaccines containing Cy5-labelled gHgL, gB and gp42 (final concentrations: antigen: 4 μg/mL; CpG: 4 μg/mL; MPLA: 4 μg/mL) for 4 h, respectively. BMDCs were washed three times by PBS and the percentage of Cy5+ BMDCs was determined by CytoFLEX LX Flow Cytometer (Beckman Coulter). The data were analyzed with FlowJo software X 10.0.7 (Tree Star).

To evaluate the maturation of BMDCs induced by different vaccines, $1 \times 10^4$ BMDCs were seeded in a 24-well plate and incubated with PBS, empty nanovaccines (without adjuvant and antigen), free-forms or nanovaccines respectively for 24 h (final concentration: antigen: 4 μg/mL; CpG: 4 μg/mL; MPLA: 4 μg/mL). After incubation 24 h, cells were collected and blocked with anti-mouse CD16/32 for 20 min at 4 °C. After three washes by PBS, cells were incubated with anti-mouse CD11c-BV650 (Biolegend), anti-mouse CD86-PE/Cy7 (Biolegend), anti-mouse CD80-PE (Biolegend) and anti-mouse MHC-II-PerCP/Cy5.5 (Biolegend) antibodies for 30 min at 4 °C. Then, cells were washed with PBS three times and the population of CD80+CD86+/CD11c+ and CD86+MHC-II+/CD11c+ were determined by CytoFLEX LX Flow Cytometer (Beckman Coulter) and analyzed with FlowJo software X 10.0.7 (Tree Star).

## Immunization design
The doses of different vaccines used in C57BL/6 J mice and New Zealand white rabbits are summarized in Supplementary Table 1. Four to six-week old female C57BL/6 J mice were immunized via subcutaneous injection at the root of tail three times at week 0, 2 and 4. Six-month-old female rabbits were immunized through intramuscular injection three times at week 0, 2 and 4. Blood samples were collected regularly to evaluate binding titers and neutralizing titers. Spleens of C57BL/6 J mice were harvest at 5 weeks post-first vaccination to evaluate cellular immune responses. LNs of C57BL/6 J mice were harvested 5 weeks post-first vaccination to evaluate germinal center (GC) formation. The body weight and temperature of immunized rabbits were recorded every week post-vaccination. Mice or rabbits injected with PBS were used as negative controls. TC, ALT, TBIL, TG, AST, ALP and UREA level of sera collected at week 5 from immunized mice or at week 6 from immunized rabbits was analyzed by Auto Chemistry Analyzer (Mindray).

The detection of binding titers and neutralizing titers of sera from vaccines containing a single antigen and cocktail vaccines immunized mice or rabbits were performed with the same procedure, on the same day, under the same conditions and with the same reagents.

Each group contains five female C57BL/6 J mice or five female New Zealand white rabbits for the immunogenicity evaluation.

## Enzyme-linked immunosorbent assay (ELISA)
Sera samples were obtained from blood after centrifugation at $500 \times g$ for 10 min at 4 °C. ELISA plates (Corning) were pre-coated with 100 ng purified gp350, gHgL, gB or gp42 per well at 37 °C for 2 h. After washing once with PBST (PBS containing 0.05% v/v Tween 20), plates were blocked with blocking buffer (PBS containing 0.5% casein, 2% gelatin, and 0.1% ProClin 300, pH 7.4) at 4 °C overnight. Serial dilutions of sera collected from human sera, immunized mice, rabbits or humanized mice (starting from 1:100) or IgG (starting from 1 mg/mL) were added to the wells and incubated at 37 °C for 1 h. After five washes with PBST, HRP-conjugated goat anti-human IgG antibody (Promega), HRP-conjugated goat anti-mouse IgG antibody (Promega) or HRP-

conjugated goat anti-rabbit IgG antibody (Promega) was diluted at 1:5000 and incubated at 37 °C for 30 min. The EL-TMB kit (Sangon Biotech) was used for color visualization and absorbance was measured at 450 nm and 630 nm using a Bio-Tek EPOCH microplate reader.

To determine the IgA and IgG subtypes (IgG1/IgG2a/IgG2c) in mouse sera, HRP-conjugated goat anti-mouse IgA (BETHYL), IgG1 (BETHYL), IgG2a (BETHYL) and IgG2c (BETHYL) were added to the plate after sera incubation as described above.

To determine the reactivity of chimeric 6H2 (c6H2) antibody to the anti-mouse HRP secondary antibody, ELISA plates (Corning) were pre-coated with 100 ng purified gHgL per well at 37 °C for 2 h. After washing once with PBST, plates were blocked with blocking buffer at 4 °C overnight. Serial dilutions of c6H2 (starting from 1 μg/mL) were added to the wells and incubated at 37 °C for 1 h. After five washes with PBST, HRP-conjugated goat anti-human IgG antibody (Promega) or HRP-conjugated goat anti-mouse IgG antibody (Promega) was diluted at 1:5000 and incubated at 37 °C for 30 min. The EL-TMB kit (Sangon Biotech) was used for color visualization and absorbance was measured at 450 nm and 630 nm using a Bio-Tek EPOCH microplate reader.

### EBV production
CNE2-EBV cells and Akata-EBV cells carrying the Akata-EBV-GFP BAC were used to produce epithelial cells-derived EBV (CNE2-EBV-GFP) and B cells-derived EBV (Akata-EBV-GFP), respectively[84]. 20 ng/mL 12-O-tetradecanoylphorbol 13-acetate (TPA; Beyotime) and 2.5 mM sodium butyrate (NaB; Sigma Aldrich) were added to induce CNE2-EBV cells for 12 h and then medium was replaced by fresh complete medium. Goat anti-human IgG (Tianfun Xinqu Zhenglong Biochem. Lab) was added at a final concentration of 100 μg/mL to induce Akata-EBV cells at the density of $2 \times 10^6$ cells/mL for 6 h and then medium was replaced by the fresh complete medium. After culturing for 72 h, cell supernatant was collected, centrifuged and filtrated through a 0.45 μm filter. Finally, viruses were concentrated 100-fold by centrifugation at $50,000 \times g$ for 2.5 h and resuspended in serum-free RPMI 1640. The viruses were stored at −80 °C.

### EBV titration and GRU determination
To determine the infectivity of CNE2-EBV-GFP, Akata cells were seeded in at the density of $1 \times 10^4$ cells in 180 μl RPMI 1640 with 10% FBS and incubated with 20 μl of 2-fold serially diluted CNE2-EBV-GFP for 48 h. To determine the infectivity of Akata-EBV-GFP, HNE1 cells were seeded in 96-well plates at the density of $0.5 \times 10^4$ cells per well and incubated with 50 μL of 2-fold serially diluted Akata-EBV-GFP at 37 °C for 3 h. Medium was changed to 200 μl RPMI 1640 with 10% FBS and the cells were cultured for 48 h. After 48 h incubation, GFP-positive infected cells were counted by a CytoFLEX LX (Beckman Coulter) and analyzed with FlowJo software X 10.0.7 (Tree Star).

The titer of CNE2-EBV-GFP and Akata-EBV-GFP used in this study is shown in Fig. S40. The same batch of the CNE2-EBV-GFP was used in all B cell neutralizing assays and the same batch of the Akata-EBV-GFP was used in all epithelial cell neutralizing assays.

To calculate the green Raji units (GRU) of EBV, 100 μl of CNE2-EBV-GFP was added to $1 \times 10^5$ Raji cells. After 48 h incubation, infected cells were counted by a CytoFLEX LX (Beckman Coulter) and analyzed with FlowJo software X 10.0.7 (Tree Star). The GRU of the virus was calculated according to previous studies by the following formula: 'Total number of Raji cells × percentage of GFP⁺ cells/volume of the virus stock used'[79,85].

### Neutralization assay
All sera used in this assay were treated at 56 °C for 45 min to inactivate complement, and no extra complement was added. For B cell neutralization, 20 μL of serially diluted sera (starting at 1:10) or IgG

(starting at 10 mg/mL) were incubated with 20 μL CNE2-EBV-GFP (30% infectivity, Fig. S40B) for 2 h. Then, the mixture was added to $1 \times 10^4$ EBV-negative Akata cells and incubated at 37 °C for 48 h. For epithelial cell neutralization, 50 μL serially diluted sera (starting at 1:10) or IgG (starting at 10 mg/mL) were incubated with 50 μL Akata-EBV-GFP (30% infectivity; Fig. S40A) at 37 °C for 2 h. The mixture was added to $0.5 \times 10^4$ HNE1 epithelial cells and medium was changed after 3 h. Untreated cells served as negative controls and cells incubated with virus and sera from mice mock-immunized by PBS were used as positive controls. After 48 h incubation, infected cells were counted by a CytoFLEX LX (Beckman Coulter) and analyzed with FlowJo softwareX10.0.7 (Tree Star).

The neutralizing efficiency of each sample was calculated as (1- % GFP positive cells of the sample with sera/%GFP positive cells of positive control) × 100%. Half maximal inhibitory dilution fold (ID50) and the half-maximal inhibitory concentration (IC50) were calculated by sigmoid trend fitting with GraphPad Prism 8.0. All neutralizing assays of human, mouse and rabbit sera were performed in the same way.

### Follicular helper CD4⁺ T (Tfh) cell detection
C57BL/6 J mice were immunized with different vaccine formulations three times at week 0, 2 and 4. At week 5, spleens from different groups were ground with a sterile syringe and the cell suspension was filtered through a 70 μm cell strainer (BD) and treated with red blood cell lysis buffer (BioLegend). Then, cells were centrifuged at $400 \times g$, washed twice with PBS, resuspended in PBS and stained with antibodies including anti-mouse CD45-APC/Cy7 (BioLegend), anti-mouse CD3-FITC (BioLegend), anti-mouse CD4-AF700 (BioLegend), anti-mouse CXCR5-BV421 (BioLegend) at 4 °C for 30 min. After washing, cells were fixed and permeabilized according to the eBioscience™ Intracellular Fixation & Permeabilization Buffer Set (Invitrogen) and stained with anti-mouse bcl6-PerCP/Cyanine5.5 antibody (BioLegend). The assays were performed with a CytoFLEX (Beckman Coulter), and the population of Tfh cells (CXCR5⁺Bcl6⁺) was analyzed using FlowJo software X 10.0.7 (Tree Star).

### Germinal center detection
C57BL/6 J mice were immunized via subcutaneous injection at the root of tail with different vaccine formulations (NP-, Free- and Al-gHgL, NP-, Free- and Al-gB, NP-, Free- and Al-gp42, NP-, Free- and Al-cocktail) three times at week 0, 2 and 4, and were sacrificed on week 5. C57BL/6 J mice mock-immunized with PBS were used as negative controls. The inguinal lymph nodes from both sides were harvested to be embedded with optimal cutting temperature (OCT) compound. Germinal center formation in inguinal lymph nodes was detected through the cryostat sectioning procedure described above and sections were stained with anti-mouse Alexa Fluor 594-IgD (Biolegend), Alexa Fluor 488-B220 (Biolegend) and Alexa Fluor 647-GL7 (Biolegend).

### Intracellular cytokine staining assay
C57BL/6 J mice were immunized with different vaccine formulations three times at week 0, 2 and 4. At week 5, spleens from different groups were ground with a sterile syringe and the cell suspension was filtered through a 70 μm cell strainer (BD) and treated with red blood cell lysis buffer (BioLegend). After washing twice with medium, cells were seeded in round bottom 96-well plates at a density of $5 \times 10^6$ cells/well. After restimulation with specific antigens (2 μg gHgL, 2 μg gB or 2 μg gp42/ well respectively) at 37 °C overnight, 5 μg/mL brefeldin A (BioLegend) and 2 μM monensin (BioLegend) were added to block intracellular cytokine secretion and incubated for 4 h. Then, cells were washed with PBS, blocked by Fc-blocking solution (5 μg/mL of CD16/CD32 mAb, eBioscience) and stained with antibodies including anti-mouse CD45-APC/Cy7 (BioLegend), anti-mouse CD3-FITC (BioLegend), anti-mouse CD4-AF700 (BioLegend) and anti-mouse CD8-APC (BioLegend) at 4 °C for 30 min. After washing, cells were fixed and permeabilized according

to the eBioscience™ Intracellular Fixation & Permeabilization Buffer Set (Invitrogen) and stained with anti-mouse IFNγ-PE/Cy7 antibody (BioLegend), anti-mouse TNFα-BV421 antibody (BioLegend) and anti-mouse IL2-PE antibody (BioLegend) at room temperature for 30 min. Cells with no stimulation were used as a negative control, while cells stimulated with phorbol myristate acetate (PMA)-ionomycin (Sigma) were used as a positive control. The population of antigen-specific T cells was measured by a CytoFLEX LX (Beckman Coulter) and the data was analyzed using FlowJo software X 10.0.7 (Tree Star).

## Memory B cells and memory T cells detection

C57BL/6 J mice were immunized with different vaccine formulations (NP-, Free- and Al-gHgL, NP-, Free- and Al-gB, NP-, Free- and Al-gp42, NP-, Free- and Al-cocktail) three times at week 0, 2 and 4. Splenocytes harvested at week 5 (day 35) and blood samples collected at week 12 from immunized C57BL/6 J mice were treated with red blood cell lysis buffer (BioLegend) at room temperature for 10 min. Then, cells were centrifuged at $400 \times g$, washed twice with PBS, resuspended in PBS and stained with antibodies including anti-mouse CD45-APC/Cy7 (BioLegend), anti-mouse B220-BV605 (BioLegend), anti-mouse CD3-FITC (BioLegend), anti-mouse CD8-APC (BioLegend), anti-mouse CD4-AF700 (BioLegend), anti-mouse IgG-PerCP/Cy5.5 (BioLegend), anti-mouse CD27-BV421 (BioLegend), anti-mouse CD44-PE/Cy7 (BioLegend) and anti-mouse CD62L-PE (BioLegend)for 30 min at 4 °C. The assays were performed with a CytoFLEX (Beckman Coulter), and the population of memory B cells ($IgG^+CD27^+$) and effector-memory T cells ($CD44^{hi}CD62L^{low}$) were analyzed using FlowJo software X 10.0.7 (Tree Star).

## Virus-free cell-cell fusion assay

The fusion blocking ability of sera was determined as previously described[14]. Briefly, for epithelial cell fusion model, CHO-K1 cells transfected with 2.5 µg each of pCAGGS-gB, pCAGGS-gH, pCAGGS-gL, pCAGGS-gp42 and pT7EMCLuc (express the luciferase reporter gene under the control of the T7 promoter) served as effector cells. The recipient epithelial cells were 293 T cells transfected with 10 µg of pCAGGS-T7 polymerase. For B cell fusion model, CHO-K1 cells transfected with 2.5 µg each of pCAGGS-gB, pCAGGS-gH, pCAGGS-gL, pCAGGS-gp42 and pT7EMCLuc served as effector cells. Daudi-T7 B cells stably expressed T7 RNA polymerase were the recipient B cells.

24 h-post transfection, $2 \times 10^5$ effector cells were trypsinized and incubated with the mouse sera (1:40 diluted) at 37 °C for 30 min prior to co-culture with $2 \times 10^5$ recipient cells in a 48-well plate at 37 °C for 24 h. The cocultured cells were lysed and luciferase activity was quantified using a Dual-Glo luciferase assay following the manufacturer's instructions (Promega).

## Competition ELISA

ELISA plates (Corning) were coated with 100 ng/well gHgL, gB or gp42 and blocked as described above. The dilution folds of mouse sera to produce 1 OD reading value were determined by serial a dilution in standard ELISA. To detect epitope competition, 100 µL of nAbs AMMO1, 6H2, 10E4, 3A3, 3A5, AMMO5, 1A7, 6G7 or 5E3 (10 µg/well) were first added to the antigens and plates were incubated at 37 °C for 30 min. After five washes with PBST, the predetermined dilution of sera (corresponding to 1 OD) was added for another 30 min. After five washes, HRP-conjugated goat anti-mouse IgG antibody (Promega) was added and incubated at 37 °C for 30 min. The blocking ratio of the nAb against each sera sample was calculated as (1- OD value of the sera binding to coated antigens treated with nAb/OD value of the sera binding to coated antigens without nAb treatment) ×100%.

## Total IgG purification from rabbit sera

New Zealand white rabbits were immunized with different vaccine formulations intramuscularly on day 0, 14 and 28 with different

vaccine formulations described above. On day 42, three immunized rabbits from each group were euthanized and the blood of rabbits from each group were pooled. Sera were obtained from blood after centrifugation at $500 \times g$ for 10 min at 4 °C, and total IgG were purified by Protein G affinity chromatography (GE Healthcare) and dialyzed into PBS. The purified total IgG were concentrated and verified by SDS-PAGE.

## Establishment of the humanized mouse model

Female 4-week-old SCID-beige mice were irradiated (1 Gy) with a RS2000 irradiator (RAD Source) and intravenously injected with Clodronate Liposomes (YEASEN; 500 µg/mouse). 24 h later, each mouse was engrafted with $1.5 \times 10^7$ human peripheral blood mononuclear cells (PBMCs) from healthy donors (Table S4). Blood samples were collected 4 weeks post engraftment and treated with red blood cell lysis buffer (BioLegend). Cells were stained with antibodies including anti-human CD45-APC/Cy7 (BioLegend), anti-human CD19-APC (BioLegend), anti-human CD3-FITC (BioLegend) and anti-mouse CD45-BV510 (BioLegend) at 4 °C for 30 min. The percentages of human $CD45^+$ cells, B cells and T cells were determined by flow cytometry with a CytoFLEX LX instrument (Beckman Coulter) and analyzed using the FlowJo software X 10.0.7 (Tree Star).

To detect the EBV DNA copy numbers in PBMCs used for the humanized mouse reconstitution, total DNA was extracted from $1 \times 10^6$ PBMCs from each donor using a DNA extraction kit (Omega). EBV copy numbers were quantified by real-time polymerase chain reaction (RT-PCR; SYBR green) with the following primers (F: 5′-GGTCA-CAATCTCCACGCTGA-3′; R: 5′-CAACGAGGCTGACCTGATCC-3′) to amplify a fragment of EBV BALF5 gene[79].

To analyze the EBV infection susceptibility of PBMCs in vitro, $1 \times 10^5$ PBMCs from each donor were seeded in 96-well plates. 100 µl serially diluted CNE2-EBV-GFP (started from 1×) were added into each well and incubated for 48 h. Akata B cells (EBV negative) incubated with EBV were used as the positive control. PBMCs without incubation with EBV were used as the negative control. After 48 h incubation, PBMCs were washed with PBS and stained by anti-human CD19-APC (BioLegend). GFP positive $CD19^+$ B cells represented the EBV infected B cells in PBMCs in vitro Gating strategy was shown in Fig. S30A. GFP-positive infected B cells were counted by a CytoFLEX LX (Beckman Coulter) and analyzed with FlowJo software X 10.0.7 (Tree Star).

To analyze the T cell activities in PBMCs during EBV infection of B cells in vitro, $1 \times 10^5$ PBMCs from each donor were seeded in 96-well plates with/without 0.4 µg/ml Cyclosporine A (CsA). 100 µl serially diluted CNE2-EBV-GFP (1×) were added into each well and incubated for 48 h. The percentage of GFP positive $CD19^+$ B cells was analyzed as described above.

## Sera and IgG passive administration and EBV infection

4 weeks post-grafting, humanized mice were injected intraperitoneally (i.p.) with 1 mg purified total IgG from sera of rabbits immunized with different vaccine formulations. Humanized mice injected with IgG purified from rabbits mock-immunized with PBS were used as controls. 24 h later, mice were challenged with a dose of 25,000 GRUs of Akata strain EBV via i.p. injection. Unchallenged mice formed the mock group. During the following weeks, peripheral blood samples were collected and body weights were tracked weekly. Mice were humanely euthanized and organs were harvested at the 8 weeks post-challenge or earlier if mice developed severe clinical symptoms of illness (e.g. >20% body weight loss).

To evaluate the passive protection efficiency of sera from immunized mice, 100 µl of sera were collected at week 5 or week 50. For each group of mice immunized with NP-gHgL, NP-gB, NP-gp42, NP-cocktail, Free cocktail and PBS, sera from 5 mice were pooled. Then humanized mice were injected intraperitoneally (i.p.) with 100 µl of pooled sera and were challenged 24 h later with a dose of 25,000 GRUs of Akata

strain EBV via i.p. injection. Unchallenged mice formed the mock group.

## Detection of human cells in peripheral blood, peritoneal lavage fluid, spleens, LNs and tumors

The spleens, LNs and tumors from different groups at week 8 or earlier (mice developed severe clinical symptoms of illness) were ground with a sterile syringe. The cell suspension was filtered through a 70 μm cell strainer (BD) and treated with red blood cell lysis buffer (BioLegend). Cells in peritoneal lavage fluid was collected by centrifuge at $400 \times g$. Peripheral blood samples collected at week 8 were also treated with red blood cell lysis buffer (BioLegend). Then, the cells were centrifuged at $400 \times g$, washed twice with PBS, resuspended in PBS, and stained with antibodies including anti-human CD45-APC/Cy7 (BioLegend), anti-human CD19-APC (BioLegend), anti-human CD3-FITC (BioLegend) and anti-mouse CD45-BV510 (BioLegend) at 4 °C for 30 min. The percentages of human $CD45^+$ cells, B cells and T cells were determined by flow cytometry with a CytoFLEX LX instrument (Beckman Coulter) and analyzed using the FlowJo software X 10.0.7 (Tree Star).

To detect the activated T cells in spleen, the single cell suspension after treated with red blood cell lysis buffer was centrifuged at $400 \times g$, washed twice with PBS, resuspended in PBS, and stained with antibodies including anti-human CD45-APC/Cy7 (BioLegend), anti-mouse CD45-BV510 (BioLegend), anti-human CD3-FITC (BioLegend), anti-human CD8-PC5.5 (BioLegend), anti-human CD4-Pacific Blue (BioLegend), anti-human CD69-PC7 (BioLegend) and anti-human CD137-APC (BioLegend) at 4 °C for 30 min. The percentages of activated ($hCD69^+hCD137^+$) $hCD8^+$ T cells were determined by flow cytometry with a CytoFLEX LX instrument (Beckman Coulter) and analyzed using the FlowJo software X 10.0.7 (Tree Star).

## Detection of EBV DNA in blood and tissues

DNA was extracted from peripheral blood (25 μL), lymph nodes, spleens, kidneys and tumors of mice using a DNA extraction kit (Omega) according to the manufacturer's instructions. EBV copy numbers were quantified by real-time polymerase chain reaction (RT-PCR; SYBR green) with the following primers (F: 5'-GGTCA-CAATCTCCACGCTGA-3'; R: 5'-CAACGAGGCTGACCTGATCC-3') to amplify a fragment of EBV BALF5 gene[79].

## Histological analysis

Tissue samples were fixed in 4% paraformaldehyde, embedded in paraffin and sectioned prior to hematoxylin and eosin staining (H&E): The hearts, livers, spleens, lungs and kidneys of immunized C57BL/6 J mice and rabbit were harvested one week and two weeks after their final immunization respectively. The hearts, livers, spleens, lungs, kidneys and tumors of humanized mice were collected at the week 8 or earlier (mice developed severe clinical symptoms of illness). The stained sections were visualized using an ECLIPSE Ni-U microscopy (NIKON).

## Human B cells staining and EBERs in situ hybridization

The spleens and tumors of humanized mice from different groups were immunostained with an anti-human CD20 antibody (Abcam) at a 1:200 dilution. EBV-encoded small RNAs (EBERs) were detected in spleens and tumors of humanized mice from different groups by in situ hybridizations with an EBER detection kit (ZSGB-BIO) according to the manufacturer's instructions.

## Statistical analysis

All statistical analysis was conducted with GraphPad Prism version 8. Statistical tests are indicated in figure legends.

## Reporting summary

Further information on research design is available in the Nature Portfolio Reporting Summary linked to this article.

## Data availability

All data supporting the findings of this study are provided in the Supplementary Information/Source Data file. Source data are provided with this paper.

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

## Acknowledgements

This work was supported by the grant from the National Key Research and Development Program of China Grant (2022YFC2305400 to M.X.); National Natural Science Foundation of China Grant (82371832 to Xiao.Zhang., 82122050 to M.X., 82341037 to Yong.Chen. and 22075324 to L.L); Natural Science Foundation of Fujian Province (2023J011235 to Xiao.Zhang.); Chongqing Education Commission of Science and Technology Research Project (KJQN202300453 to Xiao.Zhang.); Natural Science Foundation of Chongqing City (2023NSCQ-MSX1536 to Xiao. Zhang).

## Author contributions

L.L., Y.Z., Yong.Chen., M.X. and Xiao.Zhang. designed the study. L.Z., W.Z., H.L., Xin.Zhang., Z.Y., Z.W., L.C., H.C., Y.L., Yan.Chen. and Q.F. performed the experiments. L.Z., W.Z., H.L., Xiao.Zhang. and C.K. analyzed data. L.Z., W.Z., H.L., C.K., M.X. and Xiao.Zhang. wrote the manuscript. L.Z., M.Z., Q.Z., M.X. and Xiao. Zhang. participated in the discussion and interpretation of the results. All authors read and approved the final manuscript.

## Competing interests

The authors declare no competing interests.
