## [Peer Review File · Nature Communications]

REVIEWER COMMENTS

Reviewer #1 (Remarks to the Author):

Manuscript Nr: NCOMMS-23-44668-T

Zhong et al., "A cocktail nanovaccine targeting key entry glycoproteins elicits high-quality neutralizing antibodies against EBV infection"

The authors first correlated Epstein Barr virus (EBV) glycoprotein (gp350, gH, gL, gB and gp42) specific antibodies of healthy virus carriers with EBV neutralizing activity for B and epithelial cell infection. They then constructed nanovaccines containing recombinant gHgL, gB or gp42 proteins plus TLR9 or TLR4 agonists as adjuvants. According to their involvement in B and epithelial cell infection gHgL containing nanoparticles induced the highest neutralizing titers against B and epithelial cell infection and gp42 containing nanoparticles induced neutralizing activity for B cell infection. The nanoparticles were also able to mature dendritic cells and drain to lymph nodes for prolonged antigen retention. The authors then combined their three nanoparticle vaccines to elicit even higher EBV neutralizing antibody titers. With defined monoclonal antibodies against gHgL, gB and gp42 of highly EBV neutralizing ability it is then shown that many of the specificities that are elicited with the cocktail of nanoparticle vaccines overlap with the respective neutralizing epitopes. Immunizations were then repeated in rabbits with similar vaccination efficacy and antigen hierarchy. Rabbit IgG was then used to protect SCID-beige mice with human peripheral blood mononuclear cell (PBMC) transfer from EBV infection. Indeed, protection from EBV infection induced weight loss, viremia and lymphoma formation could be achieved. From these data the authors conclude that a mixed nanoparticle vaccine could elicit sterilizing immunity to prevent EBV associated diseases.

The reported study is interesting but similar to several other studies that have addressed combinations of EBV glycoproteins in eliciting neutralizing antibodies that protect humanized mice. Therefore, the novelty of the reported strategy remains unclear and the ambition to induce sterilizing immunity is questionable.

Major comments:

1. The authors report T cell responses that are induced by their vaccine formulations but the protective capacity of these remains unclear. For the induced neutralizing antibodies follicular helper CD4+ T cell induction might be more important than Th1 and CD8+ T cell responses and it would be informative to assess the capacity of the nanoparticle vaccines to induce these.

2. For their protection studies the authors used PBMC transfer models into SCID-beige mice that are then infected with 2.5×10^4 infectious EBV particles. They transfer then 1mg of rabbit IgG of their different vaccination groups. Why rabbit IgG and not IgG from the immunized mice? Would the IgG equivalent of one vaccinated mouse protect one SCID-beige-PBMC mouse from EBV infection? In previous studies sometimes a multitude of vaccinated needed to be used as antibody donors to protect recipients from EBV challenge.

3. The aim of this study seems to be sterile protection from EBV infection which the authors seem to be able to achieve with the nanoparticle cocktail vaccination (gHgL, gB and gp42). It would be interesting to determine or at least discuss the durability of such a sterilizing immune response. Since EBV infection causes infectious mononucleosis and its sequelae (increased risk for Hodgkin's lymphoma and multiple sclerosis) upon delayed primary infection in young adults, waning sterilizing immunity would delay primary infection in vaccinees, possibly with an increased risk for infectious mononucleosis.

Minor comments:

1. Interferon should be consistently abbreviated as IFN and not INF as in line 232.

Reviewer #3 (Remarks to the Author):

This manuscript designed a nano-vaccine cocktail composed of EBV core glycoprotein antigens gHgL, gB, or gp42, along with two adjuvants, CpG and MPLA. The authors elucidated that these nanovaccines can induce effective and long-lasting humoral and cellular immune responses by promoting APC internalization, targeting LN transportation, and enhancing GC formation, without any noticeable side effects. The nano-vaccines induced effective neutralizing and fusion-inhibitory antibodies by broadening the antigen spectrum. The IgG produced by these nano-vaccine cocktails provided 100% protection against viremia and EBV-associated lymphomas in humanized mice. This work is comprehensive, though there are a number of concerns that need to be addressed.

1. The correlation of EBV infection with cancer development vs. anti-tumor immunity (e.g., Nature 2021; 590(7844):157-162; Clin Cancer Res 2022; 28(20):4363-4369) is an interesting/debatable topic, which may need to be discussed in the manuscript.

2. In line 109 of the manuscript, it is mentioned that "All three nanovaccines with single antigens (gHgL, gB or gp42) induced potent and durable adaptive immune responses in mice attributed to efficient transportation to LNs, enhanced antigen internalization and maturation of dendritic cells,

and robust germinal center (GC) responses". However, the authors found that "IgG induced by a cocktail vaccine containing three nanovaccines conferred 100% protection from lethal EBV challenge in humanized mice, whereas IgG elicited by individual NP-gHgL, NP-gB, and NP-gp42 did not". What is the synergistic mechanism of these three nanovaccines? The innovation of the cocktail vaccine strategy may also be further discussed. It is not clear either why the authors used a mixture of three NPs each containing one antigen instead of one NP containing three antigens.

3. The authors may explain why the ratio 1:1:1 of the cocktail nano-vaccines was used.

4. The manuscript mentioned "high-quality neutralizing antibodies". A more precise definition for "high-quality" is needed.

5. What is the rationale of co-delivery of CpG and MPLA as adjuvants? The author found that the nano-vaccine based on CpG and MPLA is superior to the Alum-vaccine; what is the reason behind that? Do CpG and MPLA have any unique synergistic effects with gHgL, gB, or gp420-related immune response?

6. DOTAP is a cationic lipid and could also activate innate immune responses, beyond its role for complexing with proteins and CpG. This needs to be explored.

7. In Figures 1C and 1J, the results of 'r' value is 0.2997 and 0.2227 respectively. Could this reflect the importance of gHgL and gp42?

8. Figure 1O indicates that the loading efficiency of NP exceeded 90%, and even the loading of the adjuvant reached 99.9%. The authors should provide detailed description of preparation, purification, detection/analysis (for antigens and adjuvants), and calculation.

9. The differences in immunofluorescence staining in Figure 3E and F are not very clear.

10. In Figure 5, the immunization method for rabbits is "intramuscularly", while in Figure 2, the immunization method for C57BL/6J mice is "subcutaneously". Why were the same methods not adopted? When comparing intramuscular immunization in the rabbit model to subcutaneous immunization in the mice model, is there a difference in the biodistribution of the nano-vaccine?

11. Regarding the safety evaluation of the nano-vaccine, Figure S10 and Figure S16 only demonstrated that “tissue damage or inflammation was observed in the heart, liver, spleen, lung, and kidney tissues”, lacking the evaluation of biochemical functional indicators.

We commend the authors for their thorough and detailed work on their manuscript “A cocktail nanovaccine targeting key entry glycoproteins elicits high-quality neutralizing antibodies against EBV infection.” In this work, the authors developed a nanoparticle-based vaccine approach against Epstein-Barr virus (EBV) infection targeting multiple EBV entry glycoproteins, namely gHgL, gB and gp42. First, the authors analyzed the neutralizing capability of serum from individuals naturally infected with EBV, and performed correlative studies to determine which EBV antigens contributed to the neutralizing activity observed. Next, the authors developed individual nanoparticles incorporating either gHgL, gB or gp42, together with two adjuvants, CpG and MPLA. The nanoparticles were first characterized *in vitro* for their biochemical properties and composition, and were then tested for immunogenicity in C57BL/6J mice as compared to free soluble glycoproteins. Serum antibody responses, serum neutralizing capability, specific T cell and B cell responses, and biodistribution were all characterized. Then, the authors prepared a combination cocktail formulation that incorporated all three individual nanoparticles, and tested immunogenicity of this new formulation in mice as before, followed by immunogenicity assessment in rabbits. Using serum from immunized rabbits, the authors then finally tested vaccine efficacy against EBV infection in humanized mice in passive immunization experiments followed by viral challenge. While the results of these studies are encouraging and support inclusion of multiple glycoproteins in EBV vaccine design, we believe that the manuscript in its current condition is not ready for publication. Overall, this reviewer found that some key experimental details were either missing or hard to find. The authors also make several “grandiose” claims with minimal evidence to support them, and there are a few important concerns regarding the humanized mouse model used for efficacy studies. Based on this assessment, I recommend that the authors majorly revise their submission before publication on Nature Communications. Below find a point-by-point summary of our review for the authors to address:

Major comments:

- For appropriate rigor in *in vitro* neutralization experiments, it is imperative that the authors provide infectivity information for the EBV batches used in both HNE1 and Akata cell neutralizations: how infectious are 20 μ l of CNE2-EBV-GFP in Akata cells, and how infectious are 50 μ l of Akata-EBV-GFP in HNE1 cells? The authors should at the very least provide the level of infection (%) achieved in the absence of antibodies/serum for every neutralization experiment, but ideally should provide full infectivity titration curves for each batch of virus. Without this information, it is not possible to fully evaluate the impact of the obtained results; i.e. neutralizing 2% infectivity is not the same as neutralizing 20% infectivity. Similarly, the authors should provide methods and data as to how they titrated the virus used for *in vivo* humanized mice experiments.
- Although the authors present the argument that gp350 has not proved to be sufficient as a lone antigen in EBV vaccine design, they did not include a gp350 control group in their immunogenicity experiments, either in nanoparticle form or in free form, despite showing that gp350-specific serum antibodies contribute the most to B cell neutralization in Figure 1. This is a missed opportunity that dampens the message of the manuscript, and should be addressed.
- The results of the present study have shown that gp42 is capable of generating neutralizing antibodies without being found in complex with gHgL, which is very exciting, in contrast to previous evidence showing that the gHgL/gp42 conformational epitopes are critical for viral entry and generation of neutralizing antibodies. Can the authors expand/comment on this in the Discussion?
- No long-term neutralizing activity was tested for any of the immunogenicity studies, and no neutralization was performed to compare titers before and after all vaccine boosters to determine whether all three vaccinations are required, which was a missed opportunity.
- A sample size of n=3 animals per group in the rabbit studies does not allow for powered statistical comparisons.
- When combining individual nanoparticles for the cocktail formulation, how does this affect the final concentration of CpG and MPLA in the vaccine? If we are going by Supplementary table 1, where 5 μ g

individual nanoparticles are listed to contain 10µg of CpG/MPLA, then the cocktail formulation that contains 5µg of each individual nanoparticle should contain 30µg of CpG/MPLA, not 10µg as listed. If this is the case, is the adjuvant content in the free-cocktail also 30µg CpG/MPLA, or is it 10µg? Please correct/clarify as necessary.

- Please provide supporting citations as appropriate for the following:
 - Line 170 (PDF) indicates “the nanovaccines showed long-term stability.” Are 30-40 days considered long-term at 4 degrees for this type of nanoparticle? If so, please provide a supporting citation. Otherwise, consider changing to: “the nanovaccines were found to be stable at 4 degrees in up to XX days.”
 - Line 172 (PDF) indicates that the nanoparticles displayed “good biocompatibility” in Fig. S2. Please provide a citation supporting that the concentration range tested is standard for this type of nanoparticle.
 - Lines 176-178 (PDF): please provide a citation supporting that PBS+0.1% w/v Tween 80 mimics the microenvironment *in vivo*.
 - When describing nanoparticle preparation in Lines 161-164 (PDF), please provide citations supporting these statements.
- In neutralization assays, datapoints should still be provided for gp42 formulations in epithelial cells if the experiment was performed, even if negative results were obtained. Additionally, it would be useful if the full serum titrations used to arrive at ID50 were provided as Supplementary material, together with ID90.
- Since multiple glycoproteins can contribute to neutralizing activity within each donor, a multivariable analysis would have been more suitable For Fig. 1. For example, a single donor that has high levels of gp42-specific antibodies can also have high levels of gp350-specific antibodies, both of which could contribute to high neutralizing activity in B cells; however the combined contribution of both types of antibody cannot be teased out when performing individual analysis using Pearson’s correlation. Ideally, glycoprotein-specific antibody depletion should be performed to confirm results of correlative studies.
- It would be ideal to complement SDS-PAGE characterization of soluble proteins in Fig. 1K with western blots using specific antibodies against each glycoprotein. These assays should also be provided for gp350, since it was used in ELISA assays.
- SDS-PAGE analysis of gB showed two bands. Ideally, gB should be uniform in conformation for nanoparticle incorporation, which could entail inactivating the furin cleavage site to prevent cleavage. If this is not possible, then percentages of each gB form incorporated into the nanoparticles should be provided, which would be a regulatory pre-requisite for clinical translation, or at the very least this should be addressed in the text.
- Line 201 (PDF) indicates that the nanovaccines “induced [...] more durable antigen-specific antibodies than the corresponding free-form groups.” This statement is not well-supported as while it is true that the antibody levels are higher in the nanovaccine groups, the overall decay trend is the same for all groups: antibody levels for both nanovaccine and free-antigen groups begin decaying between Week5-8 and continue decaying until the end of the study.
- Line 205 (PDF) indicates that the nanovaccines induced “much higher” titers of neutralizing antibodies than free-form groups; however, based on the statistics provided this is only true for alum-adjuvanted groups (three-star statistical significance). All instances of nanoparticle comparison to free-antigen only display one-star statistical significance, thus the statement should change to “higher” as opposed to much higher.
- Line 215 (PDF) indicates that “NP-gHgL induced the highest neutralizing antibody titers” among the different vaccine formulations, but there was no official statistical comparison compared between the three to support this statement.
- Line 127 (PDF) compares neutralization across cell lines, but this comparison is not possible/appropriate given that different batches/sources of EBV are used for each cell line.

- Line 222 (PDF) makes light on the fact that the data indicates a Th1- polarized response. While this is true, the statement is dampened by the fact that C57BL/6J mice are biased towards Th1 responses, and the authors should mention/discuss this to provide the full picture.
- Line 233 (PDF) indicates the “nanovaccine formulations were the most efficient at increasing the effector-memory CD4+ and CD8+ T cell populations,” however this is not true in the case of CD4 for gB and gp42 in Fig. S8B.
- Full data and statistics should be provided for spleen in Fig. 2L as was done for blood in Fig. S8.
- Line 316 (PDF) indicates that “The combined antigens in NP-cocktail did not impact each other’s immunogenicity and titers;” however, this comparison is not possible because it is assumed that the individual NP experiments were performed on different days than the cocktail NP experiments. To make this claim, the authors had had to either perform all immunizations for comparison and associated assays at the same time, or if this wasn’t possible, to perform the assays on the same day, with the same conditions and same reagents.
- Line 316 (PDF) indicates that “The combined antigens in NP-cocktail did not impact each other’s immunogenicity and titers;” however, this comparison is not possible because it is assumed that the individual NP experiments were performed on different days than the cocktail NP experiments. To make this claim, the authors had had to either perform all immunizations for comparison and associated assays at the same time, or if this wasn’t possible, to perform the assays on the same day, with the same conditions and same reagents.
- Line 319 (PDF) indicates that “NP-cocktail elicited higher and longer-lasting total IgG titers [...] than the Free-cocktail and AI-cocktail.” However, as for Fig. 2, this statement is not well-supported in regards to response durability as the overall antibody decay trend is the same for all groups.
- Line 321 (PDF) indicates that “NP-cocktail was much more efficient than Free-cocktail and AI-cocktail” in regards to induction of cellular immune responses. However, no official statistical comparison is provided to support this claim.
- Line 330 indicates that “NP-cocktail vaccination induced more memory B cells and effector memory T cells in the spleen and PBMCs;” however this is not true for CD4+ T cells in PBMCs. Although it is implied, the authors should also finish the sentence to indicate what is being compared.
- All neutralization claims starting in Line 332 cannot be appropriately assessed if no infectivity information is provided for the EBV batches used for each assay.
- Line 432 indicates that “NP-cocktail induced the highest B cell and epithelial neutralizing titers;” however, there was no statistical difference between NP-cocktail and NP-gHgL in epithelial cells.
- There are several major concerns regarding the humanized mouse model described.
 - There should be rigor in confirming whether the PBMCs used for mouse reconstitution were EBV-infected or not – was ELISA or qPCR used to determine this?
 - How many donors were used for reconstitution? Were the donor PBMCs pooled or were distinct individual PBMCs used for individual animals? If they were not pooled, how is individual donor variability accounted for?
 - Where the PBMCs tested *in vitro* for Akata-EBV-GFP infection susceptibility?
 - Perhaps this reviewer is not familiar with this mouse model, but the % reconstitution illustrated in Fig. S19 seems rather low; can the authors comment if these are normal levels for this mouse model?
 - Since full PBMCs are used for reconstitution, can the authors comment on the potential impact of introducing previously existing EBV-specific T cells into this infection system on the experimental results?
 - Administration of 1mg of rabbit IgG per mouse is quite high compared to other recent passive immunization experiments in EBV humanized mouse models. Can the authors comment on how they arrived to this amount?

Minor comments:

- Figures contain too many panels, and as a consequence the font size in the panels is very small and hard to read. We understand this might be the result of journal restrictions on Figure numbers, but we recommend the authors re-evaluate their figures and transfer some panels to Supplementary so that main figure panels can be bigger and more legible. The choice of colors also make it difficult to discern between different treatment groups in immunogenicity studies; in this case perhaps the authors can make use of different symbols to represent different groups to aid the reader?
- On that same note, the font size and choice of colors make it very hard to read and understand the statistics in immunogenicity studies; would it be possible for the authors to use traditional lines to indicate group differences if the panels cannot be made bigger?
- It would be helpful for the reader if p-values are integrated in parenthesis when discussing statistical differences between groups within the text.
- Please include animal numbers and sex for every animal experiment both in the Methods and in the Figure legends.
- BALB/c mice are mentioned in the Methods, but they are not mentioned either in the text or Supplementary table 1; please correct/clarify as necessary.
- CHO-K1 cells are used in cell-cell fusion assays, but are not listed in the *Cell lines* section of the Methods; please add as necessary.
- In the *Neutralization assay* section in the Methods, please indicate/clarify whether all neutralizations (human, mouse, rabbit) were performed the same way, or if there were any species/experiment-specific changes.
- Please provide demographic information for the blood donors from Fig. 1, if available.
- Please address on the discussion how Fig. 1 results compare to those recently published by Bu et al in *Immunity* 2019 May 21;50(5):1305-1316.e6 (doi: 10.1016/j.immuni.2019.03.010.).
- Please provide information regarding the source of the gp350 protein used in ELISA assays in the Methods, as was provided for gHgL, gB and gp42.
- The Y-axis in Fig. 1A and F are labeled as “Neutralizing titers,” however, ID50 is provided; please modify to indicate ID50, as was done for all other neutralization panels in the manuscript for consistency.
- Line 132 (PDF) refers to Fig. 1A and F as IgG titrations, but these are neutralization results; please correct as necessary.
- Y-axis in Fig. 2F should be re-labeled to “IgA level,” since no titration was performed.
- It would be more helpful for the reader if Fig. 3A, B and C go to supplementary (Fig. S11), and instead, content in Fig. S11 comes to the main figure.
- Lines 283-285 (PDF): “[...] were encapsulated with two adjuvants (NP-) and their mixtures as controls (-Free”, please re-phrase for clarity.
- Line 287 (PDF): can axillary LNs be considered “distal” in this case, given that in mice the inguinal and axillary LNs are directly connected?
- Line 288 (PDF): “much fewer of the free-form antigens were detected” cannot be appropriately used since the antigens were not directly quantified – instead please use “less fluorescence associated with free-form antigens was detected.”
- Line 302 (PDF) uses the term “potent” to describe GC formation; can the authors define this?
- Fig. 4A, particularly the pie chart, does not contribute to the overall results and it might be better to remove it to make more space available for the rest of the panels.
- Units should be defined for Fig. 4F and G in the Figure legend.
- For the chimeric 6H2 antibody, please provide data showing that chimerization abrogated mouse reactivity, thus not interfering with the competitive ELISA.
- Fig. S15 would benefit from having more descriptive titles for each panel; for example, panel A can be renamed from “spleen” to “Memory B cells in spleen.”

Reviewer #2

- In Fig. 5D-F, why is the pre-immune/pre-bleed value not provided? Why is this not represented as a kinetic analysis as was done for mouse studies?
- In Fig. 5G, why are statistics provided for the gp42 group in epithelial cells, but no datapoints?
- Line 438 uses the term “high quality” serum; can the authors define this?
- There seems to be quite a high background for both hCD20 and EBER staining in Fig. 7.

Thank you again for submitting your manuscript "A cocktail nanovaccine targeting key entry glycoproteins elicits high-quality neutralizing antibodies against EBV infection" to Nature Communications. We have now received reports from 3 reviewers and, after careful consideration, we have decided to invite a major revision of the manuscript.

As you will see from the reports copied below, the reviewers raise important concerns. We find that these concerns limit the strength of the study, and therefore we ask you to address them with additional work. Without substantial revisions, we will be unlikely to send the paper back to review. In particular, a revised manuscript will need to provide additional experimental data and/or analysis to address concerns about the humanized mouse model regarding the PBMCs used as raised by reviewer #2, as well as about insufficient characterization of CD4⁺ T cell response and neutralization activity (raised by reviewers #1 and #2), insufficient characterization of infectivity *in vitro* (reviewer #2) and insufficient characterization of biochemical function in safety studies as raised by reviewer #3. Please ensure that these and all other concerns raised by our reviewers are addressed in full in a revised manuscript.

If you feel that you are able to comprehensively address the reviewers' concerns, please provide a point-by-point response to these comments along with your revision. Please show all changes in the manuscript text file with track changes or colour highlighting. If you are unable to address specific reviewer requests or find any points invalid, please explain why in the point-by-point response.

Response: We thank the reviewers and editors for thoroughly examining our manuscript and for providing valuable comments and suggestions. The points raised by the reviewers helped us improve our manuscript overall. We have revised our manuscript according to the reviewers' comments and suggestions.

Especially, we provided additional experiments to address concerns about the PBMCs used in the humanized mouse model and to evaluate CD4⁺ T cell responses. Besides, we added details of the *in vitro* neutralization assays. We also detected biochemical markers in the sera of immunized mice and rabbits in safety assessment studies. Below, we provide our point-by-point responses and indicate the changes

made to the revised manuscript.

REVIEWER COMMENTS

Reviewer #1 (Remarks to the Author):

Manuscript Nr: NCOMMS-23-44668-T

Zhong et al., “A cocktail nanovaccine targeting key entry glycoproteins elicits high-quality neutralizing antibodies against EBV infection”

The authors first correlated Epstein Barr virus (EBV) glycoprotein (gp350, gH, gL, gB and gp42) specific antibodies of healthy virus carriers with EBV neutralizing activity for B and epithelial cell infection. They then constructed nanovaccines containing recombinant gHgL, gB or gp42 proteins plus TLR9 or TLR4 agonists as adjuvants. According to their involvement in B and epithelial cell infection gHgL containing nanoparticles induced the highest neutralizing titers against B and epithelial cell infection and gp42 containing nanoparticles induced neutralizing activity for B cell infection. The nanoparticles were also able to mature dendritic cells and drain to lymph nodes for prolonged antigen retention. The authors then combined their three nanoparticle vaccines to elicit even higher EBV neutralizing antibody titers. With defined monoclonal antibodies against gHgL, gB and gp42 of highly EBV neutralizing ability it is then shown that many of the specificities that are elicited with the cocktail of nanoparticle vaccines overlap with the respective neutralizing epitopes. Immunizations were then repeated in rabbits with similar vaccination efficacy and antigen hierarchy. Rabbit IgG was then used to protect SCID-beige mice with human peripheral blood mononuclear cell (PBMC) transfer from EBV infection. Indeed, protection from EBV infection induced weight loss, viremia and lymphoma formation could be achieved. From these data the authors conclude that a mixed nanoparticle vaccine could elicit sterilizing immunity to prevent EBV associated diseases.

The reported study is interesting but similar to several other studies that have addressed combinations of EBV glycoproteins in eliciting neutralizing antibodies that protect humanized mice. Therefore, the novelty of the reported strategy remains unclear and the ambition to induce sterilizing immunity is questionable.

Response: We appreciate your supportive comments and constructive suggestions, which significantly improved our work. We address all the individual comments

below.

The major targets of prophylactic vaccines against EBV are the envelope glycoproteins contributing to the infection process, including gp350, gHgL, gB and gp42. Previous vaccine studies mainly focused on the most abundant glycoprotein gp350, which failed to induce sterile immunity in clinical trials¹. More recent preclinical studies showed that ferritin nanoparticles containing gH/gL/gp42 induced much higher neutralizing antibody titers compared to ferritin-gp350 and passive transfer of antibodies induced by nanoparticle displaying gH/gL, gH/gL/gp42+gp350D₁₂₃ or gH/gL+gp350D₁₂₃ protected humanized mice from EBV-associated lymphoma^{2,3,4}. These studies indicated that the combination of glycoproteins is promising for the development of prophylactic vaccines against EBV. In this study, we focused on the EBV key entry glycoproteins gHgL, gB and gp42. We characterized and applied a novel nanoparticle delivery system to co-deliver EBV antigen and adjuvants and we thoroughly evaluated the biodistribution, mechanisms and immune responses of the nanovaccines. Nanovaccines were efficiently transported to the lymph nodes, stimulated the DCs and induced potent humoral and cellular immune responses. We determined that the combination of NP-gHgL, NP-gB and NP-gp42 generated the highest neutralizing titers in mice and rabbits, which also efficiently protected humanized mice from lethal EBV challenge.

We agreed that the conclusion on the induction of sterilizing immunity by the cocktail nanovaccines somewhat ambitious for EBV. To address this possibility, we further evaluated the passive protection effect of immunized mice sera collected at week 5 and week 50 in the humanized mouse model. We modified the manuscript according to these new experimental data.

Major comments:

1. The authors report T cell responses that are induced by their vaccine formulations but the protective capacity of these remains unclear. For the induced neutralizing antibodies follicular helper CD4⁺ T cell induction might be more important than Th1 and CD8⁺ T cell responses and it would be informative to assess the capacity of the

nanoparticle vaccines to induce these.

Response: Thank you for your comments. The follicular helper CD4⁺ T (Tfh) cells play crucial roles in the formation of germinal centers where B cell go through affinity maturation and in the induction of durable protective humoral immune responses^{5,6}. Hence, we determined that nanovaccines induced higher levels of Tfh cells in the spleen of immunized mice while alum-adjuvanted vaccines induced weak Tfh cell responses (Figure 3F, also shown as appendix Figure 1). We described these results in the revised manuscript (Line 383-385). We also updated the Methods and Materials of the Tfh cell detection (Line 1118-1131).

Appendix Figure 1. Tfh cells generation in spleens of mice immunized by different vaccine formulations.

Data are shown as mean \pm SEM (n=5). Statistical analysis was performed using one-way ANOVA.

2. For their protection studies the authors used PBMC transfer models into SCID-beige mice that are then infected with 2.5×10^4 infectious EBV particles. They transfer then 1mg of rabbit IgG of their different vaccination groups. Why rabbit IgG and not IgG from the immunized mice? Would the IgG equivalent of one vaccinated mouse protect one SCID-beige-PBMC mouse from EBV infection? In previous studies sometimes a multitude of vaccinated needed to be used as antibody donors to protect recipients from EBV challenge.

Response: We greatly appreciate your questions. We used rabbit IgG to protect humanized mice from EBV challenge because a much higher blood volume can be

collected from rabbit and sufficient IgG could be purified from the serum of a single immunized rabbit. In previous studies, a large number of donor immunized mice (20~80) were needed for IgG purification and passive protection assay^{3,4}.

We agree with the reviewer that it is hard to say whether the 1 mg rabbit IgG used in passive protection assays reflects the IgG present in one vaccinated mouse. To address this concern, we repeated the experiment using pooled sera from immunized mice collected at week 5 to passively protect humanized mice from EBV challenge (Supplementary Figure 34, also shown as appendix Figure 2). We pooled 100 μ l sera of each immunized mouse (n=5) from NP-gHgL, NP-gB, NP-gp42, NP-cocktail, Free cocktail and PBS group, respectively. Then we administrated the pooled sera to the humanized mice at a dose of 100 μ l per mouse. 24 hours post sera inoculation, humanized mice received the EBV challenge. The body weight of humanized mice administrated with sera from mice inoculated with PBS significantly decreased 2 weeks after EBV challenge (Supplementary Figure 34B, also shown as appendix Figure 2B). Besides, except for the humanized mice that received sera from NP-cocktail immunized mice and the unchallenged group, at least one mouse died after EBV challenge in each of the other groups (Supplementary Figure 34C, also shown as appendix Figure 2C). EBV DNA copy numbers also significantly increased in mice of those groups (Supplementary Figure 34D and 34E, also shown as appendix Figure 2D and 2E). Only one mouse administrated with sera from NP-cocktail immunized mice showed a detectable increase of EBV DNA in PBMCs (Supplementary Figure 34E, also shown as appendix Figure 2E). Overall, the passive protection experiment with immune sera confirmed that sera from NP-cocktail immunized mice protected humanized mice from EBV-associated lymphoma (Supplementary Figure 34F-34G, also shown as appendix Figure 2F-2G).

We added these new results in the revised manuscript (Line 702-709) and we also updated the Materials and Methods accordingly (Line 1248-1253).

Appendix Figure 2. Sera collected at week 5 from mice immunized with NP-cocktail protected humanized mice from EBV associated lymphoma.

(A) Schematic illustration of humanized mice reconstitution and passive protection against EBV lethal challenge. SCID-beige mice were irradiated and administrated with clodronate liposomes before being reconstituted with PBMCs from healthy donors (EBV seropositive). Four weeks later, humanized mice were inoculated with sera of mice immunized with different vaccines. 25,000 GRU (green Raji units) Akata

strain EBV was injected 24 hours after mice sera administration. i.v., intravenously; i.p., intraperitoneally. The figure was created from Biorender.com. Humanized mice in the mock group were not challenged with EBV.

(B) Body weight of mice post-EBV challenge. Data are shown as mean \pm SEM.

Statistical analysis was performed using one-way ANOVA.

(C) Survival plot of humanized mice after EBV challenge.

(D-E) EBV DNA copy numbers in peripheral blood of each group. Each line represents an individual mouse and the dashed line indicates the detection limit.

(F) Numbers of humanized mice with different treatment developed EBV-associated lymphoma.

(G) Representative macroscopic tumors and tumor tissues stained by HE, hybridized for EBER, and immunostained for hCD20⁺ at necropsy (Scale bar=50 μ m).

The color of the asterisks denotes the group with which there is a significant difference (significant difference: * $P \leq 0.0332$; ** $P \leq 0.0021$; *** $P \leq 0.0002$; **** $P \leq 0.0001$; ns, no significant difference).

3. The aim of this study seems to be sterile protection from EBV infection which the authors seem to be able to achieve with the nanoparticle cocktail vaccination (gHgL, gB and gp42). It would be interesting to determine or at least discuss the durability of such a sterilizing immune response. Since EBV infection causes infectious mononucleosis and its sequelae (increased risk for Hodgkin's lymphoma and multiple sclerosis) upon delayed primary infection in young adults, waning sterilizing immunity would delay primary infection in vaccinees, possibly with an increased risk for infectious mononucleosis.

Response: Thank you for your comments. The reviewer raises a good question about delayed EBV infection. The delayed EBV infection in young adults without vaccination often develops infectious mononucleosis, which also increases the risk of Hodgkin's lymphoma and multiple sclerosis^{7,8}. To determine the durability of protective efficiency by sera induced by NP-cocktail, we performed additional passive protection assay using sera collected at week 50 from mice immunized with different

vaccine formulations. We pooled 100 μ l sera collected at week 50 of each vaccines immunized mouse (n=5) from NP-gHgL, NP-gB, NP-gp42, NP-cocktail, Free cocktail and PBS groups, respectively. Then we administrated pooled sera from each group to the humanized mice at a dose of 100 μ l per mouse. 24 hours post sera inoculation, humanized mice received the EBV challenge (Supplementary Figure 35, also shown as appendix Figure 3). After challenge, the body weight of humanized mice pre-inoculated with sera from mice immunized with NP-cocktail remained stable and only one humanized mouse in this group died at week 8 post EBV challenge (Supplementary Figure 35B and 35C, also shown as appendix Figure 3B and 3C). In contrast, two out of five mice treated with sera from NP-gHgL, NP-gB, NP-gp42 and Free-cocktail immunized mice died after EBV challenge, respectively (Supplementary Figure 35C also shown as appendix Figure 3C). An increase in EBV copy numbers was detected in only one of the humanized mice treated with sera from NP-cocktail immunized mice and all these mice were free of lymphoma (Supplementary Figure 35D-35F, also shown as appendix Figure 3D-3F). In contrast, two mice treated with sera from NP-gHgL, NP-gB and Free-cocktail immunized mice, respectively and one mouse treated with sera from NP-gp42 developed EBV-associated lymphoma (Supplementary Figure 35F and 35G, also shown as appendix Figure 3F and 3G). The *in vivo* protection efficiency decreased from sera collected at week 5 to sera collected at week 50 (Appendix Figure 2 and Appendix Figure 3). Hence, we thought it is inappropriate to say that NP-cocktail induced sterile immunity.

In the revised manuscript, we corrected the conclusion about 'sterile immunity' and referred to durable protective responses, which is supported by new results obtained with sera collected at week 50 from immunized mice. We modified the manuscript (Line 710-725) and discussed the durability of protective effect in discussion part (Line 815-822). We also updated the Materials and Methods accordingly (Line 1248-1253).

Appendix Figure 3. Sera collected at week 50 from mice immunized with NP-cocktail protected humanized mice from EBV associated lymphoma.

(A) Schematic illustration of humanized mice reconstitution and passive protection against EBV lethal challenge. SCID-beige mice were irradiated and administrated with clodronate liposomes before being reconstituted with PBMCs from healthy donors (EBV seropositive). Four weeks later, humanized mice were inoculated with

sera of mice immunized with different vaccines. 25,000 GRU (green Raji units) Akata strain EBV was injected 24 hours after mice sera administration. i.v., intravenously; i.p., intraperitoneally. The figure was created from Biorender.com. Humanized mice in the mock group were not challenged with EBV.

(B) Body weight of mice post-EBV challenge. Data are shown as mean \pm SEM.

Statistical analysis was performed using one-way ANOVA.

(C) Survival plot of humanized mice after EBV challenge.

(D-E) EBV DNA copy numbers in peripheral blood of each group. Each line represents an individual mouse and the dashed line indicates the detection limit.

(F) Numbers of humanized mice within different treatment groups, which developed EBV-associated lymphoma.

(G) Representative macroscopic tumors and tumor tissues stained by HE, hybridized for EBER, and immunostained for hCD20⁺ at necropsy (Scale bar=50 μ m).

The color of the asterisks denotes the group with which there is a significant difference (significant difference: * $P \leq 0.0332$; ** $P \leq 0.0021$; *** $P \leq 0.0002$; **** $P \leq 0.0001$; ns, no significant difference).

Minor comments:

1. Interferon should be consistently abbreviated as IFN and not INF as in line 232.

Response: Thanks for pointing out the mistake. We have corrected it in the revised manuscript as follows (Line 302-305): ‘Compared to non-encapsulated form vaccines, nanovaccines induced robust T cell responses manifested by the increased number of antigen-specific TNF- α , IFN- γ and IL-2 producing T cells after restimulation (Figure 2I-2K).’

Reviewer #2

We commend the authors for their thorough and detailed work on their manuscript “A cocktail nanovaccine targeting key entry glycoproteins elicits high-quality neutralizing antibodies against EBV infection.” In this work, the authors developed a nanoparticle-based vaccine approach against Epstein-Barr virus (EBV) infection targeting multiple EBV entry glycoproteins, namely gHgL, gB and gp42. First, the authors analyzed the neutralizing capability of serum from individuals naturally infected with EBV, and performed correlative studies to determine which EBV antigens contributed to the neutralizing activity observed. Next, the authors developed individual nanoparticles incorporating either gHgL, gB or gp42, together with two adjuvants, CpG and MPLA. The nanoparticles were first characterized in vitro for their biochemical properties and composition, and were then tested for immunogenicity in C57BL/6J mice as compared to free soluble glycoproteins. Serum antibody responses, serum neutralizing capability, specific T cell and B cell responses, and biodistribution were all characterized. Then, the authors prepared a combination cocktail formulation that incorporated all three individual nanoparticles, and tested immunogenicity of this new formulation in mice as before, followed by immunogenicity assessment in rabbits. Using serum from immunized rabbits, the authors then finally tested vaccine efficacy against EBV infection in humanized mice in passive immunization experiments followed by viral challenge. While the results of these studies are encouraging and support inclusion of multiple glycoproteins in EBV vaccine design, we believe that the manuscript in its current condition is not ready for publication. Overall, this reviewer found that some key experimental details were either missing or hard to find. The authors also make several “grandiose” claims with minimal evidence to support them, and there are a few important concerns regarding the humanized mouse model used for efficacy studies. Based on this assessment, I recommend that the authors majorly revise their submission before publication on Nature Communications. Below find a point-by-point summary of our review for the authors to address.

Response: We appreciate your positive comments about our findings and the

constructive suggestions. Your suggestions have significantly improved our work. We address all the comments below.

Major comments:

1. For appropriate rigor in *in vitro* neutralization experiments, it is imperative that the authors provide infectivity information for the EBV batches used in both HNE1 and Akata cell neutralizations: how infectious are 20 μ l of CNE2-EBV-GFP in Akata cells, and how infectious are 50 μ l of Akata-EBV-GFP in HNE1 cells? The authors should at the very least provide the level of infection (%) achieved in the absence of antibodies/serum for every neutralization experiment, but ideally should provide full infectivity titration curves for each batch of virus. Without this information, it is not possible to fully evaluate the impact of the obtained results; i.e. neutralizing 2% infectivity is not the same as neutralizing 20% infectivity. Similarly, the authors should provide methods and data as to how they titrated the virus used for *in vivo* humanized mice experiments.

Response: Thank you for your comments. We agreed with you that the different infectivity of different batches of virus could significantly influence the results of neutralization assays. We used 20 μ l of CNE2-EBV-GFP in Akata cells and 50 μ l of Akata-EBV-GFP in HNE1 cells in order to achieve similar infection ratios, approximately 30% (Supplementary Figure 36A and 36B, also shown as appendix Figure 4A and 4B). Besides, the same batch of the CNE2-EBV-GFP was used in all B cell neutralization assays and the same batch of the Akata-EBV-GFP was used in all epithelial cell neutralization assays. The infectivity of Akata-EBV-GFP virus is much less than the CNE2-EBV-GFP virus and 50 μ l virus is needed to reach 30% infectivity (Supplementary Figure 36C and 36D, also shown as appendix Figure 4C and 4D). As requested, we show the titration for the viruses used in *in vivo* humanized mice experiment by adding 100 μ l of CNE2-EBV-GFP in 1×10^5 Raji cells and calculated the green Raji units (GRU) of the virus according to previous studies by the following formula: 'Total number of Raji cells \times percentage of GFP⁺ cells/volume of the virus stock used'^{9,10}. Humanized mice were administrated with 100 μ l EBV, which is

equivalent to 25,000 GRU (Supplementary Figure 36E, also shown as appendix Figure 4E). We also added the methods and data about the titration of virus and the calculation of GRU in the revised manuscript (Line 1079-1098) (Supplementary Figure 36) and added the relative references.

Appendix Figure 4. Titration of virus stocks uses in neutralization assays and

infections of humanized mice.

(A-B) Percentage of infected cells (GFP-positive) achieved in the absence of antibodies/serum for epithelial cell neutralization experiment (A) and B cell neutralizing experiment (B). NC represents cells did not incubated with virus and PC1-PC3 represents three independent replicates of positive control.

(C-D) Infectivity titration curves for CNE2-EBV-GFP virus on Akata cells (C) and Akata-EBV-GFP on HNE1 cells (D) used in neutralization assay. Data are shown as mean \pm SEM of two independent replicates.

(E) GRU determination of the virus dose used in humanized mice assays.

2. Although the authors present the argument that gp350 has not proved to be sufficient as a lone antigen in EBV vaccine design, they did not include a gp350 control group in their immunogenicity experiments, either in nanoparticle form or in free form, despite showing that gp350-specific serum antibodies contribute the most to B cell neutralization in Figure 1. This is a missed opportunity that dampens the message of the manuscript, and should be addressed.

Response: Thank you for your comments. We added data on the immunogenicity of NP-gp350 formulation in the revised manuscript (Supplementary Figure 24, also shown as appendix Figure 5). As expected, sera from mice immunized with NP-gp350 showed no neutralizing ability of EBV infection of epithelial cells since gp350 is not essential for the epithelial cell infection process (Appendix Figure 5B) ¹¹. On the contrary, NP-gp350 elicited antibodies in mice sera that efficiently blocked EBV infection of B cells (Appendix Figure 5C). The B cell neutralizing titers of sera from NP-gp350 immunized mice were similar to those from NP-gHgL, NP-gB and NP-gp42 immunized mice (Appendix Figure 5D). Interestingly, although antibodies against gp350 are major contributors to neutralization of EBV infection of B cells in healthy EBV carriers, NP-cocktail immunization still induced higher B cell neutralizing titers (Appendix Figure 5D) ². We added the results in the revised manuscript (Line 455-465). We also updated the Methods and Materials accordingly.

Appendix Figure 5. Immunogenicity evaluation of NP-gp350.

(A) The total IgG titers of sera collected from C57BL/6J mice immunized with NP-gp350. Data points are shown as the mean \pm SEM (n=5). Statistical analysis was performed using unpaired welch's t test (n=5).

(B-C) Neutralization of Akata-EBV infection of HNE1 epithelial cells (B) and CNE2-EBV infection of Akata B cells (C) by individual sera collected on day 35 from five C57BL/6J mice immunized with NP-gp350 (M1-M5).

(D) Comparison of B cell neutralizing titers of sera collected on day 35 from C57BL/6J mice immunized with cocktail (NP-gHgL+NP-gB+NP-gp42) or individual nanovaccines. The indicated half maximal inhibitory dilution fold (ID₅₀) values were calculated by sigmoid trend fitting. Data are shown as mean ID₅₀ (n=5). Statistical analysis was performed using one-way ANOVA.

3. The results of the present study have shown that gp42 is capable of generating neutralizing antibodies without being found in complex with gHgL, which is very exciting, in contrast to previous evidence showing that the gHgL/gp42 conformational

epitopes are critical for viral entry and generation of neutralizing antibodies. Can the authors expand/comment on this in the Discussion?

Response: Thank you for your comments. A previous study showed that gp42 binding to its receptor HLA-II require simultaneous binding to gHgL¹². Besides, the B cell specific neutralizing antibody (nAb) F-2-1 was reported to recognize the epitope of gp42 while gp42 was co-expressed with gHgL¹³. We also previously detected the binding of HLA-II to gp42 alone or gHgL/gp42 by surface plasmon resonance assay and isolated several nAbs against gp42 from rabbits immunized with gp42 alone^{14,15}. One of these nAbs, 5E3, was also able to bind gHgL/gp42 complex¹⁵. Additionally, sera from healthy EBV carriers efficiently bound to gp42 alone (Figure 1A). Hence, these results indicate that conformational neutralizing epitopes at least partial exposed for gp42 alone. We added the results in the revised manuscript (Line 770-780).

4. No long-term neutralizing activity was tested for any of the immunogenicity studies, and no neutralization was performed to compare titers before and after all vaccine boosters to determine whether all three vaccinations are required, which was a missed opportunity.

Response: Thank you for your comments. We have added the neutralizing curves of mice sera collected at week 1, 3, 5 and 50 (Supplementary Figure 8, 9, 19 and 20, also shown as appendix Figure 6-9) and rabbit sera collected at week 2, 4 and 6 (Supplementary Figure 26D-26I, also shown as appendix Figure 10) in the revised manuscript. Considering either neutralization of EBV infection of epithelial cells or B cells, neutralizing titers of sera induced by various vaccines increased after the two boosts and the neutralizing titers of sera collected after the third vaccination was the highest (Supplementary Figure 8, 9, 19, 20 and 26D-26I, also shown as appendix Figure 6-10). Also see our response to point 3 of Reviewer#1 about the *in vivo* protection durability (Shown above as appendix Figure 3).

Appendix Figure 6. B cell infection neutralizing abilities of sera from mice immunized with various vaccine formulations collected at different time points. (A-C) B cell infection neutralizing titers of sera collected at week 1 from mice immunized with gHgL-based vaccines (A), gB-based vaccines (B) and gp42-based vaccines (C), respectively. (D-F) B cell infection neutralizing titers of sera collected at week 3 from mice immunized with gHgL-based vaccines (D), gB-based vaccines (E) and gp42-based vaccines (F), respectively. (G-I) B cell infection neutralizing titers of sera collected at week 5 from mice immunized with gHgL-based vaccines (G), gB-based vaccines (H) and gp42-based vaccines (I), respectively. (J-L) B cell infection neutralizing titers of sera collected at week 50 from mice

immunized with gHgL-based vaccines (J), gB-based vaccines (K) and gp42-based vaccines (L), respectively.

Half maximal inhibitory dilution fold (ID50) was calculated by sigmoid trend fitting.

Data points are shown as the mean \pm SEM (n=5).

Appendix Figure 7. Epithelial cell infection neutralizing abilities of sera from mice immunized with various vaccine formulations collected at different time points.

(A-C) Epithelial cell infection neutralizing titers of sera collected at week 1 from mice immunized with gHgL-based vaccines (A), gB-based vaccines (B) and gp42-based vaccines (C), respectively.

(D-F) Epithelial cell infection neutralizing titers of sera collected at week 3 from mice immunized with gHgL-based vaccines (D), gB-based vaccines (E) and gp42-based vaccines (F), respectively.

(G-I) Epithelial cell infection neutralizing titers of sera collected at week 5 from mice immunized with gHgL-based vaccines (G), gB-based vaccines (H) and gp42-based vaccines (I), respectively.

(J-L) Epithelial cell infection neutralizing titers of sera collected at week 50 from mice immunized with gHgL-based vaccines (J), gB-based vaccines (K) and gp42-based vaccines (L), respectively.

Half maximal inhibitory dilution fold (ID50) was calculated by sigmoid trend fitting.

Data points are shown as the mean \pm SEM (n=5).

Appendix Figure 8. B cell infection neutralizing abilities of sera from mice immunized with cocktail vaccines collected at different time points.

(A) B cell infection neutralizing titers of sera collected at week 1 from mice immunized with NP-cocktail, Free-cocktail and AI-cocktail.

(B) B cell infection neutralizing titers of sera collected at week 3 from mice immunized with NP-cocktail, Free-cocktail and AI-cocktail.

(C) B cell infection neutralizing titers of sera collected at week 5 from mice immunized with NP-cocktail, Free-cocktail and AI-cocktail.

(D) B cell infection neutralizing titers of sera collected at week 50 from mice immunized with NP-cocktail, Free-cocktail and AI-cocktail. Half maximal inhibitory dilution fold (ID50) was calculated by sigmoid trend fitting. Data points are shown as the mean \pm SEM (n=5).

Appendix Figure 9. Epithelial cell infection neutralizing abilities of sera from mice immunized with cocktail vaccines collected at different time points.

(A) Epithelial cell infection neutralizing titers of sera collected at week 1 from mice immunized with NP-cocktail, Free-cocktail and AI-cocktail.

(B) Epithelial cell infection neutralizing titers of sera collected at week 3 from mice immunized with NP-cocktail, Free-cocktail and AI-cocktail.

(C) Epithelial cell infection neutralizing titers of sera collected at week 5 from mice immunized with NP-cocktail, Free-cocktail and AI-cocktail.

(D) Epithelial cell infection neutralizing titers of sera collected at week 50 from mice immunized with NP-cocktail, Free-cocktail and AI-cocktail.

Half maximal inhibitory dilution fold (ID50) was calculated by sigmoid trend fitting. Data points are shown as the mean \pm SEM (n=5).

Appendix Figure 10. Neutralizing abilities of sera from rabbits immunized with various vaccine formulations collected at different time points.

(A-B) B cell infection neutralizing titers (A) and epithelial cell neutralizing titers (B) of sera collected at week 2 from rabbit immunized with different vaccine formulations.

(C-D) B cell infection neutralizing titers (C) and epithelial cell neutralizing titers (D) of sera collected at week 4 from rabbit immunized with different vaccine formulations.

(E-F) B cell infection neutralizing titers (E) and epithelial cell neutralizing titers (F) of sera collected at week 6 from rabbit immunized with different vaccine formulations. Half maximal inhibitory dilution fold (ID50) was calculated by sigmoid trend fitting.

Data points are shown as the mean \pm SEM (n=5).

5. A sample size of n=3 animals per group in the rabbit studies does not allow for powered statistical comparisons.

Response: We appreciate your comment on sample size and performed the immunization assay in 5 rabbits of each group according to your suggestions (Figure 5, also shown as appendix Figure 11). We also updated the results in the manuscript (Line 554-580).

Appendix Figure 11. Rabbit immunization and induction of humoral immune responses.

(A) Schedule of rabbit immunization. The figure was created from Biorender.com. Rabbits were inoculated intramuscularly with different vaccine formulations three

times on day 0, 14 and 28 with doses described in Table S2. Blood was collected on day 0, 14, 28 and 42. Rabbits were euthanized on day 42.

(B-D) IgG titers of sera collected on day 42 from rabbits immunized with different vaccine formulations. (B) Anti-gHgL IgG titers elicited by NP-gHgL, NP-cocktail, Free-cocktail and PBS (C) Anti-gB IgG titers elicited by NP-gB, NP-cocktail, Free-cocktail and PBS. (D) Anti-gp42 IgG titers elicited by NP-gp42, NP-cocktail, Free-cocktail and PBS.

(E-F) Neutralization of CNE2-EBV infection of Akata B cells (E) and Akata-EBV infection of HNE1 epithelial cells (F) by sera collected on day 42 from rabbits immunized with different vaccine formulations. Half maximal inhibitory dilution fold (ID50) values were calculated by sigmoid trend fitting.

(G and H) Body weight (G) and temperature (H) were monitored after immunization of rabbits with the indicated vaccine formulations. Black arrows indicate times of immunization.

(B-H) Data points are shown as the mean \pm SEM (n=5). Statistical analysis was performed using one-way ANOVA.

6. When combining individual nanoparticles for the cocktail formulation, how does this affect the final concentration of CpG and MPLA in the vaccine? If we are going by Supplementary table 1, where 5 μ g individual nanoparticles are listed to contain 10 μ g of CpG/MPLA, then the cocktail formulation that contains 5 μ g of each individual nanoparticle should contain 30 μ g of CpG/MPLA, not 10 μ g as listed. If this is the case, is the adjuvant content in the free-cocktail also 30 μ g CpG/MPLA, or is it 10 μ g? Please correct/clarify as necessary.

Response: Thank you for your comments. When the nanovaccines were used singly, each nanoparticle dose contained 10 μ g of CpG/MPLA. When the nanovaccines were used as a cocktail, each nanoparticles contained 3.33 μ g of CpG/MPLA to make the total amount of the adjuvants the same as the single nanovaccine doses. Hence, the difference of immune responses only resulted from the combination of the glycoproteins instead of the amount of adjuvants. We clarified this issue in the

supplementary table 2 in the revised supplementary file (Line 441).

7. Please provide supporting citations as appropriate for the following:

- Line 170 (PDF) indicates “the nanovaccines showed long-term stability.” Are 30-40 days considered long-term at 4 degrees for this type of nanoparticle? If so, please provide a supporting citation. Otherwise, consider changing to: “the nanovaccines were found to be stable at 4 degrees in up to XX days.”

Response: Thank you for your comments. We agreed that 30-40 days cannot represent long-term stability. We rephrased this conclusion in the revised manuscript as follows (Line 220-221): ‘The size and immunogenicity of the nanovaccines were found to be stable at 4°C for up to 30 days.’

- Line 172 (PDF) indicates that the nanoparticles displayed “good biocompatibility” in Fig. S2. Please provide a citation supporting that the concentration range tested is standard for this type of nanoparticle.

Response: Thank you for your comments. According to your suggestion, we added the corresponding references in the revised manuscript (Line 224).

- Lines 176-178 (PDF): please provide a citation supporting that PBS+0.1% w/v Tween 80 mimics the microenvironment *in vivo*.

Response: Thank you for your comments. The PBS+0.1% w/v Tween 80 is a common condition for analyzing the release profile of nanoparticles¹⁶. It is inaccurate to say that this condition mimics the microenvironment *in vivo*. Hence, we deleted the description of ‘To simulate the microenvironment *in vivo*’ in the revised manuscript and we added the reference using PBS+0.1% w/v Tween 80 to determine the release profile of nanoparticles in the revised manuscript (Line 229).

- When describing nanoparticle preparation in Lines 161-164 (PDF), please provide citations supporting these statements.

Response: Thank you for your comments. According to your suggestion, we added the corresponding reference in the revised manuscript (Line 211).

8. In neutralization assays, datapoints should still be provided for gp42 formulations in

epithelial cells if the experiment was performed, even if negative results were obtained. Additionally, it would be useful if the full serum titrations used to arrive at ID50 were provided as Supplementary material, together with ID90.

Response: Thank you for your comments. Since no neutralizing ability was detected for sera collected from mice immunized with gp42-based vaccines, the ID50 cannot be calculated. We provided the curves of sera neutralization experiments (including sera induced by gp42-based vaccines) in supplementary Figures 8, 9, 19, 20 and 26 (also shown as appendix Figure 6-10). According to the reviewer's suggestions, we also provide the ID₅₀ and ID₉₀ of the sera from immunized animals in the supplementary Table 3 (also shown as appendix Table 1)

Appendix Table 1. ID₅₀ and ID₉₀ of the sera from immunized animals

Infection model	Animal	Group	ID50	ID90
B cell infection neutralization	C57BL/6J	NP-gHgL	88.93	8.25
		Free-gHgL	49.94	NA
		Al-gHgL	29.71	2.287
		NP-gB	92.70	9.52
		Free-gB	36.88	NA
		Al-gB	36.41	5.32
		NP-gp42	137.90	50.37
		Free-gp42	88.02	27.74
		Al-gp42	35.38	NA
		NP-cocktail	290.6	91.55
		Free-cocktail	104.2	27.71
		Al-cocktail	60.59	15.18
Epithelial cell infection neutralization	C57BL/6J	NP-gHgL	853.60	217.11
		Free-gHgL	183.70	25.90
		Al-gHgL	76.90	NA
		NP-gB	516.0	121.84
		Free-gB	136.4	21.96
		Al-gB	54.59	NA

		NP-gp42	NA	NA
		Free-gp42	NA	NA
		AI-gp42	NA	NA
		NP-cocktail	2675.00	711.61
		Free-cocktail	996.40	128.90
		AI-cocktail	498.00	17.44
B cell infection neutralization	New	NP-gHgL	226.00	49.27
		NP-gB	250.10	86.15
	Zealand white rabbits	NP-gp42	386.00	82.0
		NP-cocktail	850.8	143.669
		Free-cocktail	242.0	49.249
Epithelial cell infection neutralization	New	NP-gHgL	1688	193.37
		NP-gB	791.7	47.23
	Zealand white rabbits	NP-gp42	NA	NA
		NP-cocktail	2902	467.32
		Free-cocktail	881.6	54.65

The sera were diluted from 1:10 and ID50 as well as ID90 values were calculated by sigmoid trend fitting using GraphPad Prism 8.0. NA means the value can not be calculated.

9. Since multiple glycoproteins can contribute to neutralizing activity within each donor, a multivariable analysis would have been more suitable For Fig. 1. For example, a single donor that has high levels of gp42-specific antibodies can also have high levels of gp350-specific antibodies, both of which could contribute to high neutralizing activity in B cells; however the combined contribution of both types of antibody cannot be teased out when performing individual analysis using Pearson's correlation. Ideally, glycoprotein-specific antibody depletion should be performed to confirm results of correlative studies.

Response: Thank you for your interesting observation. According to your suggestion, we performed the glycoprotein-specific antibody depletion assays to dissect the neutralizing contribution of each glycoprotein-specific antibody in healthy EBV

carriers' sera according to your suggestions (Figure 1A-1C and S2, also shown as appendix Figure 12). We performed serum antibody depletion by 293T cells overexpressing each glycoprotein. After cognate depletion, the IgG titers against gp350, gHgL, gB, and gp42 were decreased by more than 90% (Supplementary Figure 2, also shown as appendix Figure 12C). Using B cell and epithelial cell infection models, we determined the neutralizing titer of each serum before and after depletion of over 90% of each glycoprotein-specific antibody. Compared to undepleted sera, the B cell neutralizing activity decreased by approximately ~37.72%, ~16.53%, ~23.18% and ~16.95% for gp350, gHgL, gB and gp42-specific antibody depletion, respectively (Figure 1C, also shown as appendix Figure 12D). The epithelial cell neutralizing titers decreased on average by ~3.04%, ~45.13%, ~27.14% and ~3.65% for gp350, gHgL, gB and gp42-specific antibody depletion, respectively (Figure 1C, also shown as appendix Figure 12E). gp350- and gp42-specific antibody depletion did not change the epithelial neutralizing ability of sera because they are not needed for EBV infection of epithelial cells (Figure 1C, also shown as appendix Figure 12E). However, gp42-specific antibodies in sera were also important components for B cell neutralization (Figure 1C, also shown as appendix Figure 12D). Besides, gB as the fusogen playing key roles in EBV infection of B cells and epithelial cells, gB-specific antibodies in sera contributed to neutralization of EBV infection in both epithelial cells and B cells (Figure 1C, also shown as appendix Figure 12D and 12E). Consistent with a previous study by Bu. *et.al*, gp350-specific antibodies are the major component in healthy EBV carriers' sera to neutralize EBV infection of B cells while gHgL-specific antibodies are the most important antibodies in healthy EBV carriers' sera to neutralize EBV infection of epithelial cells ².

We added the results in the revised manuscript (Line 167-204). We also added the relative methods in the Materials and Methods (Line 890-899).

Appendix Figure 12. Evaluation of the neutralizing contribution of glycoprotein-specific antibodies in healthy EBV carriers.

(A) anti-gp350, anti-gHgL, anti-gB and anti-gp42 IgG titers of healthy EBV carriers' sera.

(B) Neutralization of CNE2-EBV infection of Akata B cells (left panel) and Akata-EBV infection of HNE1 cells (right panel) by sera collected from 32 healthy EBV carriers. Half maximal inhibitory dilution fold (ID50) was calculated by sigmoid trend fitting.

(C) Reduction of glycoprotein-specific IgG titers after depletion by cells overexpressing specific glycoprotein in sera from 32 healthy EBV carriers. The percentage of IgG titer reduction was calculated by the equation $(1 - \text{IgG titer-depleted} / \text{IgG titer-before}) \times 100\%$.

(D) Reduction of glycoprotein-specific antibodies depleted sera neutralizing ability of EBV infection of Akata B cells and HNE1 cells. The percentage of neutralizing reduction was calculated by $(1 - \text{ID50-depleted} / \text{ID50-before}) \times 100\%$.

Data represent the mean \pm SEM (n=32).

10. It would be ideal to complement SDS-PAGE characterization of soluble proteins in Fig. 1K with western blots using specific antibodies against each glycoprotein.

These assays should also be provided for gp350, since it was used in ELISA assays.

Response: Thank you for your comments. We added the Western blots for each purified soluble glycoproteins in the revised manuscript (supplementary Figure 1B-1D, also shown as appendix Figure 13A-13C) and we also added the SDS-PAGE and Western blot for gp350 in the supplementary Figure 1E-1F (also shown as appendix Figure 13D-13E).

Appendix Figure 13. SDS-PAGE and western blot analysis for each glycoprotein.

(A) Western blot analysis of gHgL detected by 10H3 (our group unpublished data) under reducing SDS-PAGE.

(B) Western blot analysis of gB detected by 3A3 and 3A5 under reducing SDS-PAGE. 3A3 bound the ~ 70 kDa fragment and 3A5 bound the ~ 40 kDa fragment.

(C) Western blot analysis of gp42 detected by 4H8 under reducing SDS-PAGE.

(D-E) Reducing SDS-PAGE analysis (D) and Western blot analysis (E) of gp350N detected by 3D2 (our group unpublished data) under reducing SDS-PAGE.

11. SDS-PAGE analysis of gB showed two bands. Ideally, gB should be uniform in conformation for nanoparticle incorporation, which could entail inactivating the furin cleavage site to prevent cleavage. If this is not possible, then percentages of each gB form incorporated into the nanoparticles should be provided, which would be a regulatory pre-requisite for clinical translation, or at the very least this should be addressed in the text.

Response: Thank you for your comments. We used the same sequence reported by Longnecker *et.al.*, to express EBV gB without Furin cleavage site mutation. Hence, gB can be cleaved in two fragments linked by disulfide bonding during the expression process. Hence, gB showed two bands in reducing SDS-PAGE¹⁷. We also analyzed the gB before encapsulation and gB released from nanoparticles, both forms of gB showed as trimer under the native non-reducing conditions (Supplementary Figure 5A, also shown as appendix Figure 14A). This indicates that all gB encapsulated and released was uniform in a trimer conformation. Under the reducing condition, gB before encapsulation and released gB both appeared as two bands (Supplementary Figure 5B, appendix Figure 14B). We added the results in the revised manuscript (Line 235-240).

Appendix Figure 14. Non-reducing native PAGE (A) and reducing SDS-PAGE (B) analysis of gB before encapsulation and gB released from nanoparticles.

Δ denotes heating at 100 °C for 10 min. One band with molecular weights of ~330kDa was observed in non-reducing native PAGE. Two bands with molecular weights of ~ 70 kDa and ~ 40 kDa were observed in the reducing SDS-PAGE.

12. Line 201 (PDF) indicates that the nanovaccines “induced [...] more durable antigen-specific antibodies than the corresponding free-form groups.” This statement is not well-supported as while it is true that the antibody levels are higher in the nanovaccine groups, the overall decay trend is the same for all groups: antibody levels for both nanovaccine and free-antigen groups begin decaying between Week5-8 and continue decaying until the end of the study.

Response: Thank you for your comments. We rephrased this sentence in the revised manuscript as follows (Line 257-260): ‘The nanovaccines, NP-gHgL, NP-gB and NP-gp42, induced significantly higher antigen-specific antibodies than the corresponding free-form groups, while the alum-adjuvanted vaccines elicited the lowest antibody titers (Figure 2A-2C and S11A-S11C)’

13. Line 205 (PDF) indicates that the nanovaccines induced “much higher” titers of neutralizing antibodies than free-form groups; however, based on the statistics provided this is only true for alum-adjuvanted groups (three-star statistical significance). All instances of nanoparticle comparison to free-antigen only display one- star statistical significance, thus the statement should change to “higher” as opposed to much higher.

Response: Thank you for your comments. We rephrased this sentence in the revised manuscript as follows (Line 265-268): ‘After three immunizations, each nanovaccine induced higher titers of neutralizing antibodies in serum against both B cell infection and epithelial cell infection compared to Free- and Al- groups (Figure 2D and 2E).’

14. Line 215 (PDF) indicates that “NP-gHgL induced the highest neutralizing antibody titers” among the different vaccine formulations, but there was no official statistical comparison compared between the three to support this statement.

Response: Thank you for your comments. We added the statistical analysis in the revised supplementary Figure 10 (also shown as appendix Figure 15). No significant difference was obtained between gHgL and gB based Nanovaccines. Hence, we deleted this conclusion in the revised manuscript (Line 279-280).

Appendix Figure 15. Comparison of Epithelial cell (left panel) and B cell (right panel) neutralizing titers of sera elicited by different nanovaccines.

Data points are shown as the mean \pm SEM (n=5). Left panel: Statistical analysis was performed using unpaired Welch's t test (n=5). Right panel: Statistical analysis was performed using one-way ANOVA.

15. Line 127 (PDF) compares neutralization across cell lines, but this comparison is not possible/appropriate given that different batches/sources of EBV are used for each cell line.

Response: Thank you for your comments. We agreed that it is inappropriate to compare neutralization across cell lines. We deleted the comparison in the revised manuscript (Line 280-282).

16. Line 222 (PDF) makes light on the fact that the data indicates a Th1- polarized response. While this is true, the statement is dampened by the fact that C57BL/6J mice are biased towards Th1 responses, and the authors should mention/discuss this to provide the full picture.

Response: Thank you for this insightful comment. We rephrased this sentence in the

revised manuscript as follows (Line 287-290): ‘Although C57BL/6J mice are biased to induce Th1 immune responses, the immune polarization mainly determined by adjuvants according to our data ¹⁸. Alum-adjuvanted vaccines still induced Th2-biased immune responses in the immunized mice (Figure 2G).’

17. Line 233 (PDF) indicates the “nanovaccine formulations were the most efficient at increasing the effector- memory CD4⁺ and CD8⁺ T cell populations,” however this is not true in the case of CD4 for gB and gp42 in Fig. S8B.

Response: Thank you for your comments. We rephrased this sentence in the revised manuscript as follows (Line 305-309): ‘Moreover, nanovaccine formulations were the most efficient at increasing the effector-memory CD4⁺ and CD8⁺ T cells populations in the spleen (Figure 2L, S13A and S13B). Besides, nanovaccines induced higher levels of effector-memory CD8⁺ T cells in peripheral blood and NP-gHgL elicited more effector memory CD4⁺ T cells in peripheral blood (Figure S13D and S13E).’

18. Full data and statistics should be provided for spleen in Fig. 2L as was done for blood in Fig. S8.

Response: Thank you for your comments. We added the full data and statistics in the revised supplementary files (Supplementary Figure 13A-13B, also shown as appendix Figure 16).

Appendix Figure 16. Effector memory CD4⁺ T cells (A) and CD8⁺ T cells (B) in spleen were detected 1 weeks after the third immunization.

Data are shown as mean ± SEM (n=5). Statistical analysis was performed using one-way ANOVA.

19. Line 316 (PDF) indicates that “The combined antigens in NP-cocktail did not impact each other’s immunogenicity and titers;” however, this comparison is not possible because it is assumed that the individual NP experiments were performed on different days than the cocktail NP experiments. To make this claim, the authors had had to either perform all immunizations for comparison and associated assays at the same time, or if this wasn’t possible, to perform the assays on the same day, with the same conditions and same reagents.

Response: Thank you for your comments. To clarify the experimental procedure, we performed the assays on the same day with same conditions and same reagents. We explained this in the revised manuscript as follows (Line 1032-1035): ‘The detection of binding titers and neutralizing titers of sera from vaccines containing a single antigen and cocktail vaccines immunized mice or rabbits were performed with the

same procedure, on the same day, under the same conditions and with the same reagents.’

20. Line 319 (PDF) indicates that “NP-cocktail elicited higher and longer-lasting total IgG titers [...] than the Free- cocktail and AI-cocktail.” However, as for Fig. 2, this statement is not well-supported in regards to response durability as the overall antibody decay trend is the same for all groups.

Response: Thank you for your comments. We rephrased this sentence in the revised manuscript as follows (Line 412-414): ‘NP-cocktail elicited higher total IgG titers against gHgL, gB and gp42 than the Free-cocktail and AI-cocktail (Figure 4A-4C and S18B).’

21. Line 321 (PDF) indicates that “NP-cocktail was much more efficient than Free-cocktail and AI-cocktail” in regards to induction of cellular immune responses. However, no official statistical comparison is provided to support this claim.

Response: Thank you for your comments. We added the full data and statistics in the revised supplementary files (Supplementary Figure S18C-S18D, also shown as appendix Figure 17).

Appendix Figure 17. Antigen-specific CD4⁺ (A) and CD8⁺ (B) T cell responses on day 35 (5 weeks) in the spleen of mice immunized with the indicated vaccines.

Antigen specific T cells were measured by intracellular cytokine staining assay after restimulation with gHgL, gB and gp42 *in vitro* (n=5).

22. Line 330 indicates that “NP-cocktail vaccination induced more memory B cells and effector memory T cells in the spleen and PBMCs;” however this is not true for CD4⁺ T cells in PBMCs. Although it is implied, the authors should also finish the sentence to indicate what is being compared.

Response: Thank you for your comments. We rephrased this sentence in the revised manuscript as follows (Line 438-441): ‘Moreover, NP-cocktail vaccination induced more memory B cells and effector memory T cells in the spleen (Figure S23A-S23C) and more memory B cells and effector-memory CD8⁺ T cells were also found in peripheral blood (Figure S23D-S23F).’

23. All neutralization claims starting in Line 332 cannot be appropriately assessed if no infectivity information is provided for the EBV batches used for each assay.

Response: Thank you for your comments. The same batch of the CNE2-EBV-GFP was used in all B cell neutralizing assay and the same batch of the Akata-EBV-GFP was used in all epithelial cell neutralizing assay (Shown above as Appendix Figure 4). We added the methods and data about the titration of virus in the revised manuscript (Line 1079-1092). Also see answer to point 1 from reviewer #2.

24. Line 432 indicates that “NP-cocktail induced the highest B cell and epithelial neutralizing titers;” however, there was no statistical difference between NP-cocktail and NP-gHgL in epithelial cells.

Response: Thank you for your comments. We rephrased this sentence in the revised manuscript as follows (Line 564-570): ‘After the third immunization, the B cell infection and epithelial cell infection neutralizing titers of sera induced by NP-cocktail is 2.86 (log₁₀) and 3.45 (log₁₀), respectively (Figure 5E-5F and S26H-S26I). Individually, NP-gHgL, NP-gB and NP-gp42 vaccination induced similar B cell neutralizing antibody titers in rabbits (2.34 (log₁₀), 2.39 (log₁₀) and 2.56 (log₁₀), respectively) (Figure 5E; Table S3). Among the different vaccine formulations, NP-cocktail induced the highest B cell neutralizing titers (Figure 5E).’

25. There are several major concerns regarding the humanized mouse model described.

- There should be rigor in confirming whether the PBMCs used for mouse reconstitution were EBV-infected or not- was ELISA or qPCR used to determine this?

Response: Thank you for your comments. We performed ELISA and qPCR assays to determine that PBMCs used to humanized mice reconstitution were EBV infected with undetectable EBV copy numbers (supplementary Figure 29A-29E, also shown as appendix Figure 18). The data showed that donors for humanized mice reconstitution is EBV infected and the EBV DNA copy numbers was below the detection limit (supplementary Figure 29A-29E, also shown as appendix Figure 18). In healthy carriers, EBV copy number is 1-50 copies in 1 million leukocytes^{19, 20}. We rephrased the description in the revised manuscript as follows (Line 609-611): ‘Therefore, we developed a humanized mouse model based on SCID mice engrafted with a lower amount of PBMCs from healthy EBV seropositive donors with undetectable EBV copy numbers (Figure S29A-S29E)^{19, 20}.’ We also updated the Materials and Methods (Line 1226-1231).

Appendix Figure 18. Anti-EBV glycoprotein titers and EBV DNA copy numbers of PBMCs used in humanized mice reconstitution.

(A-D) Anti-gp350 (A), anti-gHgL (B), anti-gB (C) and anti-gp42 (D) titers of sera

from Donor1 and Donor2 determined by ELISA. Data are shown as mean \pm SEM of two independent replicates.

(E) EBV DNA copy numbers of PBMCs isolated from Donor1 and Donor2. Data are shown as mean \pm SEM of three independent replicates. The dashed line indicates the detection limit.

- How many donors were used for reconstitution? Were the donor PBMCs pooled or were distinct individual PBMCs used for individual animals? If they were not pooled, how is individual donor variability accounted for?

Response: Thank you for your comments. We used PBMCs from two donors for the humanized mice reconstitution. PBMCs from Donor 1 were used to reconstitute 53 SCID-beige mice, which received 1 mg IgG antibodies purified from immunized rabbit (Supplementary Table 4, also shown as appendix Table 2). PBMCs from Donor 2 were used to reconstitute 63 SCID-beige mice, which were injected with sera collected at different times from various immunized mice (Supplementary Table 4, also shown as appendix Table 2). That is to say, PBMCs from one donor were used to reconstitute all mice used in passive protection assays by rabbit IgGs and another donor was used in passive immunization using mouse sera. Besides, each experiment contained the mock group without EBV challenge and humanized mice administrated with sera or IgGs from animals inoculated with PBS were used as negative control. Thus, there is no concern about donor variability within each experiment. We also clarified the use of donors in the revised supplementary file (Line 354 and Line 448).

Appendix Table 2. PBMCs from different donors for humanized mice reconstitution.

Donor	Numbers of reconstituted SCID-beige mice	Injection agents
Donor1	8	1 mg IgG from NP-gHgL immunized rabbit
	8	1 mg IgG from NP-gB immunized rabbit
	8	1 mg IgG from NP-gp42 immunized rabbit
	8	1 mg IgG from NP-cocktail immunized rabbit
	8	1 mg IgG from Free-cocktail immunized rabbit

	8	1 mg IgG from PBS administrated rabbit
	5	Mock1
Donor2	5	Sera (week5) from NP-gHgL immunized mice
	5	Sera (week5) from NP-gB immunized mice
	5	Sera (week5) from NP-gp42 immunized mice
	5	Sera (week5) from NP-cocktail immunized mice
	5	Sera (week5) from Free-cocktail immunized mice
	5	Sera (week5) from PBS immunized mice
	5	Sera (week50) from NP-gHgL immunized mice
	5	Sera (week50) from NP-gB immunized mice
	5	Sera (week50) from NP-gp42 immunized mice
	5	Sera (week50) from NP-cocktail immunized mice
	5	Sera (week50) from Free-cocktail immunized mice
	5	Sera (week50) from PBS immunized mice
	3	Mock2

➤ Where the PBMCs tested *in vitro* for Akata-EBV-GFP infection susceptibility?

Response: Thank you for your comments. PBMCs used for humanized mice reconstitution were susceptible to the Akata-EBV produced from CNE2-EBV cell *in vitro* (Supplementary Figure 29F, also shown as appendix Figure 19). We added the results in the revised manuscript as follows (Line 611-613): ‘PBMCs used for humanized mice reconstitution were susceptible to EBV infection *in vitro* (Figure S29F).’. We also updated the Materials and Methods (Line 1232-1237).

Appendix Figure 19. *In vitro* EBV infection of PBMCs used for humanized mice reconstitution. 1×10^5 PBMCs from each donor or Akata cells (EBV negative) were seeded in 96-well plate and serial diluted CNE2-EBV-GFP were added to the well. After 48 h incubation, the percentage of GFP positive cells were detected by flow cytometry. Cells without incubation with virus were used as negative control.

- Perhaps this reviewer is not familiar with this mouse model, but the % reconstitution illustrated in Fig. S19 seems rather low; can the authors comment if these are normal levels for this mouse model?

Response: We agreed with you that the % reconstitution appears low in the PBMCs of reconstituted mice 4 weeks post reconstitution (Figure S31). It is reported that when SCID mice were administrated with a high number (5×10^7) human peripheral blood leukocytes, ~80% human CD45⁺ cells were detected in the peritoneal lavage fluids and these mice developed tumor derived from the administrated human B cells²¹. This high dose also yields ~80% human B cells and ~20% human T cells in the spleen of reconstituted SCID mice 8 weeks post administration²². In this study, we used a relatively lower number of PBMCs (1.5×10^7) to reconstitute SCID-beige mice in order to avoid spontaneous tumor formation. Consequently, we detected fewer human immune cells in PBMCs, peritoneal lavage fluids, lymph nodes and spleen of reconstituted mice (Figure S31 and S33). Although the % reconstitution is low, increased EBV DNA copy numbers were detected in PBMCs, lymph nodes, spleens and tumors in mice from the control group (PBS-IgG) (Figure 6I-6J and 7D). Hence, EBV could infect the reconstituted human B cells and eventually cause EBV associated lymphoma in the SCID-beige mice reconstituted with 1.5×10^7 human peripheral blood leukocytes.

- Since full PBMCs are used for reconstitution, can the authors comment on the potential impact of introducing previously existing EBV-specific T cells into this infection system on the experimental results?

Response: Thank you for your comments. It is reported that the level of EBV-specific T cells is very low in PBMCs of healthy EBV carriers^{23, 24, 25}. To avoid any unwanted effect of human EBV-specific T cells, we used a lethal dose of EBV in the passive protection assay (25,000 GRU), which caused the control group (PBS-IgG) to develop EBV-associated lymphoma^{15, 17}. Hence, we speculated that the previously existing EBV-specific T cells may not effectively kill infected B cells under these conditions.

- Administration of 1mg of rabbit IgG per mouse is quite high compared to other recent passive immunization experiments in EBV humanized mouse models. Can the authors comment on how they arrived to this amount?

Response: Thank you for your comments. The administration of 400 ~ 500 µg of EBV-specific monoclonal antibodies is commonly used in passive protection assays (Appendix Table 3). However, the dose of IgGs purified from immunized animals ranges from 400 µg to 10 mg per mouse in passive protection studies and the inoculation times ranges from 1 to 3 times (Appendix Table 3). Hence, administration one time of 1 mg of rabbit total IgG per mouse is not singularly high.

Moreover, we also determined the protective effect of lower (400 µg per mouse) and higher (2 mg per mouse) doses of total IgG purified from NP-cocktail and Free-cocktail immunized rabbits (Appendix Figure 20). The results showed that at the dose of 2 mg/mouse, all humanized mice treated with IgG purified from NP-cocktail and Free-cocktail immunized rabbits survived until the end of the experiments (Appendix Figure 20G). One mouse treated with IgG purified from Free-cocktail immunized rabbit showed slightly increased EBV copy numbers in the PBMCs (Appendix Figure 20H). When inoculated with 400 µg IgG, an increase in EBV copy numbers was detectable even in humanized mice treated with IgG purified from NP-cocktail immunized mice, and one mouse died in this group ((Appendix Figure 20C and 20D). Based on these observations, we believe that 1 mg is an appropriate dose for the passive protection assays.

Appendix Table 3. Dose, agents and route of administration (ROA) of passive protection studies.

Dose	Agents	ROA	Virus	Ref
500 µg/mouse	Total IgG	i.p.	EBV	3
2 mg or 10 mg/mouse	Total IgG	i.p.	EBV	26
20 mg/kg body weight of mouse (~400 µg/mouse)	Total IgG	i.p.	EBV	4
500 µg/mouse	EBV-specific mAb	i.v.	EBV	27

400 µg/mouse	EBV-specific mAb	i.p.	EBV	14, 17, 28
20 mg/kg body weight of mouse (~400 µg/mouse)	EBV-specific mAb	i.p.	EBV	15, 29

Appendix Figure 20. Different IgG doses passive protection efficiency in humanized mouse model.

(A) Schematic illustration of humanized mice reconstitution and passive protection against EBV lethal challenge. SCID-beige mice were irradiated and administrated with clodronate liposomes before being reconstituted with PBMCs from healthy donors (EBV seropositive). Four weeks later, humanized mice were inoculated with IgG (400 µg per mouse) purified from sera of rabbits immunized with different vaccines. 25,000 GRU (green Raji units) Akata-EBV was injected 24 hours after IgG administration. i.v., intravenously; i.p., intraperitoneally. The figure was created from

Biorender.com. Humanized mice in the mock group were not challenged with EBV.

(B) Body weight of mice post-EBV challenge. Data are shown as mean \pm SEM.

Statistical analysis was performed using one-way ANOVA.

(C) Survival plot of humanized mice after EBV challenge.

(D) EBV DNA copy numbers in peripheral blood of each group. Each line represents an individual mouse and the dashed line indicates the detection limit.

(E) Schematic illustration of humanized mice reconstitution and passive protection against EBV lethal challenge. SCID-beige mice were irradiated and administrated with clodronate liposomes before being reconstituted with PBMCs from healthy donors (EBV seropositive). Four weeks later, humanized mice were inoculated with IgG (2 mg per mouse) purified from sera of rabbits immunized with different vaccines. 25,000 GRU (green Raji units) Akata-EBV was injected 24 hours after IgG administration. i.v., intravenously; i.p., intraperitoneally. The figure was created from Biorender.com. Humanized mice in the mock group were not challenged with EBV.

(F) Body weight of mice post-EBV challenge. Data are shown as mean \pm SEM.

Statistical analysis was performed using one-way ANOVA.

(G) Survival plot of humanized mice after EBV challenge.

(H) EBV DNA copy numbers in peripheral blood of each group. Each line represents an individual mouse and the dashed line indicates the detection limit.

The color of the asterisks denotes the group with which there is a significant difference (significant difference: * $P \leq 0.0332$; ** $P \leq 0.0021$; *** $P \leq 0.0002$; ns, no significant difference).

Minor comments:

1. Figures contain too many panels, and as a consequence the font size in the panels is very small and hard to read. We understand this might be the result of journal restrictions on Figure numbers, but we recommend the authors re-evaluate their figures and transfer some panels to Supplementary so that main figure panels can be bigger and more legible. The choice of colors also make it difficult to discern between different treatment groups in immunogenicity studies; in this case perhaps the authors

can make use of different symbols to represent different groups to aid the reader?

Response: Thank you for your comments. We adjusted the main figure panels according to your suggestions.

2. On that same note, the font size and choice of colors make it very hard to read and understand the statistics in immunogenicity studies; would it be possible for the authors to use traditional lines to indicate group differences if the panels cannot be made bigger?

Response: Thank you for your comments. We enlarged the main figure panels' size and labeled the p values with traditional lines in the figures to better show the statistics in immunogenicity studies according to your suggestions.

3. It would be helpful for the reader if p-values are integrated in parenthesis when discussing statistical differences between groups within the text.

Response: Thank you for your comments. We added the p values in the figures in the revised manuscript.

4. Please include animal numbers and sex for every animal experiment both in the Methods and in the Figure legends.

Response: Thank you for your comments. We added the animal numbers and sex for all the animal experiments in the Method and Figure Legends.

5. BALB/c mice are mentioned in the Methods, but they are not mentioned either in the text or Supplementary table 1; please correct/clarify as necessary.

Response: We are sorry for the mistake. We deleted the BALB/c mice in the methods.

6. CHO-K1 cells are used in cell-cell fusion assays, but are not listed in the Cell lines section of the Methods; please add as necessary.

Response: Thank you for your comments. We added the description of CHO-K1 cell

culture in the methods (Line 872-873).

7. In the Neutralization assay section in the Methods, please indicate/clarify whether all neutralizations (human, mouse, rabbit) were performed the same way, or if there were any species/experiment-specific changes.

Response: Thank you for your comments. We added the explanation in the revised manuscript as follows (Line 1116-1117): ‘All neutralizing assays of human, mouse and rabbit sera were performed in the same way.’

8. Please provide demographic information for the blood donors from Fig. 1, if available.

Response: Thank you for your comments. We added the demographic information in the supplementary file (Supplementary Table 1, also shown as Appendix Table 4).

Appendix Table 4. Demographic information of the blood donors for antibody depletion assay.

No.	Gender	Age
190001	F	54
190002	M	42
190003	M	52
190004	F	22
190005	F	24
190006	F	25
190007	F	29
190008	M	28
190009	M	46
190010	M	23
190011	F	21
190012	F	22
190013	M	50
190014	F	21
190015	F	26
190016	F	23

190017	F	26
190018	M	22
190019	F	22
190020	F	25
190021	F	26
190022	F	24
190023	M	30
190024	F	21
190025	M	27
190026	F	22
190027	M	28
190028	M	22
190029	F	45
190030	F	24
190031	F	26
190032	M	30

F: Female; M: Male

9. Please address on the discussion how Fig. 1 results compare to those recently published by Bu et al in Immunity 2019 May 21;50(5):1305-1316.e6 (doi: 10.1016/j.immuni.2019.03.010.).

Response: Thank you for your comments. We discussed this study in the revised manuscript as follows (Line 190-194) ‘Consistent with a previous study by Bu. *et.al*, our data showed that, in sera from healthy EBV carriers, gp350-specific antibodies are the major contributor (~37.72%) to neutralize EBV infection of B cells while gHgL-specific antibodies are the most important antibodies (~45.13%) to neutralize EBV infection of epithelial cells ².’ and (Line 198-204): ‘According to our data and the previous studies, depletion of gHgL-specific, gB-specific and gp42-specific antibodies from sera reduced their ability to neutralize EBV infection *in vitro* to various degrees ^{2, 17}. Besides, multiple nAbs targeting gHgL, gB and gp42 have been isolated and characterized ¹¹. Thus, gHgL, gB and gp42 are potential antigens for the development of efficient EBV vaccines.’

10. Please provide information regarding the source of the gp350 protein used in ELISA assays in the Methods, as was provided for gHgL, gB and gp42.

Response: Thank you for your comments. We added the information of gp350 protein in the methods (Line 846-854 and 875-883).

11. The Y-axis in Fig. 1A and F are labeled as “Neutralizing titers,” however, ID50 is provided; please modify to indicate ID50, as was done for all other neutralization panels in the manuscript for consistency.

Response: Thank you for your comments. We changed the name of Y-axis to ‘ID50 (Log10)’ in the revised Figure 1B.

12. Line 132 (PDF) refers to Fig. 1A and F as IgG titrations, but these are neutralization results; please correct as necessary.

Response: Thank you for your comments. We changed the results in the revised manuscript.

13. Y-axis in Fig. 2F should be re-labeled to “IgA level,” since no titration was performed.

Response: Thank you for your comments. We changed the name of Y-axis to ‘IgA level’ in the Figure 2F.

14. It would be more helpful for the reader if Fig. 3A, B and C go to supplementary (Fig. S11), and instead, content in Fig. S11 comes to the main figure.

Response: Thank you for your comments. We moved Fig 3A-3C to the supplementary Figure 16A-16C and moved the supplementary Figure S11 to the Figure 3A-3C according to your suggestions.

15. Lines 283-285 (PDF): “[...] were encapsulated with two adjuvants (NP-) and their mixtures as controls (-Free”, please re-phrase for clarity.

Response: Thank you for your comments. We rephrased the sentence in the revised

manuscript as follows (Line 365-369): ‘To evaluate the transportation of NP-gHgL, NP-gB and NP-gp42 to LNs, 5 µg Cy5-labeled gHgL, gB or gp42 were encapsulated with both adjuvants to prepare nanovaccines, NP-gHgL, NP-gB or NP-gp42, respectively. The simple mixture of Cy5-labeled gHgL, gB or gp42 and the two adjuvants were named Free-gHgL, Free-gB or Free-gp42, respectively.’

16. Line 287 (PDF): can axillary LNs be considered “distal” in this case, given that in mice the inguinal and axillary LNs are directly connected?

Response: Thank you for your comments. The “distal” means the axillary LNs are further away from the vaccine injection sites compared with inguinal LNs. We added the explanation in the Figure legends (Line 1696-1697).

17. Line 288 (PDF): “much fewer of the free-form antigens were detected” cannot be appropriately used since the antigens were not directly quantified instead please use “less fluorescence associated with free-form antigens was detected.”

Response: Thank you for your comments. We rephrased the sentence in the revised manuscript as follows (Line 374-375): ‘By comparison, less fluorescence associated with free-form antigens was detected in LNs (Figure 3D).’

18. Line 302 (PDF) uses the term “potent” to describe GC formation; can the authors define this?

Response: Thank you for your comments. We rephrased the sentence in the revised manuscript as follows (Line 390-392): ‘After three immunizations, NP-gHgL, NP-gB and NP-gp42 more efficiently induced GCs responses than other vaccine formulations (Figure 3G).’

19. Fig. 4A, particularly the pie chart, does not contribute to the overall results and it might be better to remove it to make more space available for the rest of the panels.

Response: Thank you for your comments. We moved the Fig. 4A to the supplementary Figure S18A according to your suggestions.

20. Units should be defined for Fig. 4F and G in the Figure legend.

Response: Thank you for your comments. We rephrased the sentence in the revised manuscript as follows (Line 1747): ‘RLU, relative luminescence units.’

21. For the chimeric 6H2 antibody, please provide data showing that chimerization abrogated mouse reactivity, thus not interfering with the competitive ELISA.

Response: Thank you for your comments. We tested the reactivity of the chimeric 6H2 antibody with anti-mouse IgG antibodies according to your suggestions. The chimeric 6H2 (c6H2) did not react with anti-mouse HRP-conjugated secondary antibody, thus it would not interfere with the results of the competitive ELISA (Supplementary Figure 25, also shown as Appendix Figure 21). We added the results in the revised manuscript as follows (Line 485-487): ‘The chimeric 6H2 antibody did not react with anti-mouse HRP-conjugated secondary antibody, thus it would not interfere with the results of the competitive ELISA (Figure S25)’. We also added the related method in the Materials and Methods (Line 1056-1065).

Appendix Figure 21. Reactivity of the c6H2 with anti-human or anti-mouse HRP conjugated secondary antibody.

22. Fig. S15 would benefit from having more descriptive titles for each panel; for example, panel A can be renamed from “spleen” to “Memory B cells in spleen.”

Response: Thank you for your comments. We renamed the titles for each panel in Fig. S23 according to your suggestions (Supplementary Figure 23, also shown as appendix Figure 22).

Appendix Figure 22. The nanoparticle cocktail vaccine enhances the production of memory T cells and B cells.

Splenocytes and PBMCs were analyzed by flow cytometry according to the strategy described in figure S7.

(A-C) Memory B cells (A), effector memory CD4⁺ T cells (B) and CD8⁺ T cells (C) in spleen were detected 1 week after the third immunization of C57BL/6J mice (day 35).

(D-F) Memory B cells (D), effector memory CD4⁺ T cells (E) and CD8⁺ T cells (F) in peripheral blood were detected 8 weeks after the third immunization of C57BL/6J mice (week 12).

Data are shown as mean ± SEM (n=5). Statistical analysis was performed using one-way ANOVA.

23. In Fig. 5D-F, why is the pre-immune/pre-bleed value not provided? Why is this not represented as a kinetic analysis as was done for mouse studies?

Response: Thank you for your comments. We added the pre-immune value in the

figures and adjusted these figures to a kinetic analysis (Supplementary Figure 26A-26C, also shown as appendix Figure 13).

Appendix Figure 23. IgG titers of sera collected on day 14, 28 and 42 from rabbits immunized with different vaccine formulations.

(A) Anti-gHgL IgG titers elicited by NP-gHgL, NP-cocktail, Free-cocktail and PBS, respectively.

(B) Anti-gB IgG titers elicited by NP-gB, NP-cocktail, Free-cocktail and PBS, respectively.

(C) Anti-gp42 IgG titers elicited by NP-gp42, NP-cocktail, Free-cocktail and PBS, respectively.

24. In Fig. 5G, why are statistics provided for the gp42 group in epithelial cells, but no datapoints?

Response: Thank you for your comments. We are sorry for the mistake. We corrected the Figure 5F in the revised manuscript.

25. Line 438 uses the term “high quality” serum; can the authors define this?

Response: Thank you for your comments. The ‘high quality’ represents the sera from nanovaccines immunized rabbits more efficiently neutralized EBV infection *in vitro* than other vaccine formulations. We revised this sentence in the revised manuscript as follows (Line 572-574): ‘These data indicate that nanoparticle encapsulation is beneficial to generate serum antibodies that more efficiently neutralized EBV infection *in vitro* in rabbits than the Free-group.’ For clarity, we revised this

terminology throughout the manuscript.

26. There seems to be quite a high background for both hCD20 and EBER staining in Fig. 7.

Response: Thank you for your comment. We performed the hCD20 and EBER staining again to clarify the data (Figure 7A and 7H, also shown as appendix Figure 24 and 25).

Appendix Figure 24. Representative macroscopic spleen and spleen sections stained by hematoxylin and eosin (HE), immunostained for hCD20⁺ and *in situ* hybridized for EBV-encoded RNA (EBER). Each image is representative of the experimental group (Scale bar=50 µm).

Appendix Figure 25. Representative macroscopic spleen and spleen sections stained by hematoxylin and eosin (HE), immunostained for hCD20⁺ and *in situ* hybridized for EBV-encoded RNA (EBER). Each image is representative of the experimental group (Scale bar=50 μ m).

Reviewer #3 (Remarks to the Author):

This manuscript designed a nano-vaccine cocktail composed of EBV core glycoprotein antigens gHgL, gB, or gp42, along with two adjuvants, CpG and MPLA. The authors elucidated that these nanovaccines can induce effective and long-lasting humoral and cellular immune responses by promoting APC internalization, targeting LN transportation, and enhancing GC formation, without any noticeable side effects. The nano-vaccines induced effective neutralizing and fusion-inhibitory antibodies by broadening the antigen spectrum. The IgG produced by these nano-vaccine cocktails provided 100% protection against viremia and EBV-associated lymphomas in humanized mice. This work is comprehensive, though there are a number of concerns that need to be addressed.

Response: We appreciate your valuable comments and suggestions about our manuscript. Your suggestions will greatly improve our work. We address all the individual comments below.

1. The correlation of EBV infection with cancer development vs. anti-tumor immunity (e.g., Nature 2021; 590(7844):157-162; Clin Cancer Res 2022; 28(20):4363-4369) is an interesting/debatable topic, which may need to be discussed in the manuscript.

Response: Thank you for your comments. We discussed the correlation in the revised manuscript as follows (Line 67-69): ‘Besides, EBV-infected B cells induce T cell responses against EBV-associated antigens as well as non-EBV tumor-associated antigens, which provides new strategies for future cancer therapy^{30, 31}.’

2. In line 109 of the manuscript, it is mentioned that “All three nanovaccines with single antigens (gHgL, gB or gp42) induced potent and durable adaptive immune responses in mice attributed to efficient transportation to LNs, enhanced antigen internalization and maturation of dendritic cells, and robust germinal center (GC) responses”. However, the authors found that “IgG induced by a cocktail vaccine containing three nanovaccines conferred 100% protection from lethal EBV challenge in humanized mice, whereas IgG elicited by individual NP-gHgL, NP-gB, and NP-

gp42 did not". What is the synergistic mechanism of these three nanovaccines? The innovation of the cocktail vaccine strategy may also be further discussed. It is not clear either why the authors used a mixture of three NPs each containing one antigen instead of one NP containing three antigens.

Response: Thank you for your comments. gHgL, gB and gp42 play different key roles during EBV infection process ¹¹. Thus, compared to three individual nanovaccines, NP-cocktail immunization could generate various antibodies targeting different steps of EBV infection and more efficiently prevent humanized mice against lethal EBV challenge *in vivo*. We agree that because of the already robust effectiveness of each individual NP formulation, it is hard to determine whether the cocktail allows true synergy or provides simply additive effects of its components.

We also tried to encapsulate all three proteins simultaneously into one nanoparticle. However, to obtain comparable amounts of antigens, the size of the combined nanoparticle was greater than the individual nanovaccines and displayed a high PDI, indicating that the nanoparticles are not homogenous. Therefore, we preferred to encapsulate the three antigens separately and mix the three nanovaccines to develop NP-cocktail.

We discussed the innovation of the cocktail vaccine strategy in the revised manuscript as follows (Line 785-793): 'Indeed, IgG induced by NP-cocktail in rabbits provided 100% protection to humanized mice against viremia and EBV-associated lymphoma. This efficacy was better than that provided by NP comprising individual antigens or by a cocktail of non-encapsulated antigens and adjuvants (Free-cocktail). NP-cocktail immunization generated antibodies targeting different steps of EBV infection. Hence, the combination of antigens and the nanoparticle-based formulation are two advantages to consider in the development of effective anti-EBV vaccines. Combination of different antigens involved infection process is a trend for further improvement of EBV prophylactic vaccine design.'

3. The authors may explain why the ratio 1:1:1 of the cocktail nano-vaccines was used.

Response: Thank you for your comments. We used an indirect observation to select this 1:1:1 combination of antigens. We first pooled sera (week 5) of 5 mice immunized with individual NP-vaccines. Then we mixed the above sera mixtures at different ratio *in vitro* (Appendix Figure 26) and tested the neutralizing ability of these combinations to determine the optimum mixing ratio of the three sera. We found that for both B cell infection neutralization and epithelial cell neutralization, sera mixture showed the highest ID50 at the ratio 1:1:1 (Appendix Figure 26). Therefore, we further tested the immunogenicity of immunization of combination of three nanovaccines at 1:1:1 ratio.

Appendix Figure 26. Neutralization of CNE2-EBV infection of Akata B cells (A) and Akata-EBV infection of HNE1 epithelial cells (B) by sera mixture mixed at different ratio. 50 μ l sera of each 5 C57BL/6J mice collected at week 5 from NP-gHgL, NP-gB or NP-gp42 immunization respectively were mixed. Then the above sera mixture from NP-gHgL, NP-gB and NP-gp42 immunization were mixed at different ratio *in vitro*. Data points are shown as the mean \pm SEM of two independent replicates.

4. The manuscript mentioned “high-quality neutralizing antibodies”. A more precise definition for “high-quality” is needed.

Response: Thank you for your comments. The ‘high quality’ represents the sera from nanovaccines immunized rabbits more efficiently neutralized EBV infection *in vitro* than other vaccine formulations. We changed the ‘high-quality neutralizing antibodies’ to ‘high level of neutralizing antibodies’ throughout the revised manuscript.

5. What is the rationale of co-delivery of CpG and MPLA as adjuvants? The author found that the nano-vaccine based on CpG and MPLA is superior to the Alum-vaccine; what is the reason behind that? Do CpG and MPLA have any unique synergistic effects with gHgL, gB, or gp42-related immune response?

Response: Thank you for your comments. Except for humoral immune responses, the induction of a T cell immune response has proven to be important for the protective effect of vaccines, especially vaccines against herpesviruses^{32,33}. Hence, we also tried to elicit potent T cell responses through application of proper adjuvants. CpG and MPLA are the extensively studied Toll-like receptor agonists (TLRa), which have been used in commercial vaccines³⁴. MPLA is reported to induce balanced Th1 and Th2 responses while CpG elicits potent Th1 responses^{35,36}. It has been reported that the combination of MPLA and CpG induced potent antigen-specific T cell responses and promoted Th1-biased IgG responses^{37,38}. For these reasons, we selected to co-deliver CpG and MPLA as adjuvants.

It is reported that Alum mediates its adjuvant properties through ‘depot effect’ to slow release antigens at the injection sites and induced Th2 biased immune responses³⁴. CpG and MPLA, as TLRa linking the innate immunity with adaptive immunity, efficiently induce antigen-specific humoral and cellular immune responses³⁹. Overall, our data suggested that combining CpG and MPLA in nanovaccines targeted delivered to lymph nodes, which further enhanced Tfh cell response, GC formation and potent adaptive immune responses (Figure 2 and Figure 3).

We do not have evidence to show that CpG and MPLA have unique synergistic effects in the induction of gHgL, gB or gp42-related immune responses. Direct assessment of the contributions of CpG and MPLA will be relevant in the future development of these NP vaccines but is beyond the scope of the current study.

6. DOTAP is a cationic lipid and could also activate innate immune responses, beyond its role for complexing with proteins and CpG. This needs to be explored.

Response: Thank you for your comments. It is reported that nanoparticles composed

of DOTAP could activate innate immune responses⁴⁰. Since DOTAP is a liposoluble substance, which is dissolved in absolute ethyl alcohol. The DOTAP solution is toxic for cells. Instead, we co-incubated BMDCs with empty nanoparticles to explore the innate immune stimulation effect of nanoparticles without adjuvant. Consistent with previous studies, we found that empty nanoparticles stimulated BMDCs to up-regulate the expression of CD80 and CD86 (Supplementary Figure 16D, also shown as appendix figure 27). Since empty nanoparticles could also activate innate immune responses, we added the data in the revised manuscript as follows (Line 352-354): ‘This effect is partially due to the nanoparticles itself, which also induced the up-regulation of CD80 and CD86 in BMDCs without adjuvants (Figure S16D)⁴⁰.’

Appendix Figure 27. Expression detection of CD80 and CD86 on BMDCs incubated for 24 h with empty nanoparticles (eNPs). Data points are shown as the mean \pm SEM (n=3). Statistical analysis was performed using unpaired Welch’s t test.

7. In Figures 1C and 1J, the results of ‘r’ value is 0.2997 and 0.2227 respectively. Could this reflect the importance of gHgL and gp42?

Response: Thank you for your comments. In the original Figure 1J, the p value was 0.1023, which is higher than 0.05. Hence, there is no correlation between anti-gp42 IgG titers and the sera neutralizing titers. In the original Figure 1C, the p values was 0.0273 (<0.05), indicating a correlation between anti-gHgL IgG titers and the sera neutralizing titers. To thoroughly address the neutralizing contribution of each glycoprotein antibody in sera from health EBV carriers, we performed glycoprotein-specific antibody depletion assays (Figure 1A-1C and S2, also shown as above appendix Figure 12). This is described in the revised manuscript (Line 167-204). Also see our response to point 9 from Reviewer#2.

8. Figure 1O indicates that the loading efficiency of NP exceeded 90%, and even the loading of the adjuvant reached 99.9%. The authors should provide detailed description of preparation, purification, detection/analysis (for antigens and adjuvants), and calculation.

Response: Thank you for your remark. We provided the details of the preparation, purification, detection and calculation in the revised manuscript (Line 926-950).

9. The differences in immunofluorescence staining in Figure 3E and F are not very clear.

Response: Thank you for your comments. We added a plot to compare the mean fluorescence intensity of antigens in Figure 3E in supplementary Figure 16E (also shown as Appendix Figure 28A). To help with visualization, we also circled the germinal center region in Figure 3G (also shown as Appendix Figure 28B).

A

B

Appendix Figure 28. Antigen distribution and germinal center formation of mice inguinal LNs after immunization with various vaccines.

(A) Mean fluorescence intensity of antigens in the draining lymph nodes 6 h post-immunization. Data points are shown as the mean \pm SEM (n=3). Statistical analysis was performed using unpaired welch's t test.

(B) Germinal center (circled) formation in inguinal lymph nodes of C57BL/6J mice immunized with different vaccine formulations. LNs were harvested on day 35 after the first of three immunizations and stained with Alexa Fluor 594-IgD, Alexa Fluor 488-B220 and Alexa Fluor 647-GL7 (Scale bar = 400 μ m).

10. In Figure 5, the immunization method for rabbits is “intramuscularly”, while in Figure 2, the immunization method for C57BL/6J mice is “subcutaneously”. Why

were the same methods not adopted? When comparing intramuscular immunization in the rabbit model to subcutaneous immunization in the mice model, is there a difference in the biodistribution of the nano-vaccine?

Response: Thank you for your comments. Intramuscular and subcutaneous injection are two common ways for vaccination and both ways could generate potent immune responses. It has been reported that lymph node targeted delivery can be achieved by nanoparticle vaccines following intramuscular, subcutaneous or intradermal injection⁴¹. Moreover, intramuscular and subcutaneous injection could generate similar antigen specific antibody responses in mice and human^{42,43}. Hence, we believed that there is no major difference in the biodistribution of the nanovaccines following intramuscular or subcutaneous injection in mice or rabbits. We performed intramuscular injection in rabbits since major human vaccines are injected intramuscularly. We also confirmed the biodistribution of nanovaccines in lymph nodes of mice after intramuscular injection (Appendix Figure 29). We confirmed that nanovaccines were also delivered to murine lymph nodes after intramuscular injection (Appendix Figure 29).

Appendix Figure 29. Lymph nodes (LNs) images of mice immunized with vaccines containing Cy5-labeled antigens (n=3). C57BL/6J mice were injected intramuscularly with 10 μg nanoparticle (NP) forms of Cy5-labeled gHgL, gB or gp42. Axillary and inguinal LNs were harvested after 6 h and 24 h, and Cy5 fluorescence was measured using the IVIS optical imaging system.

11. Regarding the safety evaluation of the nano-vaccine, Figure S10 and Figure S16 only demonstrated that “tissue damage or inflammation was observed in the heart, liver, spleen, lung, and kidney tissues”, lacking the evaluation of biochemical functional indicators.

Response: We appreciate your constructive suggestions. To further address the safety of our Nanovaccines, we tested the total cholesterol (TC), alanine transaminase (ALT), total bilirubin (TBIL), triglycerides (TG), aspartate aminotransferase (AST), alkaline phosphatase (ALP) and UREA levels in sera from mice or rabbits after the third immunization. The serum concentrations of all these biochemical functional indicators were similar between the nanovaccines immunized mice or rabbit and the controls (PBS administrated mice or rabbits) (Supplementary Figure 15, also shown as Appendix Figure 30 and 31). Hence, nanovaccines were safe in mice and rabbits, based on histological and biochemical evaluations. We added the results in the revised manuscript as follows (Line 316-320): ‘The total cholesterol (TC), alanine transaminase (ALT), total bilirubin (TBIL), triglycerides (TG), aspartate aminotransferase (AST), alkaline phosphatase (ALP) and UREA levels of sera from nanovaccine immunized mice collected at week 5 were similar to the control PBS group, which further confirmed the safety of nanovaccines (Figure S15B-S15H).’ and Line576-580): ‘Furthermore, no tissue damage or inflammation was observed in the heart, liver, spleen, lung and kidney tissues of immunized rabbits (Figure S27A). Besides, the level of biochemical functional indicators remained stable in the sera from nanovaccine immunized rabbits (Figure S27B-S27H). These results indicate that nanovaccines were safe for rabbits.’

Appendix Figure 30. Levels of TC (A), ALT (B), TBIL (C), TG (D), AST (E), ALP (F) and UREA (G) in the sera from mice immunized with various vaccines. Sera were collected at week 5 (n=5). Statistical analysis was performed using one-way ANOVA. total cholesterol, TC; alanine transaminase, ALT; total bilirubin, TBIL; triglycerides, TG; aspartate aminotransferase, AST; alkaline phosphatase, ALP.

Appendix Figure 31. Levels of TC (A), ALT (B), TBIL (C), TG (D), AST (E), ALP (F) and UREA (G) in the sera from rabbit immunized with various vaccines. Sera were collected at week 6 (n=5). Statistical analysis was performed using one-way ANOVA.

total cholesterol, TC; alanine transaminase, ALT; total bilirubin, TBIL; triglycerides, TG; aspartate aminotransferase, AST; alkaline phosphatase, ALP.

Reference

1. Zhong L, *et al.* Urgency and necessity of Epstein-Barr virus prophylactic vaccines. *NPJ Vaccines* **7**, 159 (2022).
2. Bu W, *et al.* Immunization with Components of the Viral Fusion Apparatus Elicits Antibodies That Neutralize Epstein-Barr Virus in B Cells and Epithelial Cells. *Immunity* **50**, 1305-1316 e1306 (2019).
3. Malhi H, *et al.* Immunization with a self-assembling nanoparticle vaccine displaying EBV gH/gL protects humanized mice against lethal viral challenge. *Cell Rep Med* **3**, 100658 (2022).
4. Wei CJ, *et al.* A bivalent Epstein-Barr virus vaccine induces neutralizing antibodies that block infection and confer immunity in humanized mice. *Sci Transl Med* **14**, eabf3685 (2022).
5. Yu D, Walker LSK, Liu Z, Linterman MA, Li Z. Targeting T(FH) cells in human diseases and vaccination: rationale and practice. *Nat Immunol* **23**, 1157-1168 (2022).
6. Crotty S. T Follicular Helper Cell Biology: A Decade of Discovery and Diseases. *Immunity* **50**, 1132-1148 (2019).
7. Hjalgrim H, *et al.* Characteristics of Hodgkin's lymphoma after infectious mononucleosis. *N Engl J Med* **349**, 1324-1332 (2003).
8. Sheik-Ali S. Infectious mononucleosis and multiple sclerosis - Updated review on associated risk. *Mult Scler Relat Disord* **14**, 56-59 (2017).
9. Chen H, *et al.* Dose-Dependent Outcome of EBV Infection of Humanized Mice Based on Green Raji Unit (GRU) Doses. *Viruses* **13**, (2021).
10. Delecluse HJ, Hilsendegen T, Pich D, Zeidler R, Hammerschmidt W. Propagation and recovery of intact, infectious Epstein-Barr virus from prokaryotic to human cells. *Proc Natl Acad Sci U S A* **95**, 8245-8250 (1998).
11. Zhong L, *et al.* Targeting herpesvirus entry complex and fusogen glycoproteins with prophylactic and therapeutic agents. *Trends Microbiol*, (2023).
12. Kirschner AN, Sorem J, Longnecker R, Jardetzky TS. Structure of Epstein-Barr virus glycoprotein 42 suggests a mechanism for triggering receptor-activated virus entry. *Structure* **17**, 223-233 (2009).
13. Li Q, Turk SM, Hutt-Fletcher LM. The Epstein-Barr virus (EBV) BZLF2 gene product associates with the gH and gL homologs of EBV and carries an epitope critical to infection of B cells but not of epithelial cells. *J Virol* **69**, 3987-3994 (1995).

14. Wu Q, *et al.* Neutralizing antibodies against EBV gp42 show potent in vivo protection and define novel epitopes. *Emerg Microbes Infect* **12**, 2245920 (2023).
15. Hong J, *et al.* Non-overlapping epitopes on the gHgL-gp42 complex for the rational design of a triple-antibody cocktail against EBV infection. *Cell Rep Med* **4**, 101296 (2023).
16. Martinez-Relimpio AM, Benito M, Perez-Izquierdo E, Teijon C, Olmo RM, Blanco MD. Paclitaxel-Loaded Folate-Targeted Albumin-Alginate Nanoparticles Crosslinked with Ethylenediamine. Synthesis and In Vitro Characterization. *Polymers (Basel)* **13**, (2021).
17. Zhang X, *et al.* Protective anti-gB neutralizing antibodies targeting two vulnerable sites for EBV-cell membrane fusion. *Proc Natl Acad Sci U S A* **119**, e2202371119 (2022).
18. Fukushima A, Yamaguchi T, Ishida W, Fukata K, Taniguchi T, Liu FT, Ueno H. Genetic background determines susceptibility to experimental immune-mediated blepharoconjunctivitis: comparison of Balb/c and C57BL/6 mice. *Exp Eye Res* **82**, 210-218 (2006).
19. Gulley ML, Tang W. Laboratory assays for Epstein-Barr virus-related disease. *J Mol Diagn* **10**, 279-292 (2008).
20. Smatti MK, Al-Sadeq DW, Ali NH, Pintus G, Abou-Saleh H, Nasrallah GK. Epstein-Barr Virus Epidemiology, Serology, and Genetic Variability of LMP-1 Oncogene Among Healthy Population: An Update. *Front Oncol* **8**, 211 (2018).
21. Baiocchi RA, Caligiuri MA. Low-dose interleukin 2 prevents the development of Epstein-Barr virus (EBV)-associated lymphoproliferative disease in scid/scid mice reconstituted i.p. with EBV-seropositive human peripheral blood lymphocytes. *Proc Natl Acad Sci U S A* **91**, 5577-5581 (1994).
22. Mosier DE, Gulizia RJ, Baird SM, Wilson DB. Transfer of a functional human immune system to mice with severe combined immunodeficiency. *Nature* **335**, 256-259 (1988).
23. Tan LC, *et al.* A re-evaluation of the frequency of CD8+ T cells specific for EBV in healthy virus carriers. *J Immunol* **162**, 1827-1835 (1999).
24. Newell EW, Klein LO, Yu W, Davis MM. Simultaneous detection of many T-cell specificities using combinatorial tetramer staining. *Nat Methods* **6**, 497-499 (2009).
25. Bentzen AK, *et al.* Large-scale detection of antigen-specific T cells using peptide-MHC-I multimers labeled with DNA barcodes. *Nat Biotechnol* **34**, 1037-1045 (2016).
26. Sun C, *et al.* A gB nanoparticle vaccine elicits a protective neutralizing antibody response

- against EBV. *Cell Host Microbe* **31**, 1882-1897 e1810 (2023).
27. Singh S, *et al.* Neutralizing Antibodies Protect against Oral Transmission of Lymphocryptovirus. *Cell Rep Med* **1**, (2020).
 28. Zhu QY, *et al.* A potent and protective human neutralizing antibody targeting a novel vulnerable site of Epstein-Barr virus. *Nat Commun* **12**, 6624 (2021).
 29. Hong J, *et al.* A Neutralizing Antibody Targeting gH Provides Potent Protection against EBV Challenge In Vivo. *J Virol* **96**, e0007522 (2022).
 30. Choi IK, *et al.* Mechanism of EBV inducing anti-tumour immunity and its therapeutic use. *Nature* **590**, 157-162 (2021).
 31. Zhang B, Choi IK. Facts and Hopes in the Relationship of EBV with Cancer Immunity and Immunotherapy. *Clin Cancer Res* **28**, 4363-4369 (2022).
 32. Cunningham AL, *et al.* Efficacy of the Herpes Zoster Subunit Vaccine in Adults 70 Years of Age or Older. *N Engl J Med* **375**, 1019-1032 (2016).
 33. Moss P. The T cell immune response against SARS-CoV-2. *Nat Immunol* **23**, 186-193 (2022).
 34. Pulendran B, P SA, O'Hagan DT. Emerging concepts in the science of vaccine adjuvants. *Nat Rev Drug Discov* **20**, 454-475 (2021).
 35. Vollmer J, *et al.* Characterization of three CpG oligodeoxynucleotide classes with distinct immunostimulatory activities. *Eur J Immunol* **34**, 251-262 (2004).
 36. Casella CR, Mitchell TC. Putting endotoxin to work for us: monophosphoryl lipid A as a safe and effective vaccine adjuvant. *Cell Mol Life Sci* **65**, 3231-3240 (2008).
 37. Raman VS, *et al.* Applying TLR synergy in immunotherapy: implications in cutaneous leishmaniasis. *J Immunol* **185**, 1701-1710 (2010).
 38. Madan-Lala R, Pradhan P, Roy K. Combinatorial Delivery of Dual and Triple TLR Agonists via Polymeric Pathogen-like Particles Synergistically Enhances Innate and Adaptive Immune Responses. *Sci Rep* **7**, 2530 (2017).
 39. Reed SG, Orr MT, Fox CB. Key roles of adjuvants in modern vaccines. *Nat Med* **19**, 1597-1608 (2013).
 40. Yan W, Chen W, Huang L. Reactive oxygen species play a central role in the activity of cationic liposome based cancer vaccine. *J Control Release* **130**, 22-28 (2008).

41. Roth GA, Picece V, Ou BS, Luo W, Pulendran B, Appel EA. Designing spatial and temporal control of vaccine responses. *Nat Rev Mater* **7**, 174-195 (2022).
42. Vink P, Shiramoto M, Ogawa M, Eda M, Douha M, Heineman T, Lal H. Safety and immunogenicity of a Herpes Zoster subunit vaccine in Japanese population aged ≥ 50 years when administered subcutaneously vs. intramuscularly. *Hum Vaccin Immunother* **13**, 574-578 (2017).
43. Gu P, Zhang Y, Cai G, Liu Z, Hu Y, Liu J, Wang D. Administration Routes of Polyethylenimine-Coated PLGA Nanoparticles Encapsulating Angelica Sinensis Polysaccharide Vaccine Delivery System Affect Immune Responses. *Mol Pharm* **18**, 2274-2284 (2021).

REVIEWER COMMENTS

Reviewer #1 (Remarks to the Author):

Manuscript Nr: NCOMMS-23-44668A

Zhong et al., "A cocktail nanovaccine targeting key entry glycoproteins elicits high-quality neutralizing antibodies against EBV infection"

The authors first correlated Epstein Barr virus (EBV) glycoprotein (gp350, gH, gL, gB and gp42) specific antibodies of healthy virus carriers with EBV neutralizing activity for B and epithelial cell infection. They then constructed nanovaccines containing recombinant gHgL, gB or gp42 proteins plus TLR9 or TLR4 agonists as adjuvants. According to their involvement in B and epithelial cell infection gHgL containing nanoparticles induced the highest neutralizing titers against B and epithelial cell infection and gp42 containing nanoparticles induced neutralizing activity for B cell infection. The nanoparticles were also able to mature dendritic cells and drain to lymph nodes for prolonged antigen retention. The authors then combined their three nanoparticle vaccines to elicit even higher EBV neutralizing antibody titers. With defined monoclonal antibodies against gHgL, gB and gp42 of highly EBV neutralizing ability it is then shown that many of the specificities that are elicited with the cocktail of nanoparticle vaccines overlap with the respective neutralizing epitopes. Immunizations were then repeated in rabbits with similar vaccination efficacy and antigen hierarchy. Rabbit IgG was then used to protect SCID-beige mice with human peripheral blood mononuclear cell (PBMC) transfer from EBV infection. Indeed, protection from EBV infection induced weight loss, viremia and lymphoma formation could be achieved. From these data the authors conclude that a mixed nanoparticle vaccine could elicit sterilizing immunity to prevent EBV associated diseases.

In their revised manuscript version, the authors have addressed all of my concerns. They describe increased Tfh induction by their nanoparticle vaccines, they now transfer more limited mouse antiserum levels into recipient PBMC containing mice that then were challenged with EBV. They document that antibody levels wane and sterilizing immunity is lost after 50 days in vaccinated antibody donor mice. Therefore, they have removed the claim to induce sterilizing immunity. While this study is well done and the revisions address all of my concerns, the novelty is still limited by previous studies on vaccination with similar EBV antigens.

Reviewer #2 (Remarks to the Author):

The author's response to reviewers' comments is commendable, as they diligently address each point and strive to conduct all proposed experiments with remarkable speed, yielding perfect results. These include:

1. Generating NP-gp350.
2. Immunizing mice with NP-gp350 and evaluating immunogenicity within 5 weeks.
3. Conducting ELISA and neutralization assays on all immunized mice, as well as repeating these assays on sera collected from previously immunized mice at different timepoints.
4. Performing depletion and neutralization assays.
5. Carrying out new rabbit immunization experiments (n=5) and analyzing immune responses through ELISA and neutralization assays in two different cell types.
6. Reconstituting humanized PBMCs mice within 4 weeks and conducting two sets of passive immunization experiments using sera collected at different time points, characterizing infection outcomes through FACS, IHC, EBER, qPCR, etc.
7. Performing ELISA and qPCR on donors' PBMCs used in transplanting humanized mice. However, clarity is needed regarding the availability and viability of the same donor cells for the second repeats of the new humanized mice.

A few still notable concerns include:

1. The authors cannot assert 100% protection without testing infectivity in the splenocytes and kidneys of infected humanized mice. High titers of EBV DNA have been found in these tissues even when PBMCs show no detection upon challenge.
2. Clarification is needed regarding the number of rabbits per group used in the study, as both n=5 and n=3 are mentioned in the rebuttal and methods, respectively.
3. Supplementary Figure 19 data raises suspicion, particularly regarding the high permissiveness of EBV seropositive donors' PBMCs to infection. Clarification is needed regarding the experimental procedure, analysis, and potential inhibition of T cell activity.

4. The question regarding the impact of introducing previously existing EBV-specific T cells into the infection system is not adequately addressed. Further elucidation on this aspect is required.

Reviewer #3 (Remarks to the Author):

The manuscript has been quite improved after the revision. Below are some remaining issues that need to be addressed before publication.

1. More details are needed for the NP preparation method. For example, the concentrations are provided but the volumes are not. In some cases, the solvent is unknown.

2. The dose is important for the immune responses. For example, the NP-cocktail has 15 ug of total antigens with 5 ug for each antigen. A higher dose of the single nanovaccine such as 15 ug of NP-gp42 may also provide a significantly enhanced immune response. The authors may thus need to do a comparison study with an equivalent dose or explain this.

3. At line 398, the authors stated that “Interestingly, although antibodies against gp350 are the major contributors to the neutralization of EBV infection of B cells in healthy EBV carriers, immunization with NP-cocktail, which does not contain NP-gp350, induced higher B cell neutralizing titers (Figure S24D)”. The authors need to elucidate the specific mechanisms responsible for this phenomenon.

4. The NP-cocktail demonstrates remarkably superior performance compared to individual NP-antigens. Do the NP-cocktails exert combination therapeutic effects or do they induce synergistic effects? If the latter, the underlying mechanism may be discussed/explained.

Thank you again for submitting your revised manuscript "A cocktail nanovaccine targeting key entry glycoproteins elicits high levels of neutralizing antibodies against EBV infection" to Nature Communications. We have now received reports from the reviewers who evaluated the original version. On the basis of their comments (copied below), we have decided to invite an additional revision of your work.

You will see that, while the reviewers find that your revisions improved the manuscript, some important points remain to be addressed. In particular, a revised manuscript will need to provide additional data, details and/or discussion to address all remaining comments raised by the reviewers. Please revise your manuscript, addressing all the remaining issues raised by the reviewers.

Response: We appreciate the reviewers and editors for their suggestions and comments. We have revised our manuscript and addressed the remaining issues. Below, we provide our point-by-point responses and indicate the changes made to the revised manuscript.

Reviewer #1 (Remarks to the Author):

Manuscript Nr: NCOMMS-23-44668A

Zhong et al., “A cocktail nanovaccine targeting key entry glycoproteins elicits high-quality neutralizing antibodies against EBV infection”

The authors first correlated Epstein Barr virus (EBV) glycoprotein (gp350, gH, gL, gB and gp42) specific antibodies of healthy virus carriers with EBV neutralizing activity for B and epithelial cell infection. They then constructed nanovaccines containing recombinant gHgL, gB or gp42 proteins plus TLR9 or TLR4 agonists as adjuvants. According to their involvement in B and epithelial cell infection gHgL containing nanoparticles induced the highest neutralizing titers against B and epithelial cell infection and gp42 containing nanoparticles induced neutralizing activity for B cell infection. The nanoparticles were also able to mature dendritic cells and drain to lymph nodes for prolonged antigen retention. The authors then combined their three nanoparticle vaccines to elicit even higher EBV neutralizing antibody titers. With defined monoclonal antibodies against gHgL, gB and gp42 of highly EBV neutralizing ability it is then shown that many of the specificities that are elicited with the cocktail of nanoparticle vaccines overlap with the respective neutralizing epitopes. Immunizations were then repeated in rabbits with similar vaccination efficacy and antigen hierarchy. Rabbit IgG was then used to protect SCID-beige mice with human peripheral blood mononuclear cell (PBMC) transfer from EBV infection. Indeed, protection from EBV infection induced weight loss, viremia and lymphoma formation could be achieved. From these data the authors conclude that a mixed nanoparticle vaccine could elicit sterilizing immunity to prevent EBV associated diseases.

In their revised manuscript version, the authors have addressed all of my concerns. They describe increased Tfh induction by their nanoparticle vaccines, they now transfer more limited mouse antiserum levels into recipient PBMC containing mice that then were challenged with EBV. They document that antibody levels wane and sterilizing immunity is lost after 50 days in vaccinated antibody donor mice.

Therefore, they have removed the claim to induce sterilizing immunity. While this study is well done and the revisions address all of my concerns, the novelty is still limited by previous studies on vaccination with similar EBV antigens.

Response: We appreciate your valuable suggestions. We agree with you that for all studies of prophylactic vaccines against EBV, the antigen candidates are limited to the glycoproteins involved in the infection process including gp350, gHgL, gp42 and gB. However, a lot of work remains needed for the development of efficient prophylactic vaccines against EBV, including the improvement of immunogenicity, the broadening of the antigen spectrums and the durability of the protective effect. We believe that our study advances our knowledge and its originality contributes to the progress of the field.

Previous attempts to improve the immunogenicity include displaying multiple copies of antigens on protein scaffold including ferritin, I3-01, LS and I53^{1,2,3,4,5,6}. These vaccines improved neutralizing titers against EBV infection compared to soluble antigens. In this study, we designed a different strategy to enhance the immunogenicity of EBV antigens. The antigen and adjuvants were co-encapsulated in a novel and safe nanoparticle system and delivered simultaneously. We also explain the increased antigenicity by an enhanced transportation to lymph nodes and a better internalization by APCs.

The selection and combination of antigens are essential in vaccine development. In this study, we selected key entry glycoproteins (gHgL, gp42 and gB) as the immunogens. For the first time, we determined the immunogenicity of such a cocktail nanovaccine (gHgL, gp42 and gB) and confirmed the superior protective efficiency of the antibodies elicited by this cocktail nanovaccine.

Our comprehensive determination of the neutralizing titers, fusion inhibitory titers, epitopes mapping of serum antibodies, T cell responses and immune memory elicited by the nanovaccines supports the superiority of the individual nanovaccines and NP-cocktail. To further evaluate the durability of passive protection, we present data on the long-term (week 50) protective effect, which suggested that serum antibodies induced by this original NP-cocktail could reduce the incidence of EBV-associated

lymphoma for an extended period.

We believe that the novel nanovaccines in this study provide valuable information for further vaccines studies. We hope that this clarification addresses your concerns about the novelty of our study.

Reviewer #2 (Remarks to the Author):

The author's response to reviewers' comments is commendable, as they diligently address each point and strive to conduct all proposed experiments with remarkable speed, yielding perfect results. These include:

1. Generating NP-g350.
2. Immunizing mice with NP-gp350 and evaluating immunogenicity within 5 weeks.
3. Conducting ELISA and neutralization assays on all immunized mice, as well as repeating these assays on sera collected from previously immunized mice at different timepoints.
4. Performing depletion and neutralization assays.
5. Carrying out new rabbit immunization experiments (n=5) and analyzing immune responses through ELISA and neutralization assays in two different cell types.
6. Reconstituting humanized PBMCs mice within 4 weeks and conducting two sets of passive immunization experiments using sera collected at different time points, characterizing infection outcomes through FACS, IHC, EBER, qPCR, etc.
7. Performing ELISA and qPCR on donors' PBMCs used in transplanting humanized mice. However, clarity is needed regarding the availability and viability of the same donor cells for the second repeats of the new humanized mice.

Response: We appreciate your constructive suggestions. Many group members worked together for four months to address all these questions, which significantly improved our manuscript.

For the PBMCs used for humanized mice reconstitution, approximately 1×10^9 PBMCs can be isolated from ~400 cc blood of one donor, which can be used to reconstitute ~66 SCID-beige mice. Hence, to avoid donor variability within each experiment, we reconstituted the new humanized mice with donor 2. We clarify this situation in the revised supplementary file (Line 448-451).

A few still notable concerns include:

1. The authors cannot assert 100% protection without testing infectivity in the splenocytes and kidneys of infected humanized mice. High titers of EBV DNA have been found in these tissues even when PBMCs show no detection upon challenge.

Response: Thank you for your suggestions. We tested EBV infection in the spleens of humanized mice through determination of EBV DNA copy numbers (Figure 6J) and *in situ* hybridization for EBER (Figure 7A). EBV DNA was undetected in spleens of humanized mice treated with IgG induced by NP-cocktail (also shown as Appendix Figure 1A). Additionally, no hCD20⁺EBER⁺ cells were observed in mice treated with IgG induced by NP-cocktail (also shown as Appendix Figure 1B).

We also measured EBV DNA copy numbers in the kidneys of humanized mice according to your suggestions. Consistent with blood and spleen samples, EBV DNA was undetected in the kidneys of humanized mice treated by IgG induced by NP-cocktail (Supplementary Figure 34, also shown as Appendix Figure 1C). However, EBV DNA copy numbers increased in kidneys of one or two mice in each group treated with IgG elicited by NP-gHgL, NP-gB and Free-cocktail vaccines (Supplementary Figure 34, also shown as Appendix Figure 1C).

To sum up, EBV DNA copy numbers remain undetectable in the blood, spleens and kidneys of humanized mice treated by IgG induced by NP-cocktail, whereas IgG elicited by individual NP-gHgL, NP-gB and NP-gp42 did not. Hence, NP-cocktail conferred superior protection from lethal EBV challenge in humanized mice. We changed ‘100% protection’ to ‘superior protection’ in the revised manuscript (Line 44, 630, 670 and 721). We added the results of EBV DNA copy numbers in kidneys in the revised manuscript (Line 575-581) and updated the corresponding methods (Line 1216).

Appendix Figure 1. EBV infection in spleens and kidneys of humanized mice.

(A) EBV DNA copy numbers in the spleens of humanized mice treated with different IgG. The dashed line indicates the detection limit. Samples of two mice in the PBS group which died on day 7 and 10 were not collected because of decay. Data are shown as mean \pm SEM (n=8 for the NP-gHgL, NP-gB, NP-gp42, NP-cocktail and Free-cocktail groups, n=6 for the PBS group and n=5 for the mock group).

(B) Representative macroscopic spleen and spleen sections stained by hematoxylin and eosin (HE), immunostained for hCD20⁺ and in situ hybridized for EBV-encoded RNA (EBER). Each image is representative of the experimental group (Scale bar=50 μ m).

(C) EBV DNA copy numbers in the kidneys of humanized mice treated with different IgG. The dashed line indicates the detection limit. Samples of two mice in the PBS group which died on day 7 and 10 were not collected because of decay. Data are shown as mean \pm SEM (n=8 for the NP-gHgL, NP-gB, NP-gp42, NP-cocktail and Free-cocktail groups, n=6 for the PBS group and n=5 for the mock group).

Statistical analysis was performed using one-way ANOVA. The color of the asterisks denotes the group with which there is a significant difference, determined by a Sidak multiple comparison test (significant difference: *** $P \leq 0.0002$; **** $P \leq 0.0001$; ns, no significant difference).

2. Clarification is needed regarding the number of rabbits per group used in the study, as both $n=5$ and $n=3$ are mentioned in the rebuttal and methods, respectively.

Response: Thank you for your comments. Five rabbits were immunized with different vaccine formulations to evaluate the immunogenicity of the vaccines. IgGs were purified from three immunized rabbits to determine the passive protective efficiency. We clarified this in the revised manuscript (Line 1135-1136).

3. Supplementary Figure 19 data raises suspicion, particularly regarding the high permissiveness of EBV seropositive donors' PBMCs to infection. Clarification is needed regarding the experimental procedure, analysis, and potential inhibition of T cell activity.

Response: Thank you for your comments. We clarified the experiment procedure, analysis and T cell activities in detail in the revised manuscript results section (Line 531-537) and Material and Methods section (Line 1158-1175). Briefly, 1×10^5 PBMCs from each donor were seeded in 96-well plates. 100 μ l serially diluted CNE2-EBV-GFP (started from $1 \times$) were added into each well and incubated for 48 h. Akata B cells (EBV negative) incubated with EBV were used as the positive control. PBMCs without incubation with EBV were used as the negative control. After a 48h incubation, PBMCs were washed with PBS and stained by anti-human CD19-APC. GFP positive CD19⁺ B cells represented the EBV infected B cells in PBMCs *in vitro* (Supplementary Figure 31A, also shown as appendix Figure 2A).

Consistent with previous studies, the results indicated that B cells in donors' PBMCs used for humanized mice reconstitution were susceptible to EBV infection *in vitro* (Supplementary Figure 31B, also shown as appendix Figure 2B)^{7, 8, 9}. We also determined that the percentage of GFP⁺ CD19⁺ B cells increased when suppressed T

cells activities by Cyclosporine A (CsA) (Supplementary Figure 31C, also shown as appendix Figure 2C). Hence, at least 48 h after infection, the majority of B cells in donors' PBMCs were infected by EBV and T cells only killed a fraction of infected B cells. Moreover, inhibition of T cell activity by CsA is beneficial to the generation of lymphoblastoid cell lines (LCLs) from infected B cells *in vitro* ⁹.

Appendix Figure 2. *In vitro* EBV infection of B cells in donors' PBMCs used for reconstitution of humanized mice.

(A) Gating strategies to analyze EBV infected B cells *in vitro*. 1×10^5 PBMCs from each donor were seeded in 96-well plate and serially diluted CNE2-EBV-GFP was added. After 48 h incubation, the percentage of GFP positive B cells was detected by flow cytometry. Cells without incubation with EBV were used as a negative control.

(B) Dose-dependent infection of B cells in donors' PBMCs infected by serially diluted CNE2-EBV-GFP virus. The gating strategy is shown in panel A. 1×10^5 Akata cells (EBV negative) infected by serial diluted CNE2-EBV-GFP were used as the positive control.

(C) Percentages of CNE2-EBV-GFP virus ($1 \times$) infected B cells in donors' PBMCs treated with/without Cyclosporine A (CsA). Data are shown as mean \pm SEM (two donors). Statistical analysis was performed using Welch's t tests.

4. The question regarding the impact of introducing previously existing EBV-specific T cells into the infection system is not adequately addressed. Further elucidation on this aspect is required.

Response: Thank you for your comments. We further determined the percentage of activated (hCD69⁺hCD137⁺) CD8⁺ T cells in the spleen of humanized mice at the end of the experiment. Consistent with the previous studies by Andrew T. McGuire *et al.*, and our group, an increase of hCD69⁺hCD137⁺ activated CD8⁺ T cell populations was observed in unprotected humanized mice in this study, which still developed lymphoma after EBV challenge (Supplementary Figure 37, also shown as appendix Figure 3) ^{10, 11, 12}. However, the specific mechanisms of inefficient killing of EBV infected B cells by the activated T cells remain to be determined. Previously existing EBV-specific T cells were introduced to the *in vivo* infection system. However, we used a lethal dose of EBV in the passive protection assay, which caused mice from control group (PBS-IgG) all died to die and 5 out of 8 mice developed EBV-associated lymphoma. The control group and the experimental group were reconstituted by PBMCs from the same donor and challenged by the same batch of EBV at the same time. Thus, the previously existing EBV-specific T cells may not effectively kill infected B cells under conditions where there is a relatively high EBV dose and a very low percentage of EBV-specific T cells ^{13, 14, 15}. To sum up, we believe that the previous existing EBV-specific T cells did not influence the results of the passive protection efficiency of the IgGs.

We added the results in the revised manuscript (Line 619-628) and updated the methods (Line 1206-1214).

Appendix Figure 3. Percentage of hCD69⁺hCD137⁺hCD8⁺ T cells in humanized mice splenocytes. Data are shown as mean \pm SEM (n=8 for the NP-gHgL, NP-gB, NP-gp42, NP-cocktail and Free-cocktail groups, n=6 for the PBS group and n=5 for the mock group). Statistical analysis was performed using one-way ANOVA. The color of the asterisks denotes the group with which there is a significant difference, determined by a Sidak multiple comparison test (significant difference: *P \leq 0.0332; **P \leq 0.0021; ns, no significant difference).

Reviewer #3 (Remarks to the Author):

The manuscript has been quite improved after the revision. Below are some remaining issues that need to be addressed before publication.

Response: Thank you for your supportive comments and suggestions. We addressed all the remaining issues below.

1. More details are needed for the NP preparation method. For example, the concentrations are provided but the volumes are not. In some cases, the solvent is unknown.

Response: Thank you for your suggestions. The specific volume of each reagent is adjustable depending on the total amount of vaccines needed. The volume ratio is fixed at mixture 1: mixture 2: PLGA: DSPE-PEG2000 = 1:18:18:120. We provided the detailed concentrations and solvent of the NP preparation in the revised manuscript (Line 830-849).

2. The dose is important for the immune responses. For example, the NP-cocktail has 15 ug of total antigens with 5 ug for each antigen. A higher dose of the single nanovaccine such as 15 ug of NP-gp42 may also provide a significantly enhanced immune response. The authors may thus need to do a comparison study with an equivalent dose or explain this.

Response: Thank you for your comments. We agree with you that the dose of the antigens is important for the immune responses. Actually, we performed the dose comparison study of NP-gHgL before. Mice were immunized by 5 µg or 15 µg NP-gHgL at week 0, 2 and 4. The humoral immune responses generated by NP-gHgL were dose dependent. 15 µg NP-gHgL induced higher antigen-specific antibodies than 5 µg NP-gHgL (Supplementary Figure 25A, also shown as appendix Figure 4A). However, the B cell neutralizing titers of sera from mice immunized with 15 µg NP-gHgL were only slightly higher than 5 µg NP-gHgL (Supplementary Figure 25B, also shown as appendix Figure 4B). In comparison, the B cell infection neutralizing

titers of sera induced by 15 μg NP-gHgL is still much lower than that elicited by NP-cocktail (2.5, log₁₀).

To sum up, increasing the dose of a single antigen increased the binding titers of sera from immunized mice, but the neutralizing titers were not efficiently improved as NP-cocktail immunization. Hence, broadening the antigen spectrum rather than solely increasing the dose of the single antigen is a superior strategy to improve neutralizing abilities, considering that EBV infection involves multiple glycoproteins. We added these results in the revised manuscript (Line 409-414) and the specific composition of these vaccines in the supplementary table 2 (Line 440).

Appendix Figure 4. Dose-dependent antibody responses of NP-gHgL.

(A) Total IgG titers of sera collected on day 35 from C57BL/6J mice immunized with 15 μg or 5 μg NP-gHgL.

(B) Neutralization of CNE2-EBV infection of Akata B cells by sera collected on day 35 from C57BL/6J mice immunized with 15 μg or 5 μg NP-gHgL.

Data points are shown as the mean \pm SEM (n=5). Statistical analysis was performed using Welch's t tests.

3. At line 398, the authors stated that “Interestingly, although antibodies against gp350 are the major contributors to the neutralization of EBV infection of B cells in healthy EBV carriers, immunization with NP-cocktail, which does not contain NP-gp350, induced higher B cell neutralizing titers (Figure S24D)”. The authors need to elucidate the specific mechanisms responsible for this phenomenon.

Response: Thank you for your comments. During the B cell infection process, gp350 binds to CR2 to close the distance of virus and host cell. Then, the gHgLgp42 complex binds to HLA-II and activates gB to induce membrane fusion¹⁶. Hence, compared to gp350, gHgL, gp42 and gB play more essential roles in the EBV infection process. Considering their more important functions, NP-cocktail targeting key entry glycoproteins induced higher sera neutralizing titers compared to NP-gp350, which is also consistent with previous studies^{3,17}. We added this clarification in the revised manuscript (Line 402-404).

4. The NP-cocktail demonstrates remarkably superior performance compared to individual NP-antigens. Do the NP-cocktails exert combination therapeutic effects or do they induce synergistic effects? If the latter, the underlying mechanism may be discussed/explained.

Response: Thank you for your comments. NP-cocktail vaccine contains three antigens playing key roles in EBV entry process so that NP-cocktail induced antibodies targeting multiple steps of the infection. We thought that the superior performance of NP-cocktail is a combination therapeutic effect of antibodies targeting different antigens.

Reference

1. Wei CJ, *et al.* A bivalent Epstein-Barr virus vaccine induces neutralizing antibodies that block infection and confer immunity in humanized mice. *Sci Transl Med* **14**, eabf3685 (2022).
2. Malhi H, *et al.* Immunization with a self-assembling nanoparticle vaccine displaying EBV gH/gL protects humanized mice against lethal viral challenge. *Cell Rep Med* **3**, 100658 (2022).
3. Bu W, *et al.* Immunization with Components of the Viral Fusion Apparatus Elicits Antibodies That Neutralize Epstein-Barr Virus in B Cells and Epithelial Cells. *Immunity* **50**, 1305-1316 e1306 (2019).
4. Kanekiyo M, *et al.* Rational Design of an Epstein-Barr Virus Vaccine Targeting the Receptor-Binding Site. *Cell* **162**, 1090-1100 (2015).
5. Kang YF, *et al.* Immunization with a Self-Assembled Nanoparticle Vaccine Elicits Potent Neutralizing Antibody Responses against EBV Infection. *Nano Lett* **21**, 2476-2486 (2021).
6. Sun C, *et al.* A gB nanoparticle vaccine elicits a protective neutralizing antibody response against EBV. *Cell Host Microbe* **31**, 1882-1897 e1810 (2023).
7. Crotty S. T Follicular Helper Cell Biology: A Decade of Discovery and Diseases. *Immunity* **50**, 1132-1148 (2019).
8. Ehlin-Henriksson B, Gordon J, Klein G. B-lymphocyte subpopulations are equally susceptible to Epstein-Barr virus infection, irrespective of immunoglobulin isotype expression. *Immunology* **108**, 427-430 (2003).
9. Nagy N. Establishment of EBV-Infected Lymphoblastoid Cell Lines. *Methods Mol Biol* **1532**, 57-64 (2017).
10. Snijder J, *et al.* An Antibody Targeting the Fusion Machinery Neutralizes Dual-Tropic Infection and Defines a Site of Vulnerability on Epstein-Barr Virus. *Immunity* **48**, 799-811 e799 (2018).
11. Zhang X, *et al.* Protective anti-gB neutralizing antibodies targeting two vulnerable sites for EBV-cell membrane fusion. *Proc Natl Acad Sci U S A* **119**, e2202371119 (2022).
12. Hong J, *et al.* Non-overlapping epitopes on the gHgL-gp42 complex for the rational design of a triple-antibody cocktail against EBV infection. *Cell Rep Med* **4**, 101296 (2023).
13. Tan LC, *et al.* A re-evaluation of the frequency of CD8+ T cells specific for EBV in healthy virus carriers. *J Immunol* **162**, 1827-1835 (1999).
14. Newell EW, Klein LO, Yu W, Davis MM. Simultaneous detection of many T-cell specificities using combinatorial tetramer staining. *Nat Methods* **6**, 497-499 (2009).

15. Bentzen AK, *et al.* Large-scale detection of antigen-specific T cells using peptide-MHC-I multimers labeled with DNA barcodes. *Nat Biotechnol* **34**, 1037-1045 (2016).
16. Mohl BS, Chen J, Longnecker R. Gammaherpesvirus entry and fusion: A tale how two human pathogenic viruses enter their host cells. *Adv Virus Res* **104**, 313-343 (2019).
17. Cui X, Cao Z, Chen Q, Arjunaraja S, Snow AL, Snapper CM. Rabbits immunized with Epstein-Barr virus gH/gL or gB recombinant proteins elicit higher serum virus neutralizing activity than gp350. *Vaccine* **34**, 4050-4055 (2016).

REVIEWERS' COMMENTS

Reviewer #3 (Remarks to the Author):

The authors have adequately the reviewer's comments. New results also further strengthen the manuscript, which could now be accepted for publication in Nature Communications.